Reducing climate model biases by exploring parameter space with large ensembles of
climate model simulations and statistical emulation
Sihan Li[1,2], David E. Rupp[3], Linnia Hawkins[3,6], Philip W. Mote[3,6], Doug McNeall[4], Sarah
N. Sparrow[2], David C. H. Wallom[2], Richard A. Betts[4,5], Justin J. Wettstein[6,7,8]
[1]Environmental Change Institute, School of Geography and the Environment, University
of Oxford, Oxford, United Kingdom
[2]Oxford e-Research Centre, University of Oxford, Oxford, United Kingdom
[3]Oregon Climate Change Research Institute, College of Earth, Ocean, and Atmospheric
Science, Oregon State University, Corvallis, Oregon
[4]Met Office Hadley Centre, FitzRoy Road, Exeter, United Kingdom
[5]College of Life and Environmental Sciences, University of Exeter, Exeter, UK
[6]College of Earth, Ocean, and Atmospheric Science, Oregon State University, Corvallis,
Oregon
[7]Geophysical Institute, University of Bergen, Bergen, Norway
[8]Bjerknes Centre for Climate Change Research, Bergen, Norway
*Correspondence to*: Sihan Li (sihan.li@ouce.ox.ac.uk)
**Abstract**
Understanding the unfolding challenges of climate change relies on climate models, many
of which have large summer warm and dry biases over Northern Hemisphere continental
mid-latitudes. This work, using the example of the model used in the updated version of
the weather@home distributed climate model framework, shows the potential for
improving climate model simulations through a multi-phased parameter refinement
approach, particularly over northwestern United States (NWUS). Each phase consists of 1)
creating a perturbed parameter ensemble with the coupled global - regional atmospheric
model, 2) building statistical emulators that estimate climate metrics as functions of
parameter values, 3) and using the emulators to further refine the parameter space. The
refinement process includes sensitivity analyses to identify the most influential parameters
for various model output metrics; results are then used to cull parameters with little
influence. Three phases of this iterative process are carried out before the results are
considered to be satisfactory; that is, a handful of parameter sets are identified that meet
acceptable bias reduction criteria. Results not only indicate that 74% of the NWUS regional
warm biases can be reduced by refining global atmospheric parameters that control
convection and hydrometeor transport, and land surface parameters that affect plant
photosynthesis, transpiration and evaporation, but also suggest that this iterative approach
to perturbed parameters has an important role to play in the evolution of physical
parameterizations.

**Introduction**

Boreal summer (June-July-August, JJA) warm and dry biases over North Hemisphere (NH)
continental midlatitudes are common in many global and regional climate models (e.g.,
Boberg and Christensen, 2012; Mearns et al., 2012; Mueller and Seneviratne, 2014;
Kotlarski et al., 2014; Cheruy et al., 2014; Merrifield and Xie, 2016), including very high
resolution convection-permitting models (e.g. Liu et al., 2017). These biases can have non-
negligible impacts on climate change studies, particularly where relationships are non-
linear, such as is the case of surface latent heat flux as a function of water storage (e.g.
Rupp et al., 2017). Biases in present-day climate model simulations reduce the reliability
of the future climate projections from those models. As shown by Boberg and Christensen
(2012), after applying a bias correction conditioned on temperature to account for model
deficiencies, the Mediterranean summer temperature projections were reduced by up to
1°C. Cheruy et al. (2014) demonstrated that of the climate models contributing to the
Coupled Model Intercomparison Project Phase5 (CMIP5), the models that simulate a
higher-than-average warming overestimated the present climate net shortwave radiation
which increased more than multi-model average in the future; those models also showed a
higher-than-average reduction of evaporative faction in areas with soil moisture-limited
evaporation regimes. Both studies suggested that models with a larger warm bias in surface
temperature tend to overestimate the projected warming. The implication of the warm bias
goes beyond climate model simulations, as many impact modeling (e.g. hydrological, fire,
crop modeling) studies (e.g. Brown et al., 2004; Fowler et al., 2007; Hawkins et al., 2013;
Rosenzweig et al., 2014) use climate model simulation results as driving data. Recently,
there have been coordinated research efforts (Morcrette et al., 2018; van Weverberg et al.,
2018; Ma et al., 2018; Zhang et al., 2018) to better understand the causes of the near-surface
atmospheric temperature biases through process level understanding and to identify the
model deficiencies that generate the bias. These studies suggest that biases in the net
shortwave and downward longwave fluxes as well as surface evaporative fraction are
contributors to surface temperature bias.
In the aforementioned climate models, many small-scale atmospheric processes have
significant impacts on large-scale climate states. Processes such as precipitation formation,
radiative balance, and convection, occur at scales smaller than the spatial resolution
explicitly resolved by climate models, though very high resolution regional climate models
are able to resolve or partially resolve some of these processes (e.g., convection). These
processes must be represented by parameterizations that include parameters whose
uncertainty are often high because: 1) there are insufficient observations with which to
constrain the parameters, 2) a single parameter is inadequate to represent the different ways
a process behaves across the globe, and/or 3) there is incomplete understanding of the
physical process (Hourdin et al., 2013). Many studies have demonstrated the importance
of considering parameterization uncertainty in the simulation of present and future climates
by perturbing single and multiple model parameters within plausible parameter ranges
usually established by expert judgment (e.g., Murphy et al., 2004; Stainforth et al., 2005;
Sanderson et al., 2008a, b, 2010, 2011; Collins et al., 2011; Bellprat et al., 2012a,b, 2016).
These studies have argued for careful tuning of models not only to reduce model parameter
uncertainties by selecting parameter values that result in a better match between model
simulation results with observations, but also to better understand relationships among
physical processes within the climate system via systematic experiments that alter
individual parameter values or combinations thereof, in order to assess model responses to
perturbing parameters.
Older generation Hadley Centre coupled models (HadCM2 and HadCM3), and
atmospherere-only global (HadAM) and regional (HadRM) models have been used in
numerous attribution studies (e.g., Tett et al., 1996; Stott et al., 2004; Otto et al., 2012;
Rupp et al., 2017a; van Oldenborgh et al., 2016; Schaller et al., 2016; van Oldenborgh et
al., 2017; Uhe et al., 2018), and the same models have been used for future projections
(e.g., Rupp and Li, 2017; Rupp et al., 2017b; Guillod et al., 2018).  These model families
exhibit warm and dry biases during JJA over continental midlatitudes, biases that have
persisted over model generations and enhancements (e.g., Massey et al., 2015; Li et al.,
2015; Guillod et al., 2017). The more recent generations of Hadley Centre models –
HadGEMx (HadGEM1, Johns et al, 2016;  HadGEM2, Collins et al., 2008 ) also have the
same biases to some extent.

Many of the aforementioned studies using HadAM and HadRM generated simulations
through a distributed computing system known as climateprediction.net (CPDN, Allen et
al., 1999), within which a system called weather@home is used to dynamically downscale
global simulations using regional climate models (Massey et al., 2015; Mote et al., 2016;
Guillod et al., 2017).  As with the previous version of weather@home, the current
operational version of weather@home (version 2: weather@home2) uses the coupled
HadAM3P/HadRM3P with the atmosphere component based on HadCM3 (Gordon et al.,
2000), but updates the land surface scheme from the Met Office Surface Exchange Scheme
version 1 (MOSES1, Cox et al., 1999) to version 2(MOSES2, Essery et al., 2003).

Although the current model version in weather@home2 produces some global-scale
improvements in the global model's simulation of the seasonal mean climate, warm biases
in JJA increase over North America north of roughly 40° compared with the previous
version in weather@home1 (Fig. 2 in Guillod et al., 2017).  The warm and dry JJA biases
appear clearly in the regional model simulations over the northwestern US region (NWUS,
defined here as all the continental US land points west of 110° and between 40°N-49°N -
the grey bounding box in Fig.S1). These biases may be related to, among other things, an
imperfect parameterization of certain cloud processes, leading to excess downward solar
radiation at the surface, which in turn triggers warm and dry summer conditions that are
further amplified by biases in the surface energy and water balance in the land surface
model (Sippel et al., 2016; Guillod et al., 2017).  The fact that recent model enhancements
did not reduce biases over most of the northwest US motivates the present study, which
aims at reducing these warm/dry biases by way of adjusting parameter values, herein
referred to as 'parameter refinement'.

Improving a model by parameter refinement can be an iterative process of modifying
parameter values, running a climate simulation, comparing model output to observations,
and refining the parameter values again (Mauritsen et al., 2012; Schirber et al., 2013).  This
iterative process can be both computationally expensive and labor-intensive. Any
parameter refinement process performed with the intent of improving the model also
involves unavoidably arbitrary decisions - though guided by expert judgement - about
which parameter(s) to adjust, which metric(s) to evaluate (i.e., which feature(s) of the
climate system to simulate at some level of accuracy), and which observational dataset(s)
to use as the basis for the evaluation metric(s). Nonetheless, model tuning through
parameter refinement is invariably needed to better match model simulations with
observations (Schirber et al., 2013).

One systematic, yet computationally demanding, approach to model tuning is through
perturbed parameter experiments (Allen et al., 1999; Murphy et al., 2004). These
experiments use a perturbed parameter ensemble (PPE) of simulations from a single model
where a handful of uncertain model parameters are varied systematically or randomly. Each
set of perturbed parameter (PP) values is considered to be a different model variant - a PP
set refers to a combination of parameter values from herein on. PPEs can be treated as a
sparse sample of behaviours from a vast, high-dimensional parameter space (Williamson
et al., 2013). A PPE directly informs us about model behaviour at those points in parameter
space where the model is run (the PP sets), and helps us infer model behavior in nearby
parameter space where the model has not been run. Besides parameter refinement, PPEs
have also been used in many studies to estimate probability distribution functions (PDFs)
of equilibrium climate sensitivity (e.g., Murphy et al., 2004) and transient regional climate
change (e.g., Sexton et al., 2012a,b), permitting probabilistic projection of climate change
(Murphy et al., 2007, 2009; Harris et al., 2013). PPEs are becoming common as a means
to assess the range of uncertainty in climate model projections (Murphy et al., 2004;
Stainforth et al., 2005; Collin et al., 2006; Sanderson, 2011; Sexton et al., 2012a,b2019;
Shiogama et al., 2012; Karmalkar et al., 2019).

Studies of climate model tuning using PPEs generally fall into three categories. The first
category makes only direct use of the ensemble itself (e.g., Murphy et al., 2004; Rowlands
et al., 2012) by screening out ensemble members that are deemed too far from the observed
target metrics. This is often referred to as ensemble filtering. However, this approach can
overlook certain critical parts of the parameter space not sampled by the PPE. One
promising improvement of this approach is to estimate the response of metric(s) in a
geophysical (e.g., atmospheric) model to parameter perturbations using a computationally
efficient statistical model (i.e. emulator) that is trained from the PPE results. The emulator's
skill is evaluated based on its metric prediction accuracy using independent simulations of
the model and, if deemed sufficiently skilful, can be used to estimate the model's output
metrics as a function of the model parameters in the parameter space not sampled by the
PPE.

The second category uses a PPE to train a statistical emulator, or establish some cost
function, which is then used to automatically search for optimal parameter values that
produce simulations closest to observations (e.g., Bellprat et al., 2012a, 2016; Zhang et al.,
2015; Tett et al., 2017). Different approaches have been used in optimization, ranging from
ensemble Kalman filters (Annan et al., 2005; Annan and Hargreaves, 2007 and the
references therein), stochastic Bayesian approach (e.g. Jackson et al., 2004), Markov chain
Monte Carlo integrations (Jackson et al., 2008; Järvinen et al., 2010), as well as
optimization over multiple objectives (Neelin et al., 2010). These studies advocated for this
approach particularly because of the efficiency and automation of available searching
algorithms. However, as with any model evaluation effort, the use of a cost function with
multiple target metrics means that optima for different metrics may occur at different
parameter values. This approach (automatically searching for optimal parameters) also runs
the risk of being trapped into local minima in the associated cost function; thus, searching
results are heavily dependent on the initial parameter values. Admittedly, the idea of
automatic searching to obtain optimal combinations of model parameters is appealing, but
in reality there is still a high level of subjectivity, e.g. selecting which model performance
metrics and observation(s) to use in evaluating the model, and the methods of optimization
and searching algorithm.

Unlike the second category, which searches for the optimal parameter values that result in
the closest match to observations, the third category, named 'history matching' (McNeall
et al., 2013, 2016; Williamson et al., 2013, 2015, 2017), seeks to rule out parameter choices
that do not adequately reproduce observations. History matching uses PPEs to train
statistical emulators that predict key metrics from the model output, and then uses the
emulators to rule out parameter space that is implausible. Williamson et al. (2017)
demonstrated that this method is more powerful when iterative steps are taken to rule out
implausible parameter space, where each step helps refine the parameter space containing
potentially better performing model variants. A drawback is that iterative history matching
requires more model runs in the not-ruled-out-yet parameter space for later iterations. It is
worth pointing out that the second and the third categories may not be different from each
other if a sufficient number of model simulations are used to train a statistical emulator
over the full parameter space. With a good emulator, it is possible to rule out parameter
space and optimize parameter values, in which case categories two and three are post-
processing steps. The method we adopted in this study fits into the third category,
borrowing the idea of 'iterative refocusing' where parameter values are refined through
phases of experiments. Our methodology differs from history matching in that we do not
employ a formal statistical framework based on the definition of implausibility.

All three approaches begin with an initial PPE, which can be computationally expensive
even with a modest number of free parameters. To cope with the computational demand,
many previous studies have generated PPEs from a global climate model (GCM) using
CPDN. The studies span a range of topics, from the earlier studies focusing on climate
sensitivity (e.g., Murphy et al., 2004; Stainforth et al., 2005; Sanderson et al., 2008a,b,
2010, 2011), to later ones attempting to generate plausible representations of the climate
without flux adjustments (e.g. Irvine et al., 2013; Yamazaki et al., 2013) and using history
matching to reduce parameter space uncertainty (Williamson et al., 2013). More recently,
Mulholland et al. (2016) demonstrated the potential of using PPEs to improve the skill of
initialized climate model forecasts of 1 month lead time, and Sparrow et al. (2018) showed
that large PPE can be used to identify subgrid scale parameter settings that are capable of
best simulating the ocean state over the recent past (1980-2010). However, very little
(Bellprat et al., 2012b; 2016) has been published on using PPEs for parameter refinement
with the aim of improving regional climate models (RCMs).

The goals of this study were to: 1) identify model parameters that most strongly control the
annual cycle of near-surface temperature and precipitation over the NWUS in
weather@home2, and 2) select model parameterizations that reduce the warm/dry summer
biases without introducing or unduly increasing other biases. We acknowledge that
changing a model in any way inevitably involves making sequences of choices that
influence the behaviour of the model. Some of the model behavioural changes are targeted
and desirable, but parameter refinement may have unintended negative consequences.
There is a general concern that 'improved' performance arises because of compensation
among model errors, and an 'accurate' climate simulation may very well be achieved by
compensating errors in different processes, rather than by best simulating every physical
process. This concern motivated us to select multiple parameter sets from the tuning
exercise rather than seek an "optimal" set. Though having multiple parameter sets does not
eliminate the problem, to the degree that each parameter set compensates for errors
uniquely, obtaining a similar model response to some change in forcing across parameter
sets may provide more confidence in that response. An alternative approach would be to
interpolate between the sampled points in the parameter space, and estimate a posterior
parameter probability density function (PDF), which could then be used to produce a PDF
of model outputs of interests (e.g., Murphy et al., 2004; Sexton et al., 2012a,b). We chose
to select multiple parameter sets instead of using parameter PDFs because the intended use
is to make projections with a small ensemble of parameter sets with reduced biases in
summer temperature and precipitation.

It is worth noting that this study looks mainly at atmospheric parameters because we
intended to focus this study on larger-scale atmospherics dynamics that influence the
boundary conditions of the regional model, especially how much moisture and heat is
advected to the regional model, while local land surface/atmosphere interactions are being
examined in a subsequent study that perturbs a suite of atmospheric and land surface
parameters in the regional model.

**2. Methodology**
Throughout this paper we use 'simulated' to refer to outputs from climate models, and
'emulated' results to refer to estimated/predicted outputs from statistical emulators.

**2.1. Overview of the parameter refinement process**
This study carried out an iterative parameter refinement exercise, or an 'iterative
refocusing' procedure to use a term coined in Williamson et al. (2017). The multi-
dimensional parameter space is reduced in phases, where each phase includes the following
steps:
1) Using space-filling Latin hypercube sampling (McKay et al., 1979) to randomly sample
the initially defined parameter space (defined by the bounds of the 17 parameters listed in
Table1) to generate sets of parameter combinations;
2) generate a PPE with the parameters sets from step (1) through weather@home;
3) train statistical emulators for multiple climate metrics using the PPE from step (2);
4) reduce the parameter space (i.e., narrow the ranges of acceptable values for parameters)
such that the space excludes ensemble parameter sets that are 'too far away' from target
metrics;
5) randomly sample the reduced parameter space to design a new set of parameter
combinations;
6) use the trained emulators to filter the sample from step (5), and reject a parameter set if
the emulator prediction is too far away from a target value;
7) repeat steps (2) through (6) until the desired outcome is achieved.
Detailed descriptions of the parameter refinement process throughout three phases is
presented in Appendix A, including decisions on what key climate metrics to use in each
phase, and the stopping point of this iterative exercise - after three phases.

Here we briefly summarize the objective of each phase. The objective of Phase 1 was to
eliminate regions of parameter space that led to top-of-atmosphere (TOA) radiative fluxes
that are too far out of balance. The objective of Phase 2 was to reduce biases in the
simulated regional climate of NWUS, while not straying too far away from TOA radiative
(near-) balance. Lastly, the objective of Phase 3 was to further refine parameter space,
specifically to reduce the JJA warm and dry bias over the NWUS.

The principle climate metrics used to access the effect of parameter perturbation are: Phase
1) TOA radiative fluxes, where we considered outgoing (reflected) shortwave radiation
(SW) and outgoing longwave radiation (LW) separately; Phase 2) NWUS regional surface
metrics - the mean magnitude of the annual cycle of temperature (MAC-T), and mean
temperature (T) and precipitation (Pr) in December-January-February (DJF) and (JJA),
while still being mindful of SW and LW; and Phase 3) same as Phase 2, except for selecting
model parameterizations that reduce the JJA warm and dry biases over the NWUS.

## 2.2. Climate simulations with weather@home

The climate simulations used in this study were generated through the weather@home
climate modelling system (Massey et al., 2015; Mote et al., 2016) with updates (Guillod et
al., 2017) that includes MOSES2. MOSES2 simulates the fluxes of $CO_2$, water, heat, and
momentum at the interface of the land and atmospheric boundary layer, and is capable of
representing a number of sub-grid tiles within each grid box, allowing a degree of sub-grid
heterogeneity in surface characteristics to be modeled (Williams et al., 2012).

The western North America application of weather@home (weather@home-WNA)
consists of HadRM3P (0.22° × 0.22°) nested within HadAM3P (1.875° longitude ×1.25°
latitude). Weather@home-WNA prior to recent enhancements was evaluated for how well
it reproduced various aspects of the recent historical climate of the western US by Li et al.
(2015), Mote et al. (2016), Rupp and Li (2016), and Rupp et al. (2017). Notable warm/dry
biases in JJA were present over the NWUS and these biases persist with MOSES2 (Fig.
S1), with a temperature bias of 3.9 °C and a precipitation biases of -8.5 mm/month (-32%)
in JJA over Washington, Oregon, Idaho and western Montana, as compared with the
PRISM gridded observational dataset (Daly et al., 2008). Note these were biases using
default, i.e. standard physics (SP), model parameter values.

Each simulation in the PPE spanned 2 years, with the first year serving as spin-up and only
the second year used in the analysis. Simulations began on 1 December of each year for
the years 1995 to 2005, except for Phase 1 (see description of Phases in Appendix A).
Climate metrics were averaged over December 1996 to November 2007 (except Phase 1).
This time period was chosen because it contained a wide range of SST anomaly patterns -
including the very strong 1997-98 El Niño – which helps reduce the influence that any
particular SST anomaly pattern may have on the sensitivities of chosen climate metrics to
parameters.

**2.3. Perturbed parameters**
In our PPE, we initially selected 17 model parameters to perturb simultaneously, 16 in the
atmospheric model, and one in the land surface model (Table 1). The parameters reside in
the global model as well as the regional model, and are set to the same values in HadAM3P
and HadRM3P in the experiments performed for this study, thus any reduction of regional
biases are considered to have been achieved through the improvement of boundary fluxes
from the GCM to the RCM, and improvement of the RCM itself. The atmospheric
parameters are a subset of those perturbed in Murphy et al. (2004) and Yamazaki et al.
(2013); both studies also perturbed ocean parameters, and Yamazaki et al. (2013) perturbed
forcing parameters (e.g., scaling factor for emission from volcanic emissions) as well. Our
selection of parameters was constrained to those available to be perturbed using
weather@home at the time. Ranges for most parameter perturbations were 1/3 to 3 times
the default value, but for certain parameters (e.g., empirically adjusted cloud fraction,
EACF), only values greater than the default value were used (Table 1). We intentionally
began with ranges generally wider than those used in previous studies (Murphy et al. 2004;
Yamazaki et al. 2013) because we intended to refine the ranges through multiple phases of
PPEs.

Though a principal objective was to evaluate sensitivity of the regional climate to
atmospheric parameters, sensitivities may be a function of land-atmosphere exchanges
(Sippel et al., 2016; Guillod et al., 2017).    While many parameters influence land-
atmosphere energy and water exchanges in MOSES2, one (V_CRIT_ALPHA) has been
shown to be particularly important (Booth et al., 2012) so was included in our tuning
exercise.   V_CRIT_ALPHA defines the soil water content below which transpiration
begins being limited by soil water availability and not solely the evaporative demand.

**2.4 Observational data**
The regional biases in MAC-T, JJA-T, JJA-Pr, DJF-T and DJF-Pr  - were all calculated
with respect to the 4-km resolution monthly PRISM dataset, after regridding the PRISM
data to the HadRM3P grid. To consider observational uncertainty, we also compared JJA-
T biases using four other observational datasets: 1) NCEP/NCAR Reanalysis 1 (NCEP,
Kalnay et al., 1996), 2) the Climate Forecast System Reanalysis and Reforecast (CFSR,
Saha et al., 2010), 3) the Modern-Era Retrospective Analysis for Research and
Applications Version2 (MERRA2, Gelaro et al., 2017), and 4) Climatic Research Unit
temperature dataset v4.00 (CRU, Harris et al., 2014).  The four datasets are not shown here
for the regional analysis because the maximum regionally averaged difference (0.71 °C)
among the datasets is less than 1/5 of  the regionally averaged JJA-T bias. Throughout this
paper, biases of the regional model outputs are calculated with respect to PRISM.

The biases in global temperature were calculated with respect to CRU, MERRA2, CSFR,
NCEP, and the Climate Prediction Centre global land surface temperature data; the latter
is a combination of the station observations collected from Global Historical Climatology
Network version 2 and the Climate Anomaly Monitoring System (GHCN-CAMS, Fan and
van den Dool, 2008).  The biases in global precipitation were calculated with respect to
CRU, MERRA2, CFSR, Global Precipitation Climatology Project monthly precipitation
(GPCP, Adler et al., 2003), Global Precipitation Climatology Centre monthly precipitation
(GPCC, Schneider et al., 2013), ERA-Interim reanalysis dataset (ERAI, Dee et al., 2011),
Japanese 55-year Reanalysis (JRA-55, Onogi et al., 2007), NOAA-CIRES 20th Century
Reanalysis version 2c (20CRv2c, Compo et al.. 2011), the CPC Merged Analysis of
Precipitation (CMAP; Xie and Arkin, 1996), and the Version 7 TRMM Multi-Satellite
Precipitation Analysis -3B42 research version (TRMM; Huffman et al., 2014). All the
datasets were regridded to the HadAM3P grid before biases were calculated.

For all the observational datasets, data from December 1996 to November 2007 (the same
time period the model simulations cover as shown in Table2) was used to calculate model
biases, except TRMM, which is only available starting from 1998.

2.5 Emulators
In Phase 1, a 2-layer feed-forward Artificial Neural Network (ANN, Knutti et al., 2003;
Sanderson et al., 2008; Mulholland et al., 2016) was used. Although other machine-
learning algorithms could be suitable (Rougier et al., 2009; Neelin et al., 2010; Bellprat et
al., 2011, 2012, 2016), we chose ANN because it permits multiple simultaneous emulator
targets (i.e., TOA SW and LW at the same time). Although ANN has the advantage of
using multiple metrics as targets simultaneously, the underlying emulator structure remains
obscure. From Phase 2, for the sake of simplicity and transparency,  we used kriging -
which is similar to a Gaussian process regression emulator -  following McNeall et al.
(2016) as coded in the package DiceKriging (Roustant et al., 2012) in the statistical
programming environment R. We used universal kriging, with no 'nugget' term, meaning
that the uncertainty on model outputs shrinks to zero at the parameter input points that have
already been simulated by the climate model (Roustant et al., 2012). Please refer to
Appendix A for further details.

**2.6 Sensitivity Analysis**
The response of the climate model to perturbations in the multidimensional parameter
space can be non-linear. In order to isolate the influence of each parameter on key climate
metrics and eliminate parameters that do not have a strong control on those metrics, we
performed two types of sensitivity analysis. One determines the sensitivity of a single
parameter by perturbing one parameter with all other parameters fixed, i.e. one-at-a-time
(OAAT) sensitivity analysis. Following Carslaw et al. (2013) and McNeall et al. (2016),
we also used a global sensitivity analysis using Fourier Amplitude sensitivity test (FAST)
for qualitative sensitivity analysis to validate the results of OAAT and to estimate
interactions among parameters. FAST allows the computation of the total contribution of
each input parameter to the output's variance, where total includes the factor's main effect,
as well as the interaction terms involving that input parameter. In the FAST method, the
fraction of the total variance due to the interactions is not resolved as the sum of individual
interactions, but is computed from the parameter contribution to the residual variance , i.e.,
variance not accounted for by the main effects. The computational aspects and advantages
of FAST are described in Satelli et al. (1999). Emulators are used for the sensitivity
analysis.

**3. Results and Discussion**
Top-of-atmosphere (TOA) radiative balance is an emergent property in GCMs (Irvine et
al., 2013), and the fact that the models of the IPCC Assessment Report 4 did not need flux-
adjustment was seen as an improvement over earlier models (Solomon et al., 2007).
Although climate models approximately balance the net absorption of solar radiation with
the outward emission of longwave radiation (OLR) at the TOA, the details of how solar
absorption and terrestrial emission are distributed in space and time depend on global
atmospheric and oceanic circulation, clouds, ice, and other aspects of model behaviour.
The surface expression of those global processes is also important given that a primary and
practical purpose of climate modelling is to understand how (surface) climate will change.
We describe the responses of both global TOA and regional surface climate to parameter
refinement.

**3.1. TOA radiative fluxes**
In Fig. 1, we show the TOA energy flux components from the PPEs from each of the three
phases. The ranges of acceptability for SW and LW (as denoted by the ellipse in Fig. 1)
were defined by taking the observational uncertainty ranges given in Stephens et al. (2012),
but tripling them (deliberately setting a lenient elimination criteria), and then expanding
both the negative and positive thresholds by an additional 1 W m-2 to account for internal
variability as estimated from SP (Fig. S5). Please refer to Appendix A for further details.
In Phase 1, many parameter sets (72%) resulted in TOA energy fluxes that vastly exceeded
our ranges of acceptability (as defined in Appendix A).  In Phase 2, most of the parameter
sets resulted in TOA energy fluxes that fell within the ranges of acceptability; the 20% that
did not reveal the error in our predictions using the emulator since the parameter sets were
chosen to specifically achieve TOA fluxes within the region of acceptability.  In Phase 3,
nearly all (97%) the parameter sets yielded acceptable results.  It is worth mentioning again
that in Phase 3, selection of parameter sets was based only secondarily on TOA fluxes and
primarily on regional climate metrics (see detailed description of Phase 3 in Appendix A).
Fig. B1 and B2 (in Appendix B) show predictions from emulators against model-simulated
values for model output metrics as validations of the emulators. The linear relationships
between the emulated and simulated results are very strong (regression coefficient
regcoef>0.9 for both LW and SW), while the emulated results can predict the simulated
results relative well, with coefficient of determination R2 > 0.9 for both LW and SW.
Please refer to Appendix B for further details on emulator validations.

Rowlands et al. (2012) discarded any ensemble member that required a global annual mean
flux adjustment of absolute magnitude greater than 5 W m-2 (see red lines in Fig. 1) and
Yamazki et al. (2013) defined a confidence region of (SW, LW) that corresponded to a
TOA imbalance of less than 5 W m$^{-2}$ as one that did 'not drift significantly' from a realistic
TOA state.  Although the ranges of acceptability (Fig.1) permits net TOA imbalance
greater than 5 W m$^{-2}$, more than half (55.8%) of the Phase 3 parameter sets generated a
TOA imbalance less than 5 W m$^{-2}$, and the smallest TOA imbalance was less than 0.1 W
m$^{-2}$.

The entrainment coefficient (ENTCOEF) and the ice fall speed (VF1) were the dominant
controls on the TOA outgoing SW and LW fluxes, respectively (see SW and LW response
to these two parameters shown in the bottom two rows of Fig. S2).  Why these parameters
are important becomes clear from understanding their respective roles in the climate model,
especially with respect to convection and hydrometeor transport.

The atmospheric model simulates a statistical ensemble of air plumes inside each
convectively unstable grid cell. On each model layer, a proportion of rising air is allowed
to mix with surrounding air and vice-versa, representing the process of turbulent
entrainment of air into convection and detrainment of air out of the convective plumes
(Gregory and Rowntree, 1990). The rate at which these processes occur in the model is
proportional to ENTCOEF, which is a parameter in the model convection component
(Table1). The implication of perturbing ENTCOEF has been investigated by (Sanderson et
al, 2008b) using single perturbation experiments, and they showed that a low ENTCOEF
leads to a drier middle troposphere and moister upper troposphere. Conversely, increasing
ENTCOEF results in increased low level moisture (more low level clouds) and decreased
high level moisture (less high level clouds). Because the albedo effects of low clouds
dominate their effects on emitted thermal radiation (Hartmann et al., 1992; Stephens,
2005), increasing ENTCOEF increases the outgoing SW fluxes.

VF1 is the speed at which ice particles may fall in clouds. A larger ice fall speed is
associated with larger particle sizes and increased precipitation. Wu (2002) studied ice fall
speed parameterization in radiative convective equilibrium models, and found that a
smaller ice fall speed leads to a warmer, moister atmosphere, more cloudiness, weak
convection and less precipitation, which could lead to decreased outgoing LW TOA flux
due to absorption in the cloud itself and/or in the moist air. Higher ice fall speeds produce
the opposite - a cooler, clearer, less cloudiness, strong convection and more precipitation,
which increases the outgoing LW flux.

**3.2. Regional climate improvements**
A primary and practical purpose of climate modelling is to understand how (surface)
climate will change, but model biases can have non-negligible impacts on projections. In
Phase 2 and 3 we evaluate the response of regional surface climate to parameter
perturbations, and refine the parameter space to reduce biases in regional temperature and
precipitation.

In Phase 2, we identified ENTCOEF and VF1 as distinct from the other 15 parameters with
respect to their influence on the overall suite of climate metrics to a first order
approximation (Fig. S3). Recall the regional surface metrics considered were MAC-T, JJA-
T, JJA-Pr, DJF-T, and DJF-Pr. Though MAC-T is our principal metric (section2.1), MAC-
T co-varies with JJA-T, JJA-Pr, and DJF-T (Fig. S3), so moving in parameter space toward
lower bias in MAC-T reduces biases in JJA-T, JJA-Pr, and DJF-T. MAC-T does not co-
vary strongly with DJF-Pr.

Each OAAT relationship in Fig. 2 depends on the initial ranges of the input parameters
from the ensemble design, and is computed while holding all other parameters at their
ensemble mean values. OAAT results while holding all other parameters at their SP values
are similar to those shown in Fig. 2 (results not shown here). Because sensitivity can change
as one moves through the parameter space (e.g. CW_LAND and ENTCOEF in Fig. 2),
these relationships must be interpreted with care. Within the refined parameter space in
Phase 2, ENTCOEF and the parameter that limits photosynthesis (and thereby latent heat
flux via transpiration) as a function of soil water (V_CRIT_ALPHA) were the most
influential individual parameters  and counter each other when both increased (Fig. 2 and
Fig. S3).   The parameter that controls the cloud droplet to rain threshold over land
(CW_LAND) also had strong influence on MAC-T across the lower end of the parameter
perturbation range (up to 0.004). The other parameters had little to effectively no influence
on MAC-T. The results of OAAT sensitivity analysis for the other output metrics
considered in Phase 2 are presented in Fig. S6-S11.

The global sensitivities of the simulated outputs (the ones considered in Phase 2) due to
each input, as both a main effect and total effect, including interaction terms, are presented
in Fig. 3. ENTCOEF was the most important parameter for all three surface temperature
metrics, with a total sensitivity index of ~0.7, 0.5, and 0.4 for MAC-T, JJA-T, and DJF-T
respectively , where maximum sensitivity is 1 (see Satelli et al. 1999). For the metrics
MAC-T and JJA-T, V_CRIT_ALPHA was the next most important, with a total sensitivity
index of ~0.3 for both metrics. For JJA-Pr, the most important parameter was VF1,
followed by ENTCOEF; for DJF-Pr, the most important parameter was ENTCOEF, closely
followed by the parameter that controls the roughness length for free heat and moisture
transport over the sea (Z0FSEA).

The interaction terms were relatively small, accounting for a few percent of the variance,
except for the effect of ENTCOEF on DJF-Pr, where the interaction with other parameters
accounts for ~ 1/3 of the variance. In a study constraining carbon cycle parameters by
comparing emulator output with forest observations, McNeall et al. (2016) also found the
importance of the interaction terms negligible. In contrast, Bellprat et al. (2012b) used
quadratic emulator to objectively calibrate a regional climate model, and found non-
negligible interaction terms. They showed that excluding the interactions in the emulator
increased the error of the emulated temperature and precipitation results by almost 20%.
Further work could be done to assess the magnitude and functional form (i.e. linear or
nonlinear) of the interaction terms, but is beyond the scope this study.

Only the parameters with a total sensitivity index larger than ~0.1 for MAC-T, JJA-T, DJF-
T, JJA-Pr, or DJF-Pr were retained for perturbation in Phase 3: CW_LAND, VF1,
ENTCOEFF, V_CRIT_ALPHA, ASYM_LAMBDA, G0, and Z0FSEA. Although the
parameter that controls the rate at which cloud liquid water is converted to precipitation
(CT) had a total sensitivity index of ~0.1 for SW, it was excluded from further perturbation
because the primary interest in Phase 2 was in regional surface metrics, not TOA radiative
fluxes.

Phase 3 demonstrated the power of our approach for reducing regional mean biases in
MAC-T, JJA-T and JJA-Pr. Simulations from Phase 3 resulted in MAC-T biases 1- 3°C
lower than SP (Fig.4 middle row). All Phase 3 parameter sets improved the JJA-Pr dry bias
with several eliminating the bias entirely. Many parameter sets reduced the bias in JJA-T
to less than 1.5°C, a dramatic improvement (~63%) over the 4°C SP bias. However, these
improvements come at a small price, namely a larger regional (NWUS) dry bias in DJF-Pr
(about -15% compared with PRISM in the worst case). Because our primary goal was to
reduce JJA warm and dry biases, any model variant from Phase 3 is preferable to SP. Any
subset of parameterizations from phase 3 can now be used in subsequent experiments.

V_CRIT_ALPHA plays an important role in controlling JJA-T and MAC-T (as shown in
Fig. 2 and Fig. S6) due to its role in the surface hydrological budget. V_CRIT_ALPHA
defines the critical point as a fraction of the difference between the wilting soil water
content and the saturated soil water content (as described in Appendix C).  The critical
point is the soil moisture content below which plant photosynthesis becomes limited by
soil water availability. When V_CRIT_ALPHA is zero, transpiration starts to be limited as
soon as the soil is not completely saturated, whereas when it is one, transpiration continues
unlimited until soil moisture reaches wilting point at which point transpiration switches
off. Lower values of V_CRIT_ALPHA reduce the critical point allowing plant
photosynthesis to continue unabated at lower soil moisture levels, i.e. plants are not water-
limited. As plants photosynthesize water is extracted from soil layers and transpired,
increasing the local atmospheric humidity and lowering the local temperature through
latent cooling. Our results are consistent with previous findings by Seneviratne et al.
(2006), who also show reducing the temperature and increasing humidity can feedback
onto the regional temperature and precipitation during the summer months.

The only apparent constraints on ranges of parameter values through three phases of
parameter refinement were seen for V_CRIT_ALPHA and ENTCOEF. Values of
V_CRIT_ALPHA  lower than 0.7 were required to keep the bias of MAC-T under 3 °C.
For ENTCOEF, the range between 3 and 5 contains the best candidates to reduce regional
warm/dry biases. The range of ENTCOEF identified here is consistent with findings of
Irvine et al. (2013), which also show that low values of ENTCOEF tend to give warmer
conditions. However, results from other previous studies varies. Williamson et al. (2015)
found that low values of ENTCOEF are implausible, and that there are more plausible
model variants at the upper end of its perturbed range, whereas Sexton et al. (2012a) and
Rowlands et al. (2012) consider the range between 2 and 4 to contain the best model
variants. The discrepancy in optimal ranges for ENTCOEF are to be expected given that
the primary metrics used to evaluate the effect of parameter refinement are different, with
ours being JJA warm/dry biases over the NWUS, William et al. (2015) being the behaviour
of Antarctic Circumpolar Current, and other previous studies being climate sensitivities.
This demonstrates that any parameter refinement process is tailored to a specific objective,
and choices regarding metrics (e.g., variables, validation dataset(s), and / or cost functions)
may determine which part of parameter space is ultimately accepted.

**3.3. Effects on global scale climate**

To avoid introducing or increasing biases over other parts of the globe by our regionally-focused model improvement effort,  we investigated the large-scale effects of the selected 10 'good' (least biased in MAC-T) sets of global parameter values. We focused on surface temperature and precipitation because they are key variables of the climate system and are of high interest for impact studies.

Figure 5 shows the meridional distribution of Northern Hemisphere (NH) mid-latitude temperature (over land) and precipitation in DJF and JJA. Because of the wide range of parameter values in the PPEs of Phase 1 and Phase 2, the spread for these PPEs is quite large, whereas the ensemble spread in Phase 3 is substantially smaller. Compared with the SP ensemble, the new parameter values (final 10 sets) reduced the zonal mean JJA temperature throughout the NH mid-latitudes (30 °N -60 °N), by ~1 °C – 4 °C (depending on the particular combination of parameters), and increased JJA precipitation over the same latitude bands, except for latitudes south of 33 °N and north of 58 °N. In DJF, the effects are not as large nor are the changes consistent in sign across the NH mid-latitude region (though south of ~38 °N all 10 parameter sets give increasing precipitation). The SP simulations have warm and dry biases over NWUS and mid-latitude land in general (as shown in Fig. 4, Fig. 6 and Fig. 7). In JJA all the selected PP model variants show considerably different results compared with the SP-cooler and wetter, i.e. reduced biases and improved model performance. Figure 5 also demonstrates that varying model

parameters has a bigger influence than varying initial conditions, as seen from the wider
spread of PP results compared with the spread of SP initial condition perturbation results.

To examine how parameter refinements affect spatial patterns of biases, we compare the
seasonal mean biases of temperature (Fig. 6) and precipitation (Fig. 7) under SP and the
selected PP settings, against CRU data. The SP simulations have large warm biases in JJA
(and to a lesser extent in MAM and SON, Fig. 6 b-d) over the NH mid-latitude land region,
that are substantially lower in the PP simulations (Fig. 6 f-h and Fig.6 j-l).  In the tropics,
the SP simulations have cold biases over northern South America, central Africa and
southern Asia in most seasons that are ameliorated in the PP simulations in some cases
(e.g. central Africa in DJF and SON) - even though the focus of the PP simulations was
improving the climate of the NWUS. The SP simulations also have cold biases over most
of the Southern Hemisphere continents in mid-latitudes in most seasons. A large fraction
of the JJA temperature biases were reduced in the PP simulations, as shown in Fig. 6c, g
and k. These salient features in JJA temperature biases under SP and PP are not particular
to the selection of observational dataset (see Fig. S12-S13 for comparison with other
datasets). In the other three seasons, however, the spatial patterns of temperature biases are
not consistent across observational datasets.

The reduction of JJA temperature from SP to PP (Fig. 6k) and the resulting reduction in
bias are accompanied by reduction in precipitation in the equatorial regions; increased
precipitation over northern North America, northern Africa, and Europe (Fig. 7k); and
decreased incoming shortwave radiation at the surface and increased evaporation (Fig.
S14). Stronger evaporative cooling and reduced surface radiation lead to a cooling of the
JJA climate, which roughly agrees with the geographical pattern of reduced mean JJA
temperature, consistent with findings in Zhang et al. (2018) that both overestimated surface
shortwave radiation and underestimated evaporation contribute to the warm biases in JJA
in CMIP5 climate models.

For precipitation, the largest biases in SP are over Amazonia in DJF and MAM (Fig. 7a
and b), and northern South America, equatorial Africa, and south Asia in JJA (Fig. 7c).
These summer biases are increased in the PP simulations (Fig. 7k). However, it is difficult
to know whether we are improving the model's global precipitation patterns because of the
large uncertainty in historical precipitation observational datasets. Still, it is worth
comparing the PP simulations with both a variety of observational-based datasets and other
GCMs (Fig. 8). The precipitation amounts differ substantially across different
observational datasets, as well as across climate models. In the tropics, Phase 3 PP
simulated precipitation is mostly lower (except DJF just north of the equator) and has
narrower range than the observations or other climate models, but is higher in DJF and JJA
(up to 25% higher) than the SP simulation results. Outside the tropics, the precipitation
distributions in PP remain similar to those of SP, and differences from observational
datasets and other GCMs are less affected by the use of PP. The tropical precipitation
improvements in JJA can be taken as a general improvement, though not with high
confidence due to the variability across observational datasets. To further highlight the
uncertainties in precipitation, global maps of differences in biases between SP and our
selected parameter settings, in comparison with other observational-based datasets, are
presented in Fig. S15-22.

The fact that the large JJA warm bias (shared with many other GCMs and RCMs; see e.g.
Mearns et al., 2012; Kotlarski et al., 2014) could be reduced substantially through the use
of PP is a notable result, especially since the bias persisted through initial tuning efforts
and through the recent updates from version 1 to version 2 of weather@home. We
demonstrated here that significant improvements in the simulation of JJA temperature can
be made through parameter refinements, and that these JJA temperature biases are not
necessarily structural issues of the climate model. These improvements in simulating JJA
temperature generally did not overall improve JJA precipitation patterns across the globe,
and even worsened the bias in some places (e.g. South America).

**4. Conclusions**
Through an iterative parameter refinement approach to improve model performance, we
identified a region of climate model parameter space in which HadAM3P outperforms the
SP variant in simulating summer climate over the NWUS specifically, and over NH mid-
latitude land in general, while approximately maintaining TOA radiative (near-) balance.
Improving the northwest US climate comes with tradeoffs, e.g. larger JJA dry bias over
Amazonia. However, it is important to note that there are large uncertainties in observed
precipitation climatology, especially outside of the North American and European mid-
latitudes, so both apparent increases and decreases in biases should be treated with caution,
and compared against the range across observational datasets.  In the end, we consider the
cost of increasing biases in parts of the globe acceptable for the purposes of selecting
multiple global model variants to drive the regional model with reduced JJA biases over
NWUS. The fact that improvements can be made at all (for a substantial area of the world)
through targeted PPE is encouraging.

Our parameter refinement yielded important improvements in the representation of the
summer climate over the NWUS, and it follows that biases in other models may also be
reduced by refining certain parameters that, although may not be identical to those in
HadAM3/RM3P, influence the same physical processes similarly. We found ENTCOEF
and V_CRIT_ALPHA to be the dominant parameters in reducing JJA biases. These
parameters control cloud formation and latent heat flux, respectively. Bellprat et al. (2016)
found the key parameter responsible for reduction of JJA biases is increased hydraulic
conductivity, which increases the water availability at the land surface and leads to
increased evaporative cooling, stronger low cloud formation, and associated reduced
incoming shortwave radiation. We only perturbed one land surface parameter, but the
effects of additional land surface parameters are being explored in a subsequent study.
Given that land model parameters such as V_CRIT_ALPHA could reasonably be expected
to interact with sensitive atmospheric parameters like ENTCOEF, it is particularly
interesting to consider the multivariate sensitivity of a range of parameters that span across
component models (e.g., land, ice, atmosphere, ocean). We argue that this frontier of
parameter sensitivity exploration should be done in a transparent and systematic manner,
and we have demonstrated that statistical emulators can be effectively leveraged to reduce
computational expense.

The fact that V_CRIT_ALPHA (which is a parameter in the land surface scheme MOSES2)
was found to be an important parameter on regional MAT-C and JJA-T, has much further
implications beyond this study. MOSES2 is the land surface scheme used in HadGEM1
and HadGEM2 family, which were used in CMIP4 and CMIP5. Moreover, the Joint UK
Land Environment Simulator (JULES) model (which is the land surface scheme of the
CMIP6 generation Hadley Centre models HadGEM3 family, https://www.wcrp-
climate.org/wgcm-cmip/wgcm-cmip6) is a development of MOSES2. What we have
learned about the atmosphere-land surface interactions here is relevant to even the most
recent HadGEM model generation and the in-progress CMIP6.

The reduction of JJA biases that we achieved in our multi-phase parameter refinement is
notable. However, despite out efforts, the 'best' performing parameter set still simulates a
MAC-T bias of 1.5 °C, and a JJA-T bias of 1 °C, over the NWUS. Future work could be
done to determine whether the model can be further improved by tuning additional land-
surface scheme parameters, and/or to what extent the remaining biases are due to structural
errors of the model for which we cannot (nor even should not) compensate by refining
parameter values. However, with the reduction in JJA temperature bias, future projections
using the new parameter settings over the SP should be at less risk of overestimating
projected warming in summer (as discussed in the introduction).

It is also worth noting that we restricted our analysis to seasonal and annual mean climate
metrics. Given the use of weather@home for attribution studies of many extreme weather
events (e.g., Otto et al., 2012; Rupp et al., 2017a) as well as their impacts, such as flooding-
related property damages (Schaller et al., 2016) and heat-related mortality (Mitchell et al.,
2016), an important next step would be to investigate how the tails of distributions of
weather variables respond to parameter perturbations. Furthermore, looking at biases in
seasonal mean temperature and precipitation is insufficient to fully assess model
performance. As a follow-up step to this study, we recommend a process-based model
evaluation and physical explanation of model improvements to further refine the parameter
space that provides improvements (e.g., reduce summer biases) through appropriate
physical mechanisms. For example, more accurate representation of clouds in the model
could lead to better simulated downward solar radiation at the surface, as well as better
simulated surface energy and water balance.

Another important next step would be to apply the selected PPE over the weather@home
- European domain, given the non-trivial JJA warm bias identified over Europe by previous
studies (Massey et al., 2014; Sippel et al., 2016; Guillod et al., 2017). Bellprat et al. (2016)
showed that regional parameters tuned over Europe domain also produced similar
promising results over North America domain but the same model parameterization yielded
larger overall biases over North America than for Europe. One could test the transferability
of parameter values over different regional domains in the weather@home framework,
given weather@home currently uses the same GCM to drive several RCMs over different
parts of the world, all using the  same parameter values.

The methodology presented in this study could be applied to other models in the evolution
of physical parameterizations, and we advocate that parameter refinement process should
be more explicit and transparent as done here. Choices and compromises made during the
refinement process may significantly affect model results and influence evaluations against
observed climate, hence should be taken into account in any interpretation of model results,
especially in intercomparison of multimodel analyses to help understanding of model
differences.

**Code availability**
HadRM3P is available from the UK Met Office as part of the Providing REgional
Climates for Impacts Studies (PRECIS) program. Access to the source code is dependent
on attendance at a PRECIS training workshop
(http://www.metoffice.gov.uk/research/applied/international-development/precis/obtain).
The code to embed the Met Office models within weather@home is proprietary and not
within the scope of this publication.

**Data availability**
The model output data for the experiment used in this study will be freely available at the
Centre for Environmental Data Analysis (http://www.ceda.ac.uk) in the next few months.
Until the point of publication within the CEDA archive, please contact the corresponding
author to access the relevant data.

**Appendix A: Detailed experimental process**
The overarching goal is to refine parameter values to reduce warm and dry summer bias in
the NWUS. In total four ensembles were generated, one using the SP values and one for
each of 3 PPE phases.  Details of each ensemble are listed in Table 2.

Internal variability of the atmospheric circulation can confound the relationship between
parameters values and the response being sought (i.e. result in a low signal-to-noise ratio).
Averaging over multiple ensemble members with the same parameter values but different
atmospheric initial conditions (ICs) can clarify the true sensitivity to parameters by
increasing the signal-to-noise ratio. We set up multiple ICs for each parameter set, but the
numbers of ICs applied was not consistent throughout the experiment. The IC applied in
each phase was determined somewhat subjectively, trying to strike a balance between
running a large enough PPE to probe as many processes and interactions between
parameters as possible, having multiple ICs so that the results were representative of the
parameter perturbations instead of reflecting the influence of any particular IC, while under
the practical limitation of data transfer, storage, and analysis. The actual IC ensemble size
used in the final analysis was also constrained by the number of successfully completed
returns from the distributed computing network.

The four ensembles are summarized below:
**SP:** A preliminary "standard physics" (SP) ensemble with 10 ICs that used only the default
model parameters was generated to provide a benchmark to access the effects of parameter
perturbations.

**Phase 1:** The objective of this phase was to eliminate regions of parameter space that led
to top-of-atmosphere (TOA) radiative fluxes that are strongly out of balance. Exclusion
criteria were deliberately lenient, to avoid eliminating regions of the parameter space that
could potentially reproduce the observed temperature and precipitation over the western
US. We perturbed 17 parameters simultaneously, using space-filling Latin hypercube
sampling (McKay et el., 1979) - maximizing the minimum distance between points - to
generate 340 sets of parameterizations across the range of parameter values described in
Table 1. To generate enough ensemble members for a statistical emulator, Loeppky et al.
(2009) suggested that the number of sets of parameter values be 10 times the number of
parameters ($p$). We used more than $10p$ sets of parameter values in this, and subsequent
phases of PPE. A total of 2040 simulations (340 sets of parameter values x 6 ICs) were
submitted to the volunteer computing network. This phase was considered finalized when
simulations with 220 sets of parameter values and 3 IC ensemble members per set were
returned from the computing network.

Model results were used to train a statistical emulator which maps the relationship between
parameter values and key climate metrics. In this phase, the metrics were outgoing LW and
(reflected) SW TOA radiative fluxes. We considered these two metrics separately because
the total net radiation could mask deficiencies in both types of radiation through
cancellation of errors.

For the emulator, a 2-layer feed-forward Artificial Neural Network (ANN, Knutti et al.,
2003; Sanderson et al., 2008; Mulholland et al., 2016) was used. Although other machine-
learning algorithms could be suitable (Rougier et al., 2009; Neelin et al., 2010; Bellprat et
al., 2012a,b, 2016), we chose ANN because it permits multiple simultaneous emulator
targets (i.e., TOA SW and LW at the same time). We used an ellipse (Fig. 1) to define the
space of acceptability for SW and LW, starting with the observational uncertainty ranges
given in Stephens et al. (2012), but tripling them (deliberately setting a lenient elimination
criteria), and then expanding both the negative and positive thresholds by an additional 1
W m$^{-2}$ to account for internal variability as estimated from SP (Fig. S5).  Sets of parameter
values that fall within our range of acceptability were retained, and the ranges of these
refined/restricted parameter values defined the remaining  parameter space.

A new set of 1,000 parameter configurations was generated from the remaining parameter
space using space-filling Latin hypercube sampling. With this new ensemble we increased
the sample density within the refined parameter space. The statistical emulator was used to
predict SW and LW for each of these 1,000 new sets of parameters, and 41% fell within
our range of acceptability, reflecting the deficiency of the emulator to some extent.
Parameter sets that fell within the acceptable range were used in Phase 2.

**Phase 2:** The objective of this phase was to reduce biases in the simulated climate of the
NWUS, where the warm summer biases were the most obvious (Fig. S1), while not straying
far from TOA radiative (near-) balance. The climate metrics considered were the mean
magnitude of the annual cycle of temperature (MAC-T), and mean temperature (T) and
precipitation (Pr) in December-January-February (DJF) and June-July-August (JJA).
Although a primary motivation for this study was to investigate and reduce the warm and
dry bias in JJA over NWUS, MAC-T was treated as the primary metric in Phase 2 because
it is a comprehensive measure of climate feedbacks in response to a large change in forcing,
e.g., solar SW (Hall and Qu 2006).  MAC-T is also strongly correlated to the other regional
metrics (particularly JJA-T) as evident in Fig. S3 – MAC-T against other metrics. We chose
a NWUS average MAC-T of +/-3 °C as the bias threshold over which parameter space
would be eliminated.  Though this threshold is arbitrary, falling below it would mean
reducing the MAC-T bias for the NWUS by about 50%.

We did not treat all metrics as equally important.  The order of importance in this second
phase was MAC-T > JJA-T, JJA-Pr, DJF-T, and DJF-Pr > SW and LW.

The 410 sets of new PPE from Phase 1 became the starting point for Phase 2.  A total of
27,060 simulations (410 sets of parameter values x 6 ICs x 11 years) was submitted to the
computing network. This phase was considered finalized when simulations with 170 sets
of parameter values and 3 IC ensemble members per set and per year were completed.
These 5,610 simulations were used to train a suite of statistical emulators for various
climate metrics.  An additional 94 sets of parameters with 3 IC ensemble members per set
and per year completed after starting Phase 3 and were used to validate the emulators
trained within Phase 2 (see Appendix B).

Separate statistical emulators were trained for MAC-T, JJA-T, JJA-Pr, DJF-T, DJF-Pr, SW,
and LW. Although ANN has the advantage of using multiple metrics as targets
simultaneously, the underlying emulator structure remains obscure, because an ANN is a
network of simple elements called neutrons which are organized in multilayer, and
different layers may perform different kinds of transformations on the inputs. For the sake
of simplicity and transparency, in Phase 2 we used kriging instead - which is similar to a
Gaussian process regression emulator - following McNeall et al. (2016) as coded in the
package DiceKriging (Roustant et al., 2012) in the statistical programming environment R.
We used universal kriging, with no 'nugget' term, meaning that the uncertainty on model
outputs shrinks to zero at the parameter input points that have already been run through our
climate model (Roustant et al., 2012). To validate if the emulators were adequate to predict
outputs at unseen parameter inputs, we needed to assure that it predicted relatively well
across our designed parameter inputs. For each emulator, we performed 'leave-one-out'
cross validation. The cross validation results showed no significant deviations in prediction
of the outputs (results not shown).

In addition to reducing parameter space in Phase 2, we also looked for parameters that
consistently showed little influence on our metrics of interest, as any reduction in
parameters could benefit subsequent experiments by reducing the overall dimensionality.
To identify which parameters have the most influence over the metrics of interest, we
performed two types of sensitivity analyses as described in Section 2.5. In the end, the 7
most influential parameters were retained after parameter reduction in Phase 2; these are
the bold-faced parameters in Table 1.

After eliminating parameter space resulting in MAC-T biases larger than 3°C, and reducing
the number of perturbed parameters to 7, we continued the parameter refinement process,
and randomly selected 100 parameter sets that emulated MAC-T biases less than 3°C and
had large spread in ENTCOEF and VIF1 (within the refined ranges of Phase 2). 100 was
subjectively chosen as a cut off number of new PPE sets to run through weather@home in
the next phase, mainly due to concern of not knowing how many more phases would be
required to reach our goal, while recognizing the practical constraints posed by the large
datasets that would potentially be generated in the following phases.

**Phase 3:**  This objective of this phase was to further refine parameter space to reach the
target of northwest US regional bias in MAC-T less than 3°C, and then select 10 sets of
parameter values that met this criterion. The results in this phase satisfied our target, so we
stopped the iterative process here.

We were aware that our approach of regionally targeted parameter refinements might
degrade model performance elsewhere. Upon achieving our regional target, we
investigated the effects of our model tuning on global model metrics.

**Appendix B: Emulated vs. simulated results**
We used 94 additional ensemble members returned from Phase 2 (the 94 simulations that
completed after building the emulators from the Phase 2 PPE and starting Phase 3) to
provide out-of-sample validations of the emulators trained in Phase 2.  In Fig. B1, we show
predictions from emulators against model-simulated values for all the output metrics. In all
cases, the linear relationship between the emulated and simulated is very strong (regression
coefficient regcoef>0.9), while the emulated results can predict the simulated results
relative well, with coefficient of determination R2 > 0.9 in the best cases (SW, LW and
JJA-T). It is not surprising that R2 for DJF-Pr is the smallest, considering precipitation in
DJF over NWUS is dominated by larger-scale atmospheric features such as the polar jet
stream, the Pacific subtropical high, and storm tracks (e.g.,Mock, 1996; Neelin et al., 2013;
Seager et al., 2014; Langenbrunner et al., 2015), and the internal variability of this metric
is the highest among those considered.

In Fig. B2, we present the emulated vs. simulated results in Phase 3 for the 95 PP sets that
were returned in Phase3. These 95 PP sets were run through the emulators from Phase 2 to
predict the climate metrics, then the emulated results were compared with the simulated
results returned from weather@home simulations. In most cases, r and R2 are lower than
the Phase 2 results (Fig. B1), except for LW and DJF-T, where R2 increases by a few
percent. This decrease in emulator prediction accuracy could be due to the fact that in Phase
3, only 7 parameters were perturbed simultaneously while keeping the rest at their default
values, so we have eliminated parts of the parameter space, which are no longer available
to the emulators.

The comparisons between simulated and emulated results from Phase 2 to Phase 3 highlight
the necessity of doing parameter refinement exercise in phases. Training a statistical
emulator once, then using it to search for optimal parameter settings may not always yield
optimum results. An emulator may not fully capture the behaviour of the climate model in
every aspect, especially when the number of parameters perturbed was changed during the
process, such as in our case.

## Appendix C: Soil moisture control on plant photosynthesis in MOSES

The critical point $\theta_{crit}$ ($m^3$ of water per $m^3$ of soil) is the soil moisture content below which plant photosynthesis becomes limited by soil water availability and is calculated by:

$$\theta_{crit} = \theta_{wilt} + V\_CRIT\_ALPHA\ (\theta_{sat}-\theta_{wilt})$$

where $\theta_{sat}$ is the saturation point, i.e. the soil moisture content at the point of saturation; and $\theta_{wilt}$ is the wilting point, below which leaf stomata close. V_CRIT_ALPHA varies between zero and one, meaning that $\theta_{crit}$ varies between $\theta_{wilt}$ and $\theta_{sat}$ (Cox et al., 1999).

## Author contributions

The model simulations were designed by S. Li ,D. E. Rupp, L. Hawkins, with inputs from P. W. Mote, and D. McNeall. All the results were analysed and plotted by S. Li. The paper was written by S. Li, with edits from all co-authors.

## Competing interests

The authors declare that they have no conflict of interest.

## Acknowledgements

This work was supported by USDA-NIFA grant 2013-67003-20652. We would like to thank our colleagues at the Oxford eResearch Centre for their technical expertise. We would also like to thank the Met Office Hadley Centre PRECIS team for their technical and scientific support for the development and application of weather@home. Finally, we

would like to thank all of the volunteers who have donated their computing time to
climateprediction.net and weather@home.

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

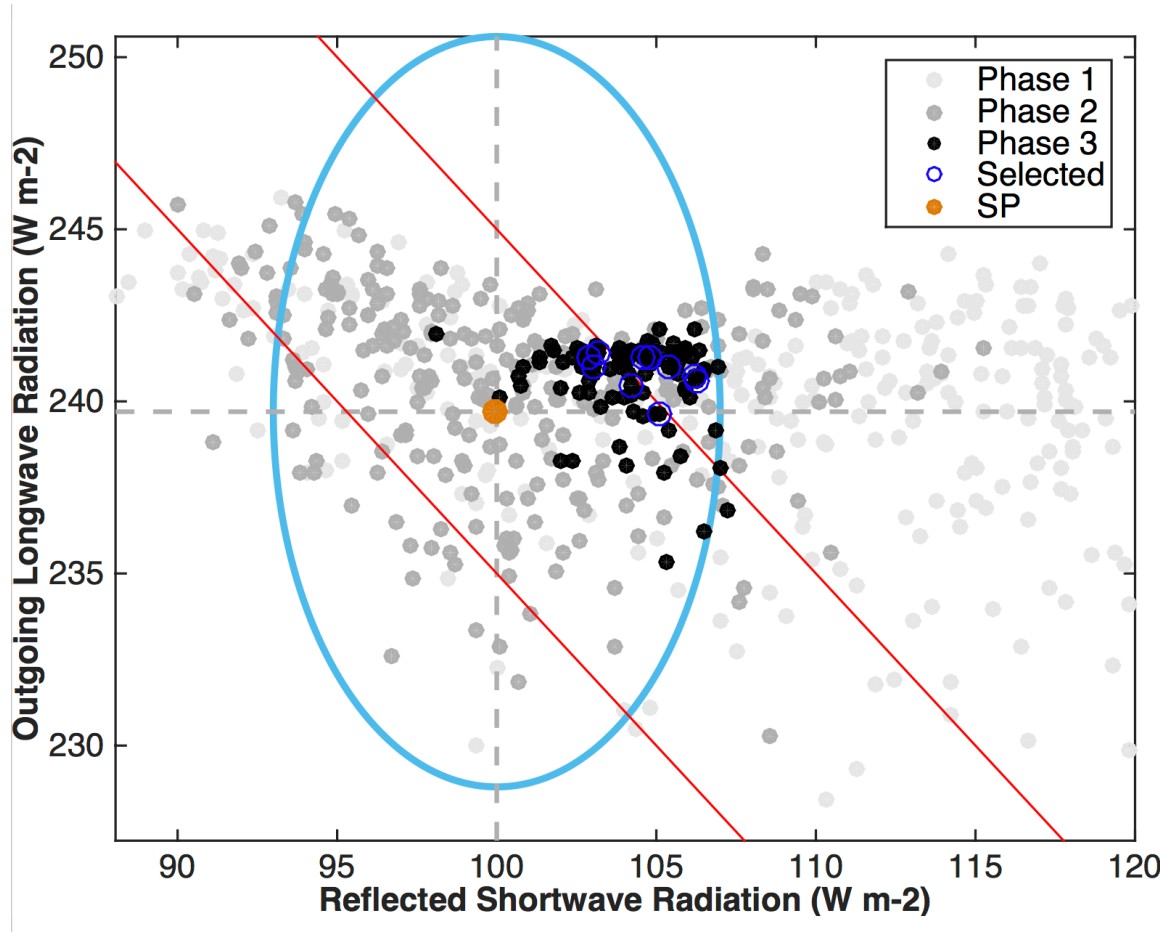


**Figure 1.** Global mean top-of-atmosphere (TOA) outgoing (reflected) shortwave radiation
(SW) and outgoing longwave radiation (LW) from the four ensembles run through
weather@home2. Horizontal and vertical dashed lines denote the reference values for SW
and LW taken from Stephens et al. (2012). The filled brown circle denotes our SP. The
ellipse indicates the uncertainty ranges we are willing to accept for SW and LW
respectively, which includes the observational uncertainty range taken from Stephens et al.
(2012), but tripled, plus the uncertainty range due to initial condition perturbations
estimated from our SP reference ensemble. The red solid lines highlight net TOA energy
flux of +/- 5 Wm$^{-2}$.

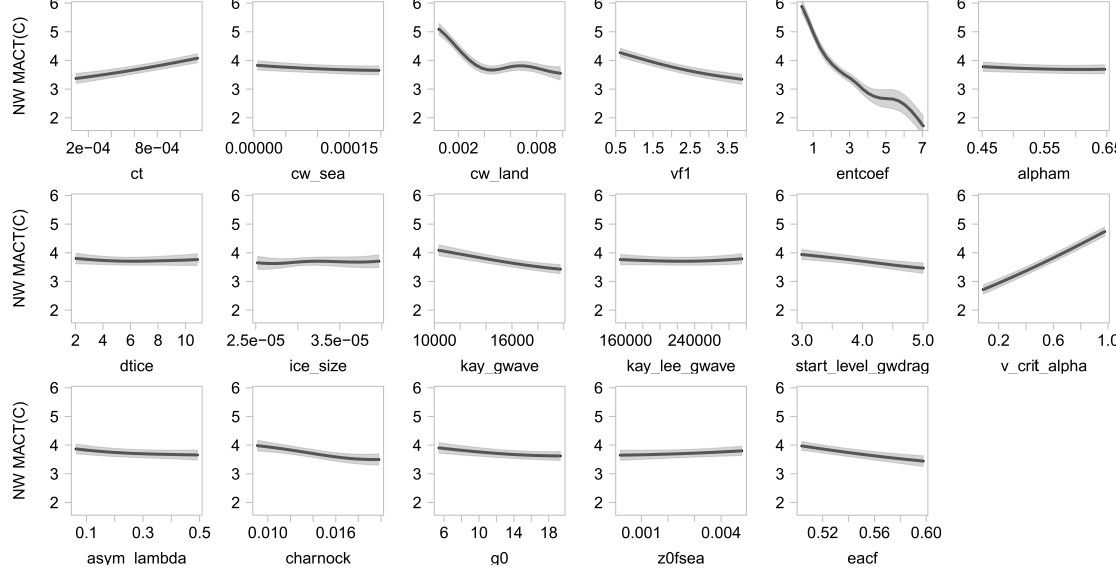

**Figure 2.** One-at-a-time sensitivity analysis of magnitude of annual cycle of temperature
(MAC-T) over Northwest to each input parameter in turn, with all other parameters held at
mean value of all the designed points. Heavy lines represent the emulator mean, and shaded
areas represent the estimate of emulator uncertainty, at the ±1 SD level.











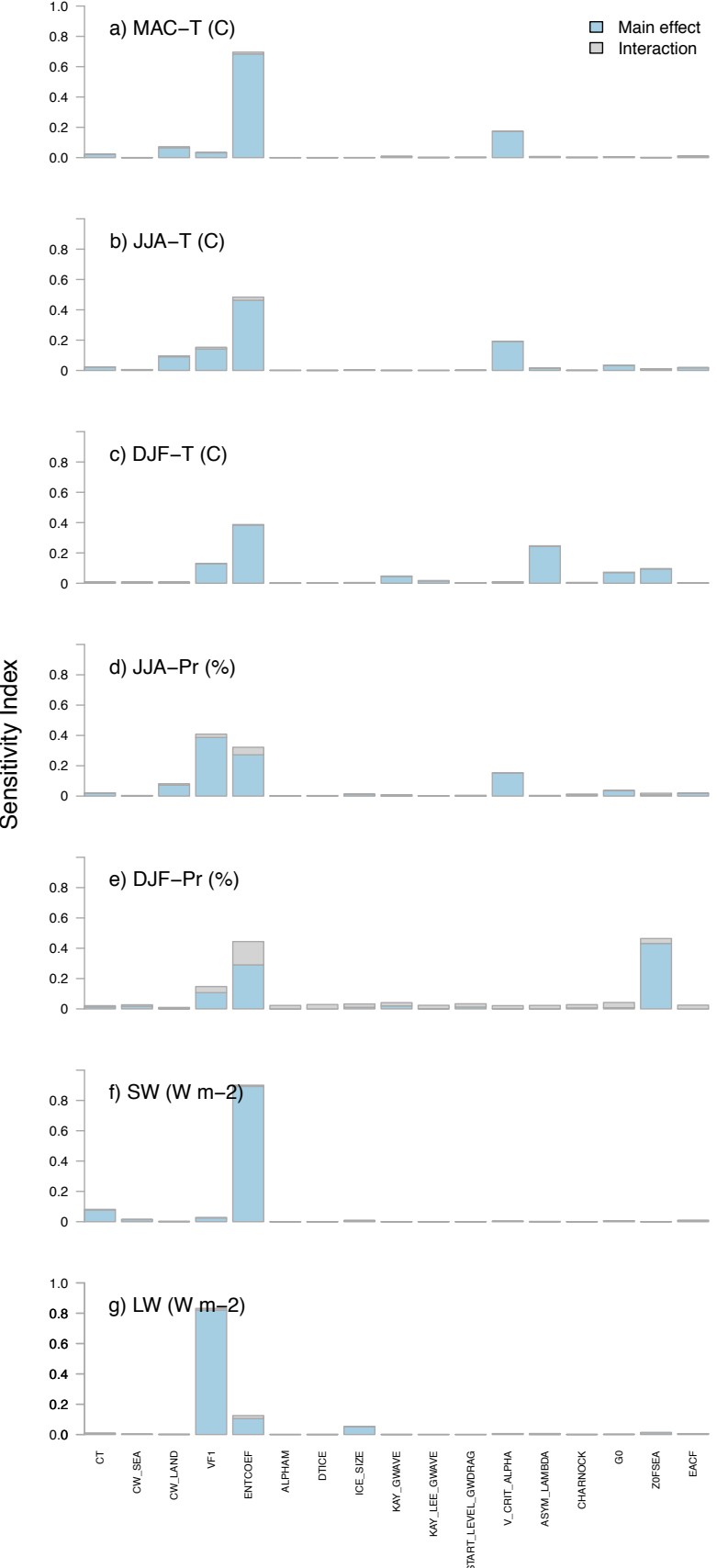


**Figure 3.** Sensitivity analysis of model output metrics in Phase 2 via the FAST algorithm
of Saltelli et el. (1999).

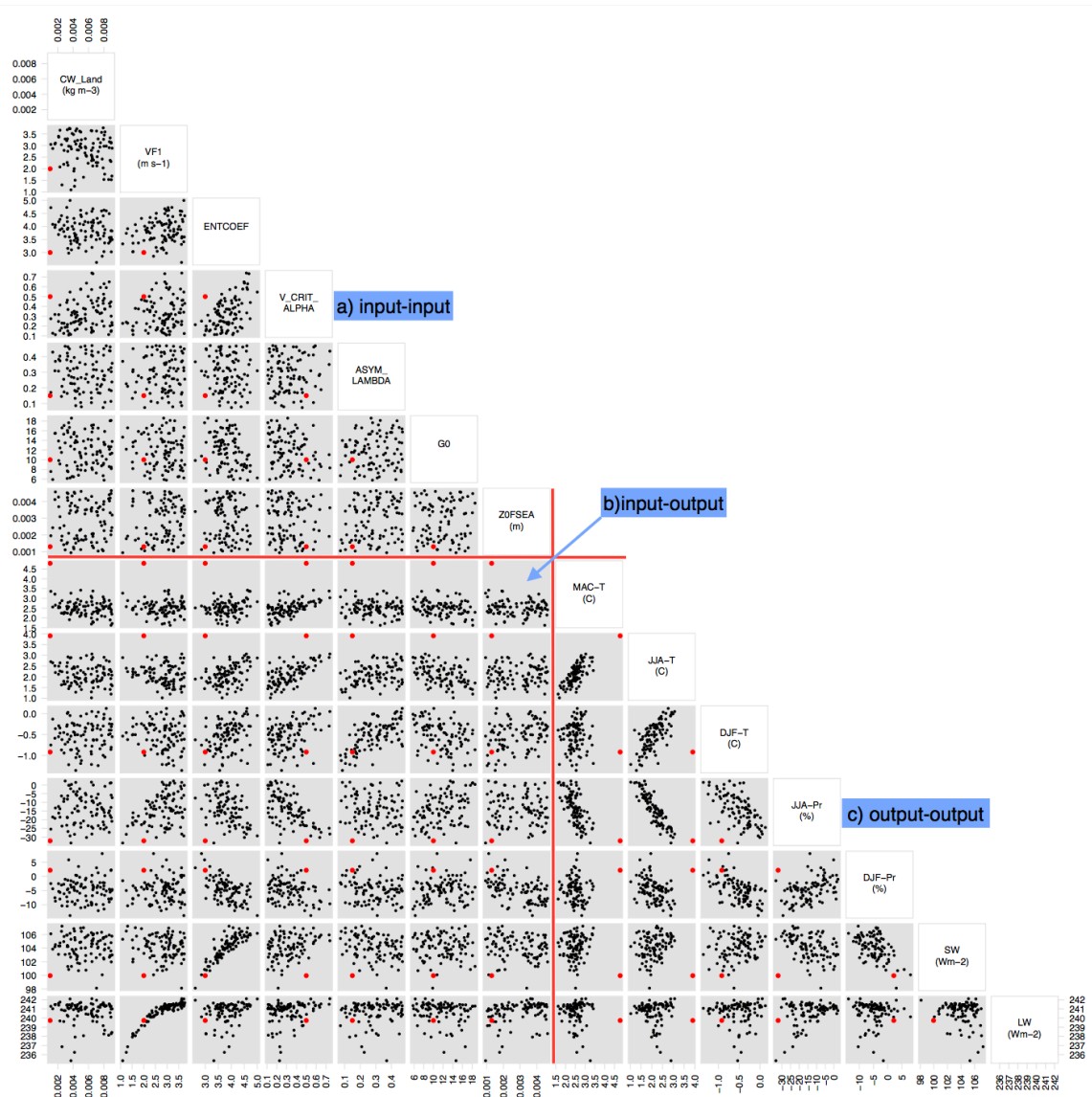


**Figure 4.** Phase 3 PPE parameter inputs and summary model output metrics evaluated. 95
parameter sets are shown. The parameter values and model outputs under SP are marked
in red. The horizontal and vertical red lines mark the transition from parameter inputs and
model output metrics.

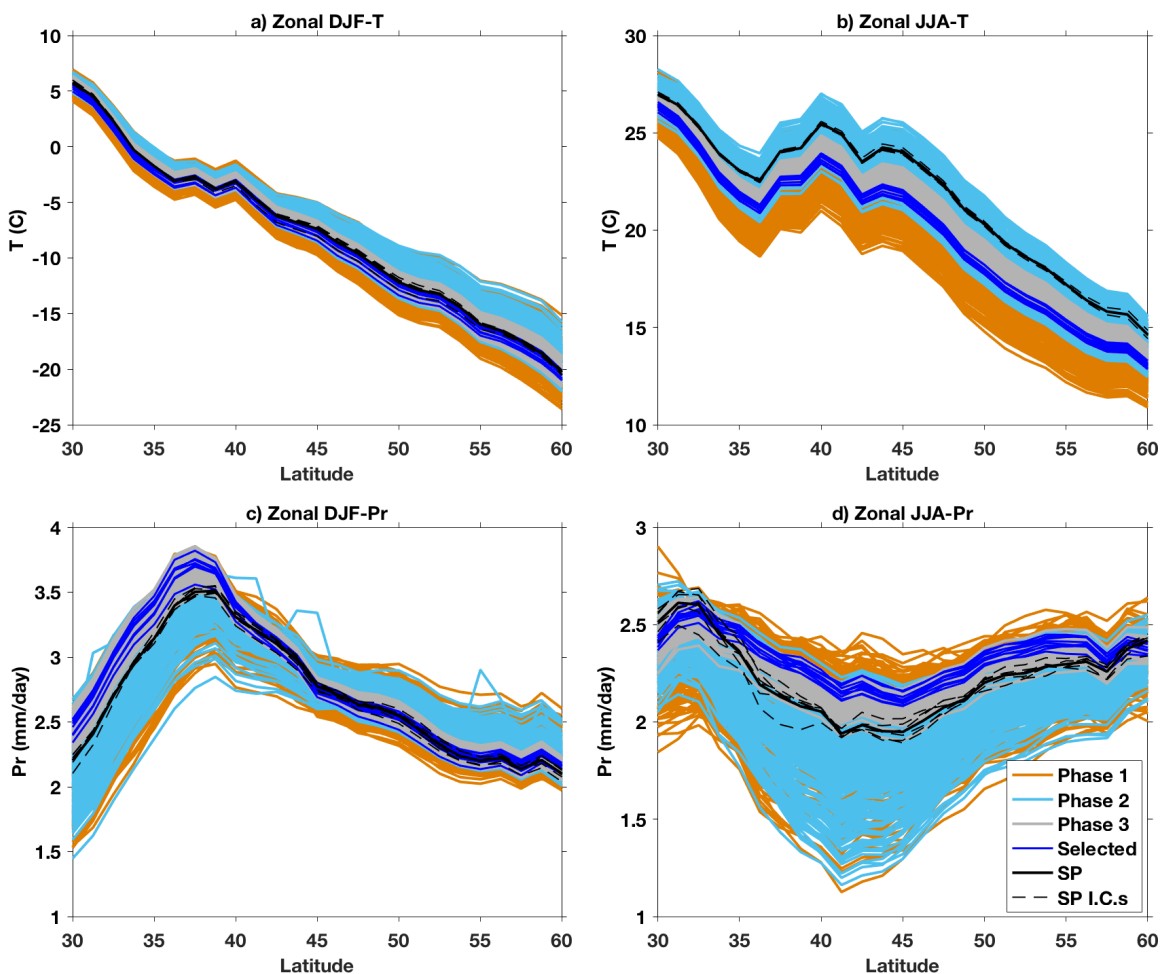


**Figure 5.** Comparison between three PPEs and SP zonal mean HadAM3P simulated North

Hemisphere mid-latitude (30°N-60°N) a) DJF mean temperature over land, b) JJA mean

temperature over land, c) DJF mean precipitation, and d) JJA mean precipitation. Output

from the selected 10 parameter sets selected, based on NWUS MAC-T, are shown in blue.

Note that the plotting order is the same as the legend, so most Phase 1 curves are obscured

by subsequent phases. The results from different initial conditions (I.C.s) under SP are

shown as black dashed lines.





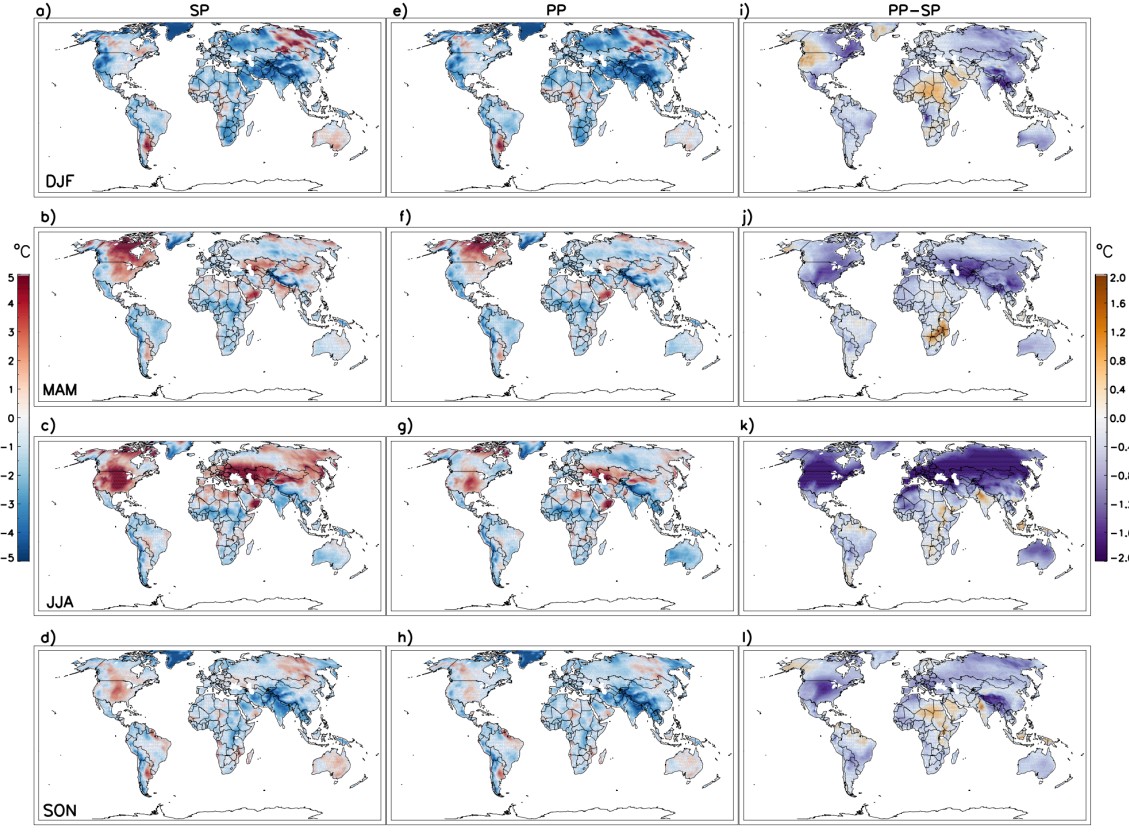


**Figure 6.** Biases of SP temperature over land in a) DJF, b) MAM, c) JJA, and d) SON,

compared with CRU over December 1996 through November 2007. Biases of selected PP

compared with CRU are shown in e)-h), while the differences between selected PP and SP,

i.e. the absolute increase or decrease of biases in PP with respect to the SP values, are

shown in i) - l). The PP results are the composites of the 10 selected sets, 6 IC per set.

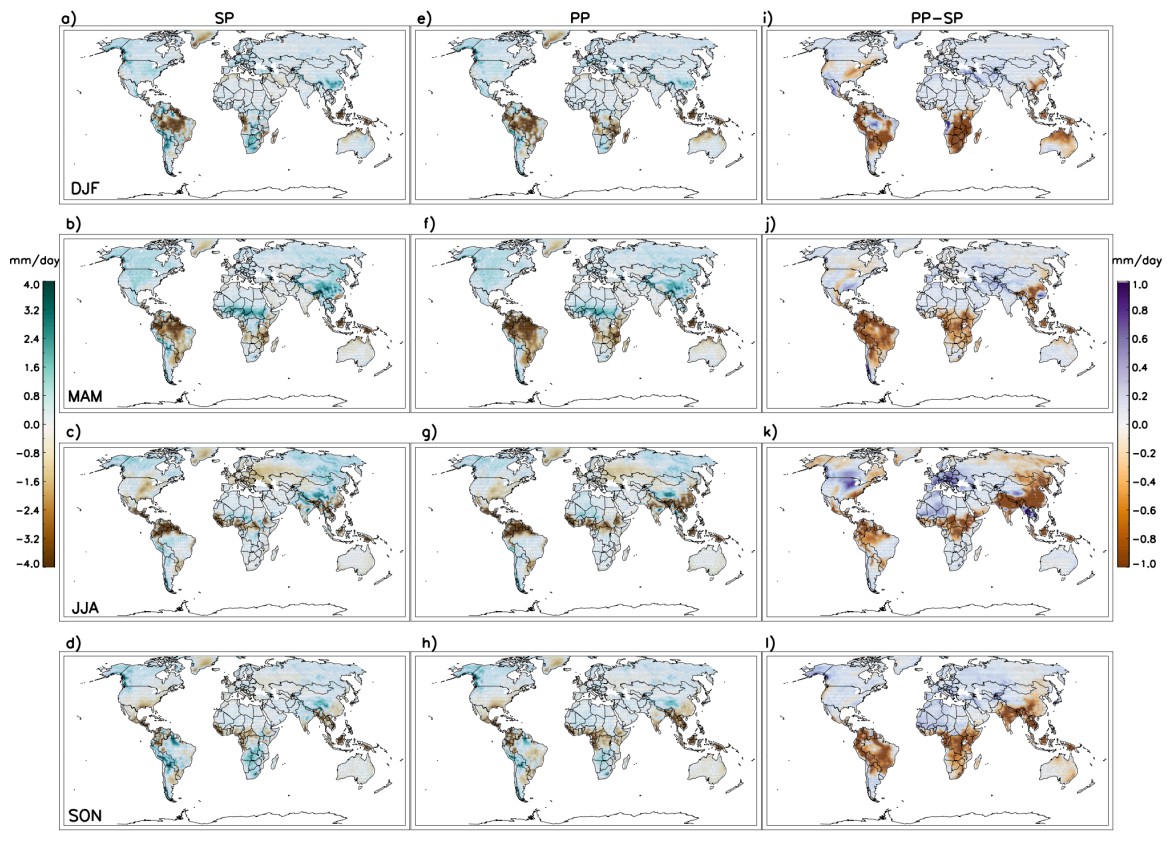


**Figure 7.** Same as Fig. 6, but for precipitation.

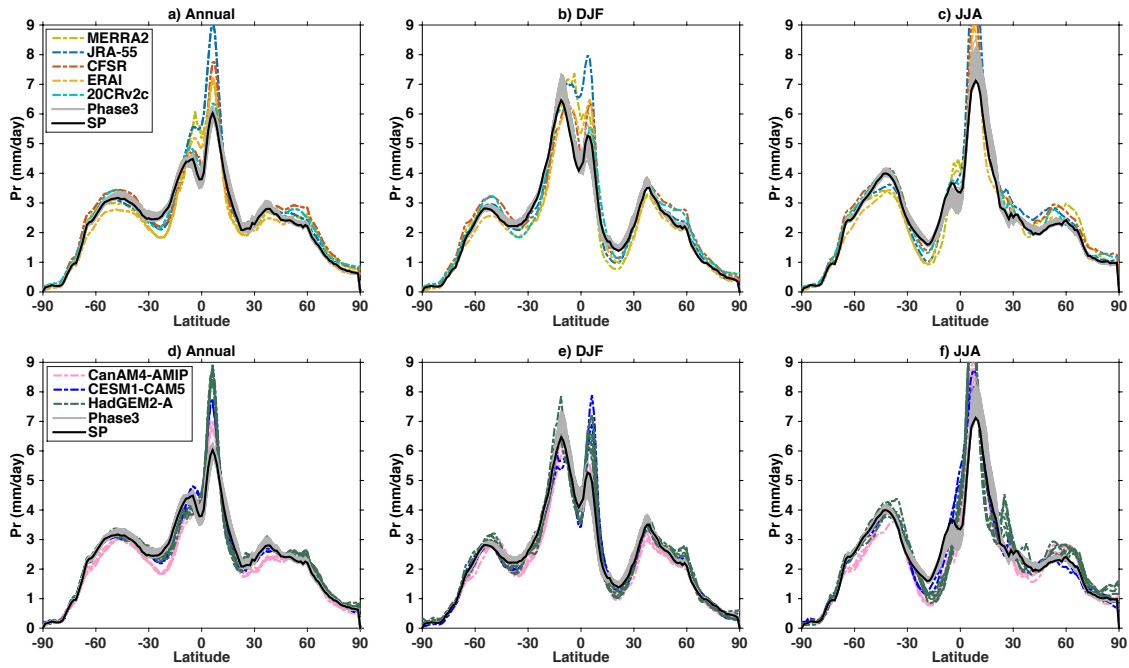


**Figure 8.** Annual (a,d), DJF (b,e) and JJA (c,f) meridional distributions of precipitation
from Phase 3 and SP (all panels), reanalysis datasets MERRA2, JRA-55, CFSR, ERAI and
20CRv2c shown (a - c) and GCMs CanAM4-AMIP, CESM1-CAM5, and HadGEM2-A,
shown in (d - f ).

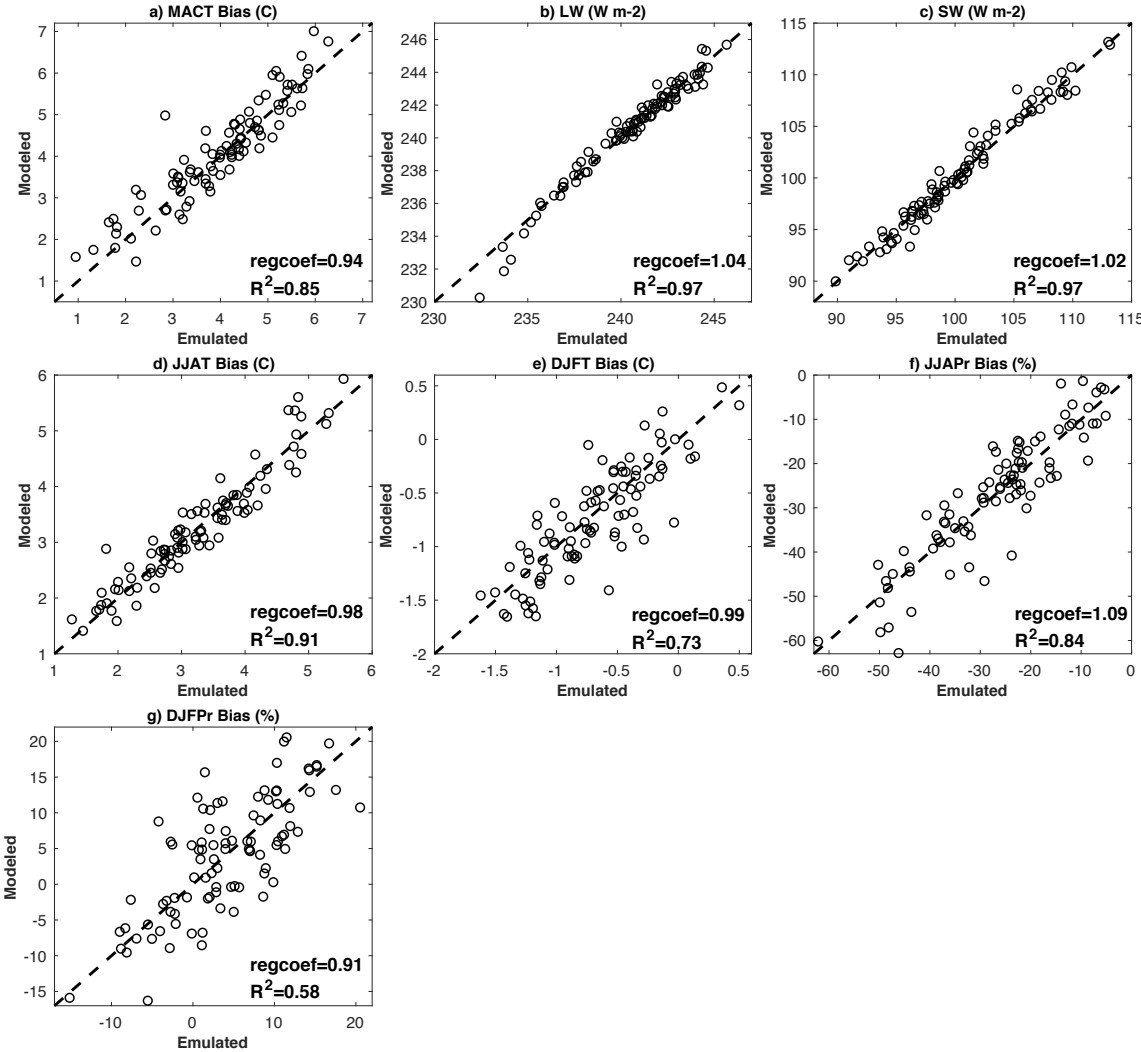

**Figure B1.** Emulator predicted results vs. model simulated results in Phase 2 for different
model output metrics based on 94 parameter sets not used to train the emulator (the 94 sets
that finished after starting Phase3). The regression coefficient (regcoef) and coefficient of
determination ($R^2$) by emulated results are shown in each panel. The dashed line in each
panel denotes the 1:1 line.

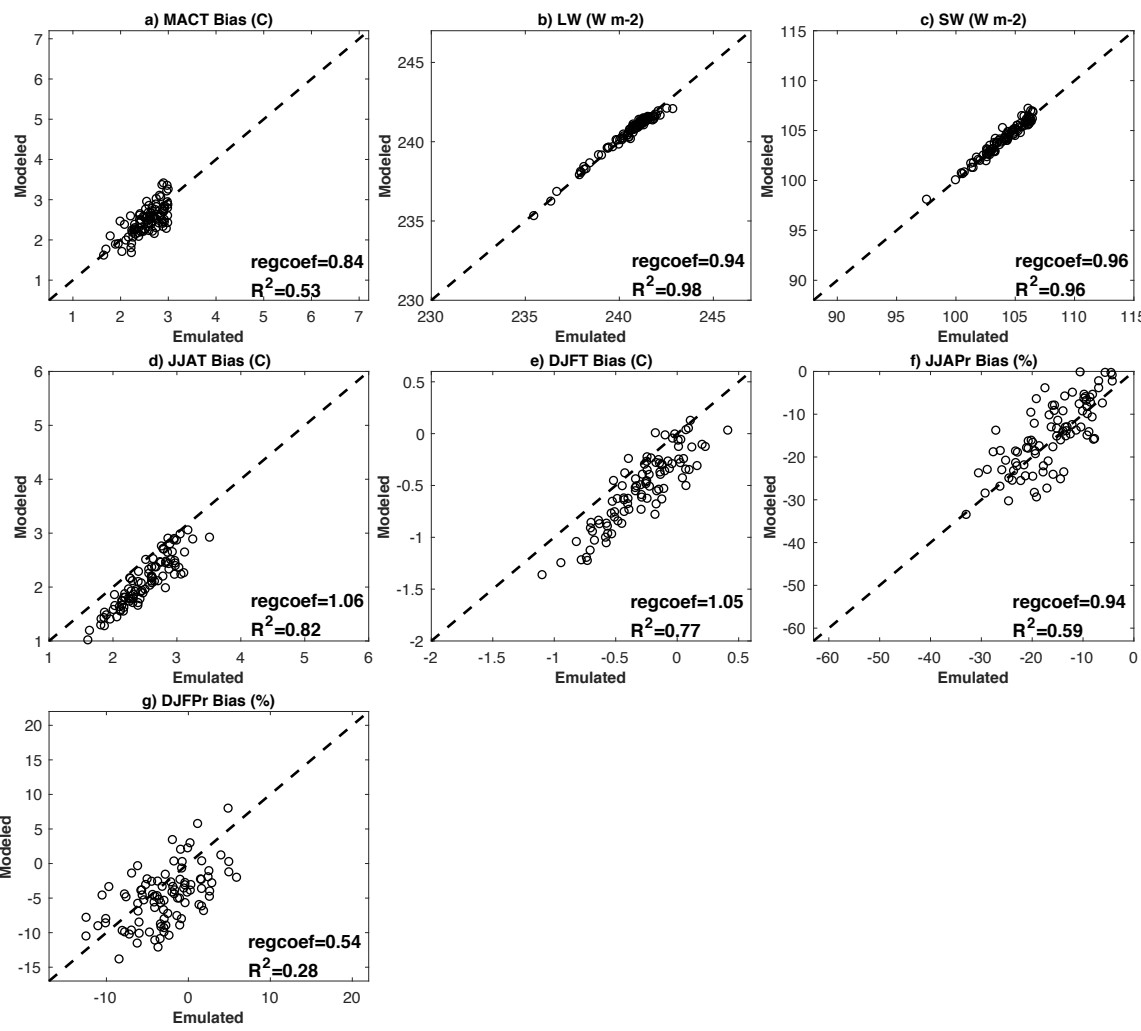


**Figure B2.** Same as Fig. B1, but for the 95 parameter sets in Phase 3. Note the ranges of
x- and y-axis are set to be the same as in Fig. B1.






**Table 1.** Parameters perturbed in our tuning exercise with the post-culling parameters highlighted in bold.

| Parameter | Default | Low | High | Description | Model component |
|---|---|---|---|---|---|
| CT (s$^{-1}$) | $6\times10^{-4}$ | $0.5\times10^{-4}$ | $1.2\times10^{-3}$ | Rate at which cloud liquid water is converted to precipitation | Cloud |
| CW_SEA (kg m$^{-3}$) | $2.0\times10^{-5}$ | $0.5\times10^{-5}$ | $2.0\times10^{-4}$ | Threshold cloud liquid water content over sea | Cloud |
| **CW_LAND** (kg m$^{-3}$) | $1.0\times10^{-3}$ | $0.5\times10^{-3}$ | $1.0\times10^{-2}$ | Threshold cloud liquid water content over land | Cloud |
| EACF | 0.5 | 0.5 | 0.6 | Empirically adjusted cloud fraction | Cloud |
| **VF1** (m s$^{-1}$) | 2 | 0.5 | 4 | Ice fall speed | Cloud |
| **ENTCOEF** | 3 | 0.3 | 9.5 | Entrainment rate coefficient | Convection |
| ALPHAM | 0.5 | 0.45 | 0.65 | Albedo at melting point of sea ice | Radiation |
| DTICE (°C) | 10 | 2 | 11 | Temperature range over which ice albedo varies | Radiation |
| ICE_SIZE (m) | $3.0\times10^{-5}$ | $2.5\times10^{-5}$ | $4.0\times10^{-5}$ | Ice particle size | Radiation |
| KAY_GWAVE (m) | $1.8\times10^{4}$ | $1.0\times10^{4}$ | $2.0\times10^{4}$ | Surface gravity wave drag: typical wavelength | Dynamics |
| KAY_LEE_GWAVE (m$^{-3/2}$) | $2.7\times10^{5}$ | $1.5\times10^{5}$ | $3.0\times10^{5}$ | Surface gravity wave trapped lee wave constant | Dynamics |
| START_LEVEL_GWDRAG | 3 | 3 | 5 | Lowest model level for gravity wave drag | Dynamics |
| **V_CRIT_ALPHA** | 0.5 | 0.01 | 0.99 | Control of photosynthesis with soil moisture | Land surface |
| **ASYM_LAMBDA** | 0.15 | 0.05 | 0.5 | Vertical distance over which air parcels travel before mixing with their surroundings | Boundary layer |

| | | | | | |
|---|---|---|---|---|---|
| CHARNOCK | 0.012 | 0.009 | 0.020 | Constant in Charnock formula for calculating roughness length for momentum transport over sea | Boundary layer |
| **G0** | 10 | 5 | 20 | Used in calculation of stability function for heat, moisture, and momentum transport | Boundary layer |
| **Z0FSEA** (m) | $1.3\times10^{-3}$ | $2.0\times10^{-4}$ | $5\times10^{-3}$ | Roughness length for free heat and moisture transport over the sea | Boundary layer |


**Table 2.** The specifics of four ensembles used in this study.

| Experiment | Start dates | Number of parameters | Number of parameter sets in PPE | IC per parameter set per year used in the analysis |
|---|---|---|---|---|
| SP | 1 Dec 1995, 1996, …, 2005 | 1 | 1 | 6 |
| PPE Phase 1 | 1 Dec 1995 | 17 | 220 | 3 |
| PPE Phase 2 | 1 Dec 1995, 1996, …, 2005 | 17 | 264 | 3 |
| PPE Phase 3 | 1 Dec 1995, 1996, …, 2005 | 7 | 95 | 6 |



 **Supplementary Information**

Subsequent to the model tuning in this study, a large ensemble of climatology simulations
(from October1988 to September2015) were run with PP set2 from the final selected 10
sets, with more than 100 simulations per year. Some initial analysis of the surface energy
budget and surface radiative fluxes from this PP climatology were compared with a large
ensemble of climatology simulations under SP to better understand the reduction in near
surface temperature biases, shown in Fig. S16.
**Table S1.** Information of models used in Fig. 8, including the modelling institutions,
model standard names, pertinent references, and ensemble members shown for each model.

| Modeling institution | Model name | References | Ensemble member |
|---|---|---|---|
| Canadian Centre for Climate Modeling and Analysis | CanAM4 | Chylek et al. (2011) | 4 |
| National Center for Atmospheric Research Community Earth System Model | CESM-CAM5 | Neale et al. (2010) | 2 |
| Met Office Hadley Centre | HadGEM2-A | Martin et al. (2006)  Collins et al. (2011) | 6 |


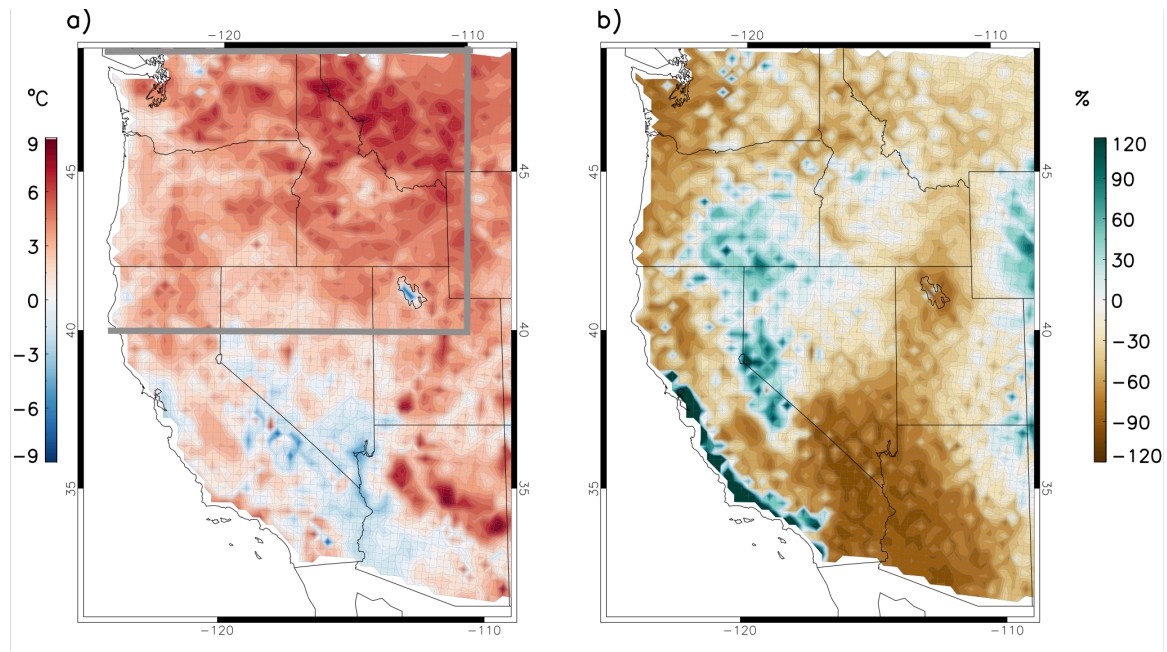


**Figure S1.** Biases in a) June-July-August (JJA) mean temperature (°C) , and b)
precipitation (%) simulated by HadRM3P compared with PRISM over dec1996-nov 2007
under standard physics (SP) setting. The NWUS is defined as the land region bounded by
the heavy grey line.

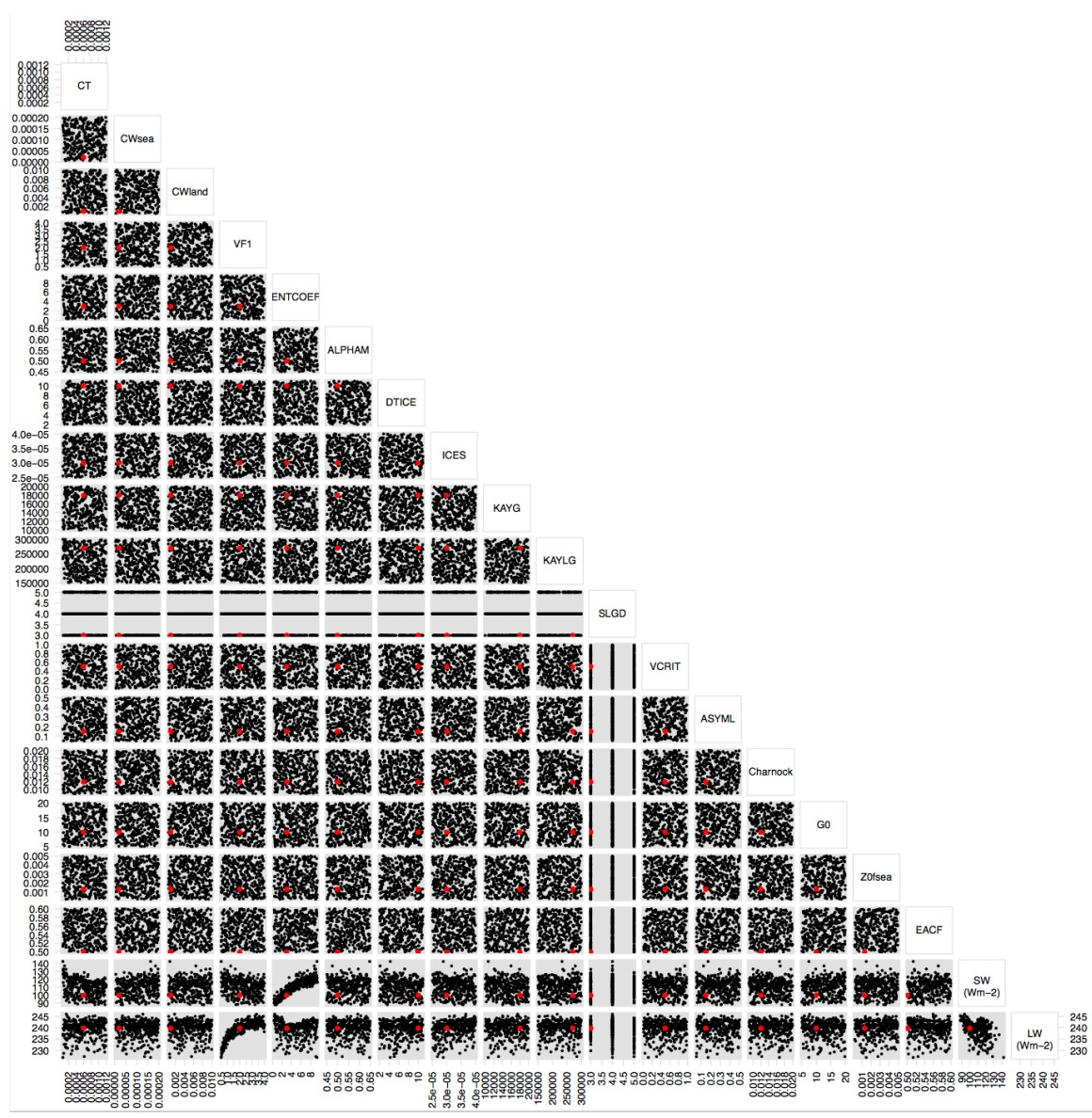


**Figure S2.** Phase 1 PPE parameter inputs and TOA outgoing SW and LW fluxes. 328

parameter sets are shown. The parameter values and model outputs under SP setting are

marked in red.

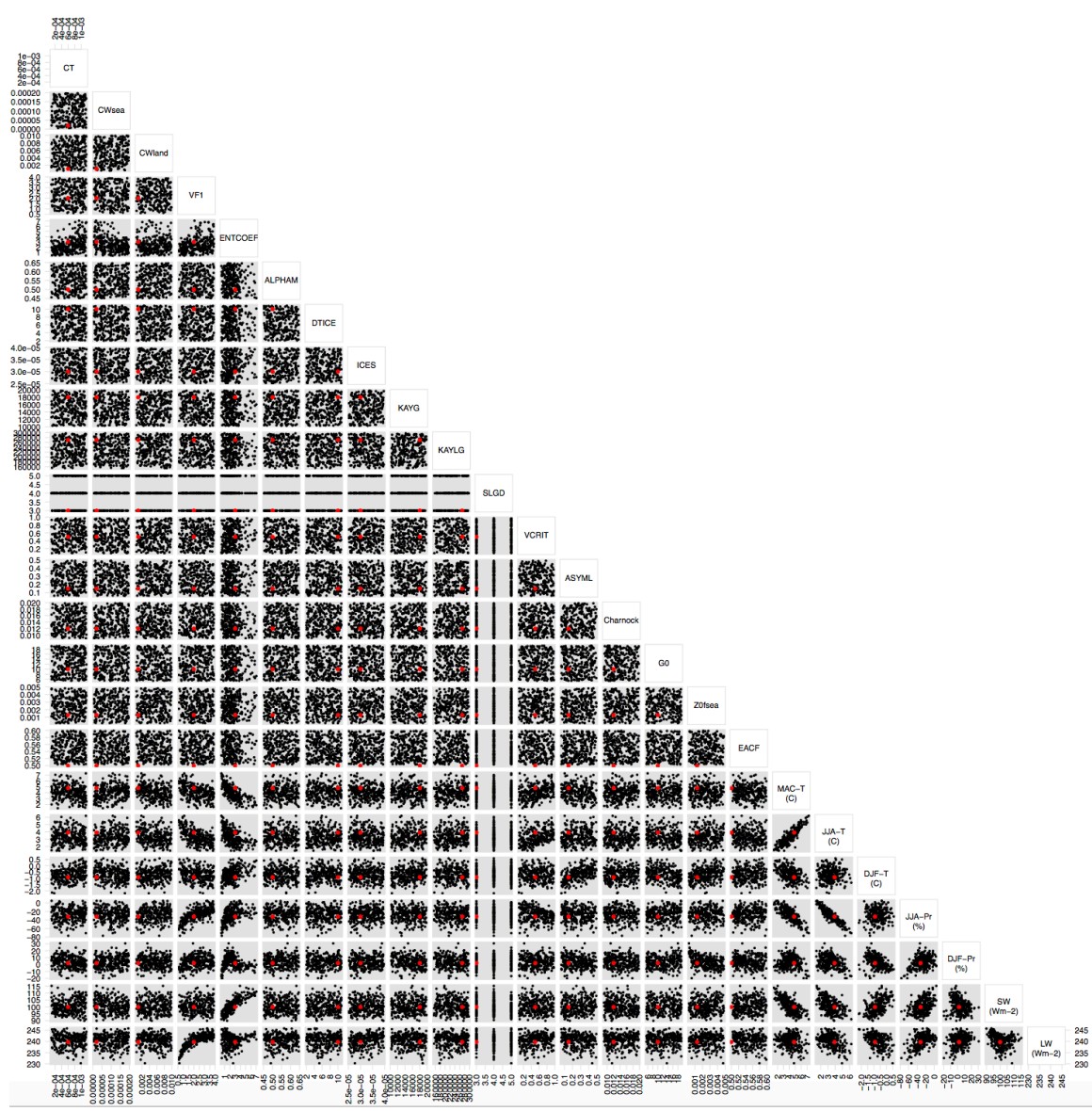


**Figure S3.** Same as Fig. S3, but for Phase 2 parameter inputs and summary model output metrics considered in this phase. 264 parameter sets are shown.

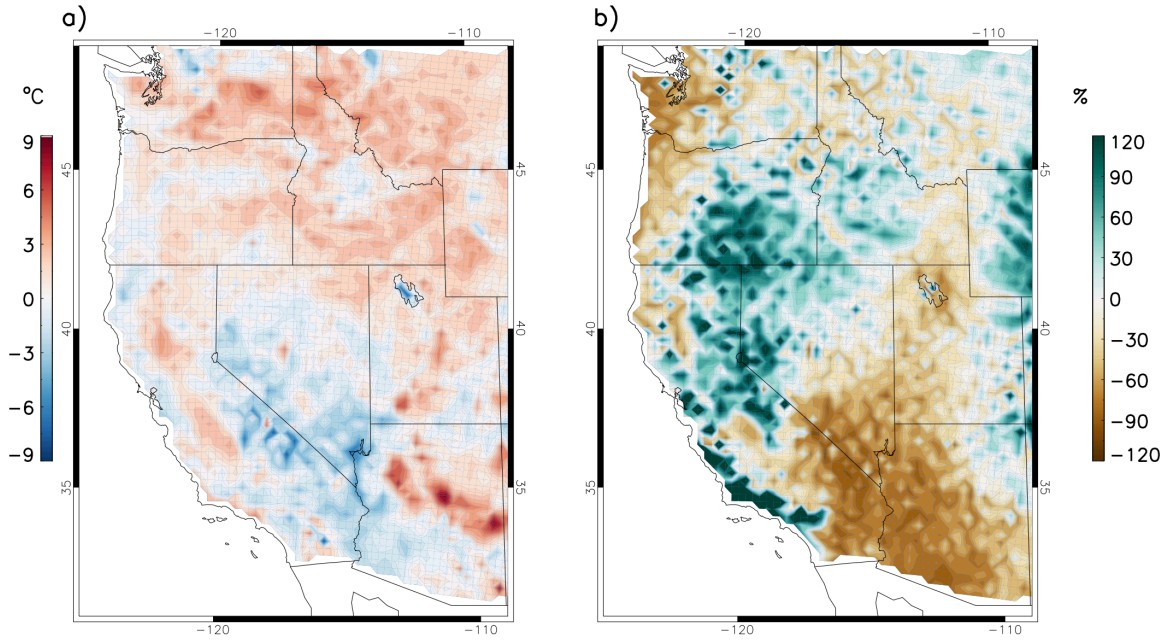


**Figure S4.** Biases in a) June-July-August (JJA) mean temperature (°C) , and b)
precipitation (%) simulated by HadRM3P compared with PRISM over dec1996-nov 2007
under the selected PP settings, where the composite of the final 10 are taken.

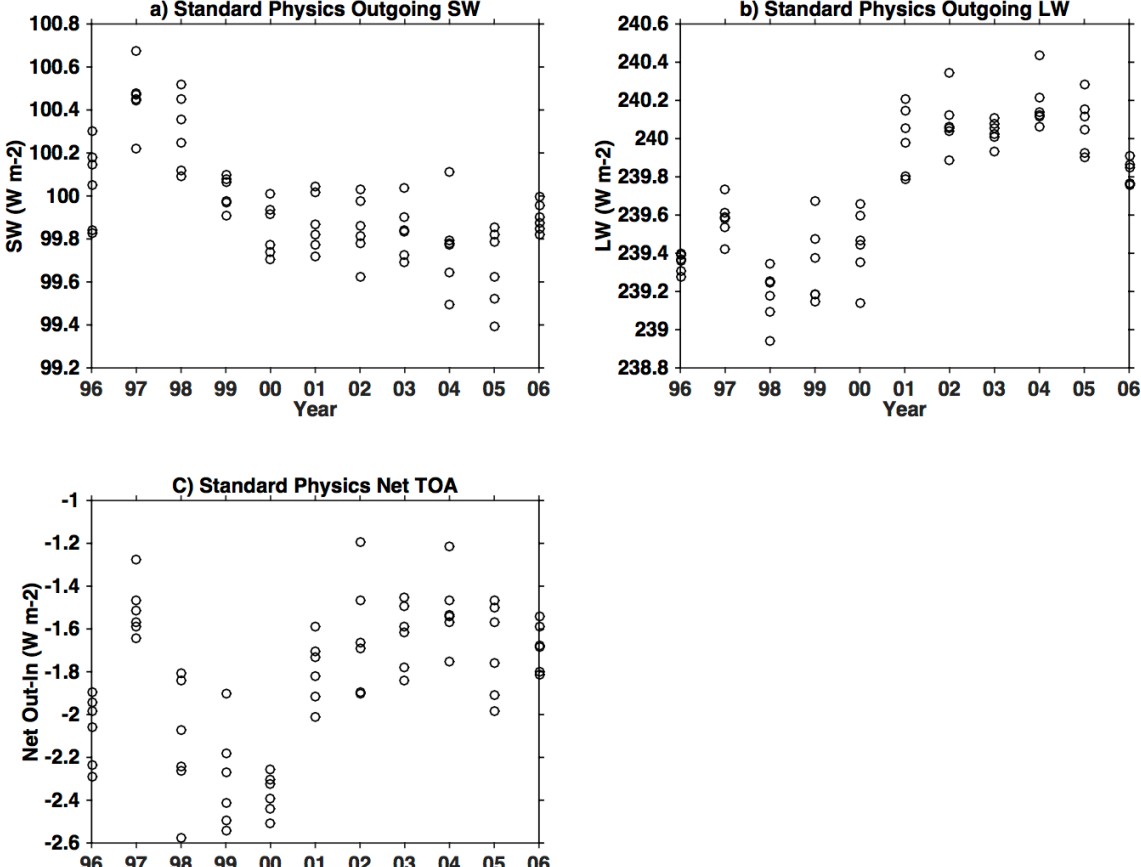


**Figure S5.** The range of internal variability for top-of-atmosphere a) outgoing shortwave
radiation, b) outgoing longwave radiation, and c) net (outgoing minus incoming) under SP
setting for each year. We rounded to the nearest $Wm^{-2}$ ($\pm 1$) to account for internal
variability.

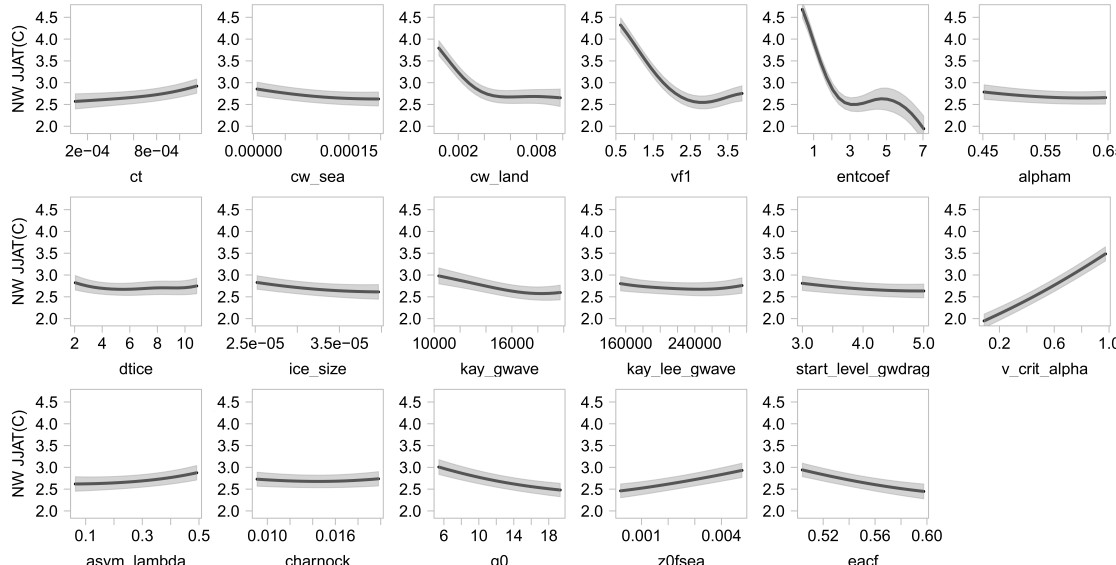


**Figure S6.** One-at-a-time sensitivity analysis of JJA temperature bias (compared with
PRISM) over Northwest to each input parameter in turn, with all other parameters held at
mean value of all the designed points. Central lines represent the emulator mean, and
shaded areas represent the estimate of emulator uncertainty, at the ±1 SD level.

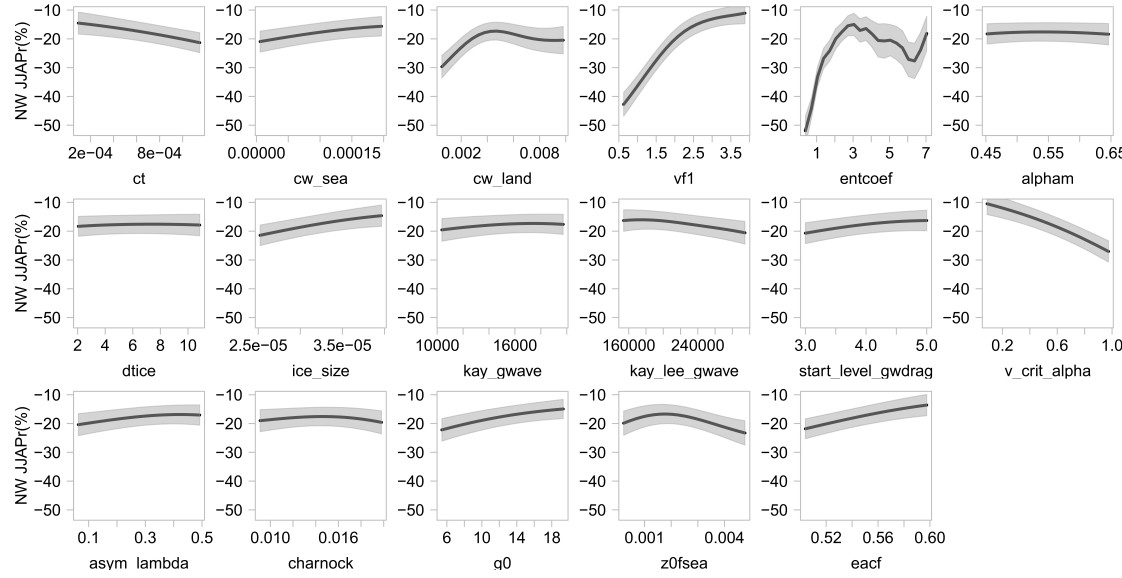


**Figure S7.** Same as Fig. S6, but for DJF temperature bias.

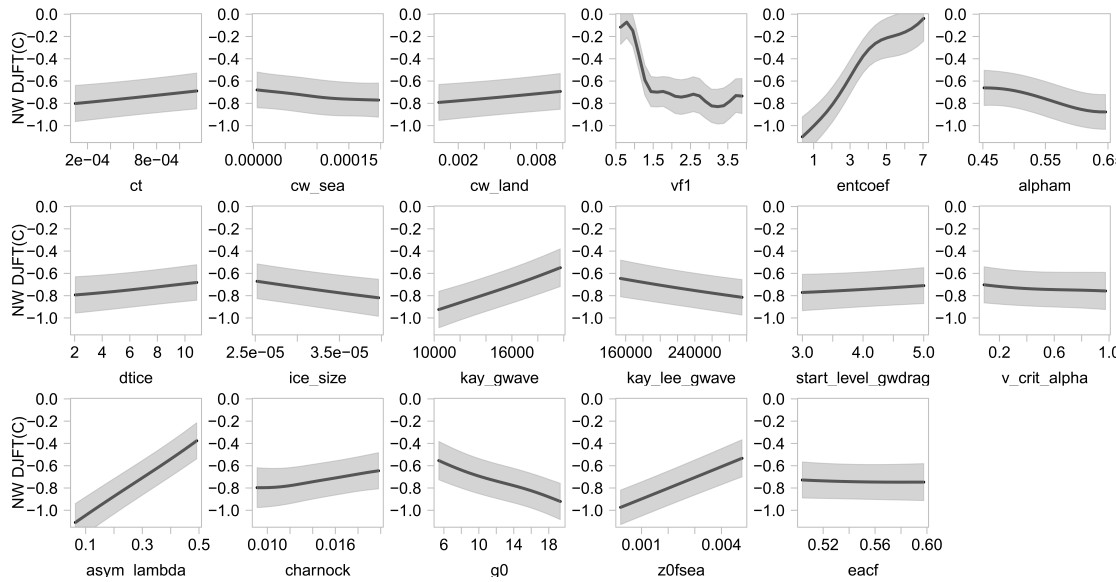


**Figure S8.** Same as Fig. S6, but for JJA precipitation bias.

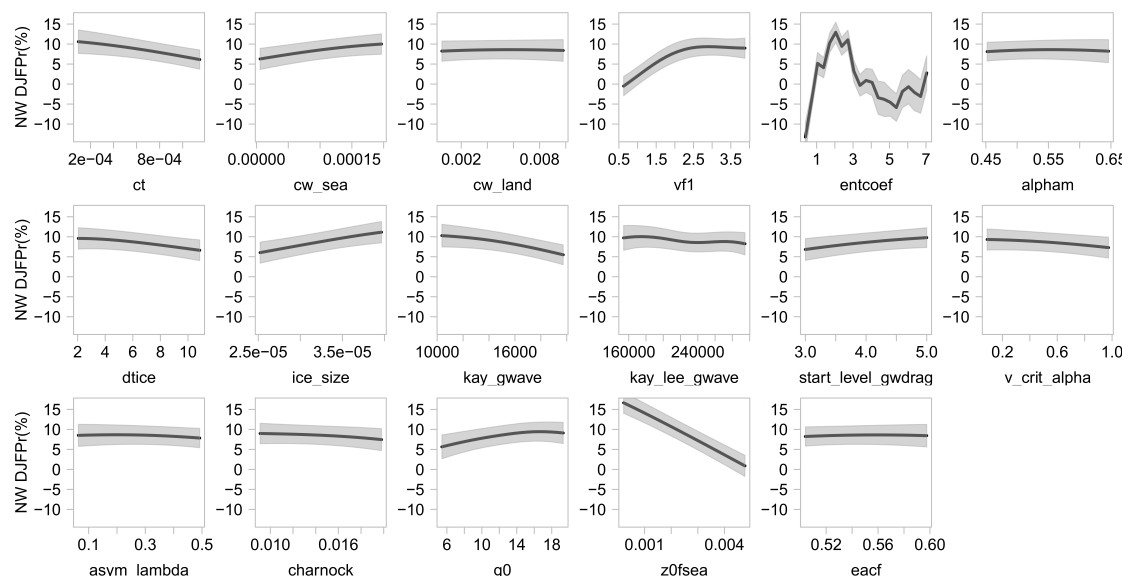


**Figure S9.** Same as Fig. S6, but for DJF precipitation bias.

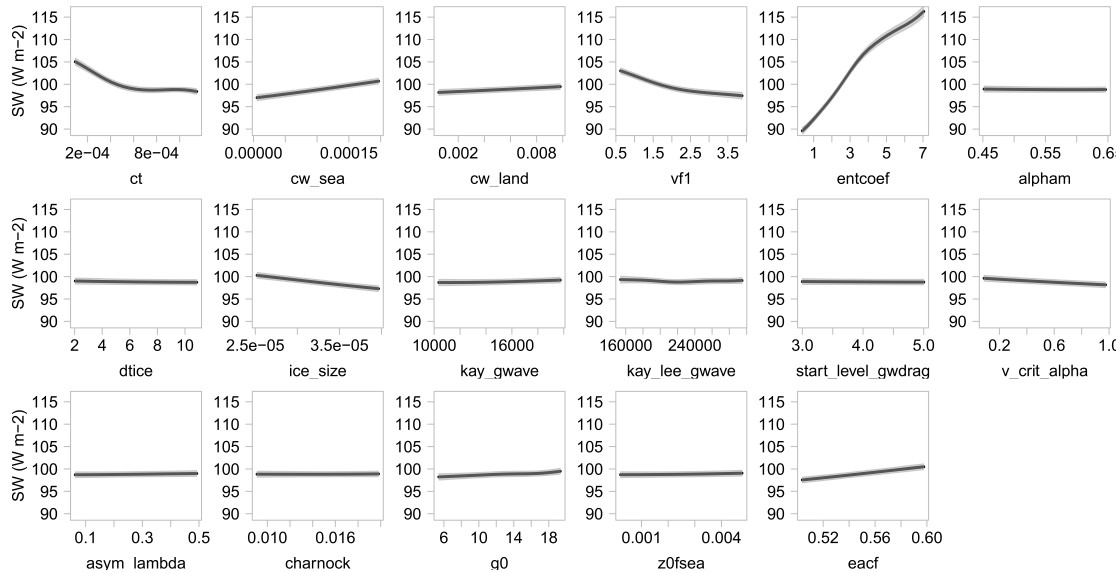


**Figure S10.** Same as Fig. S6, but for TOA SW fluxes.

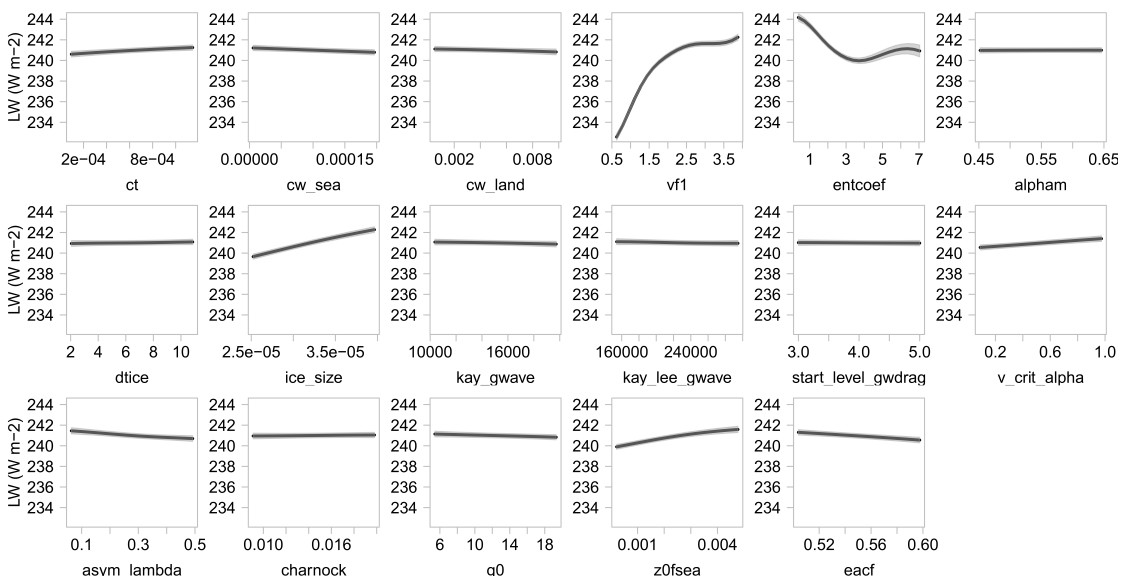


**Figure S11.** Same as Fig. S6, but for TOA LW fluxes.

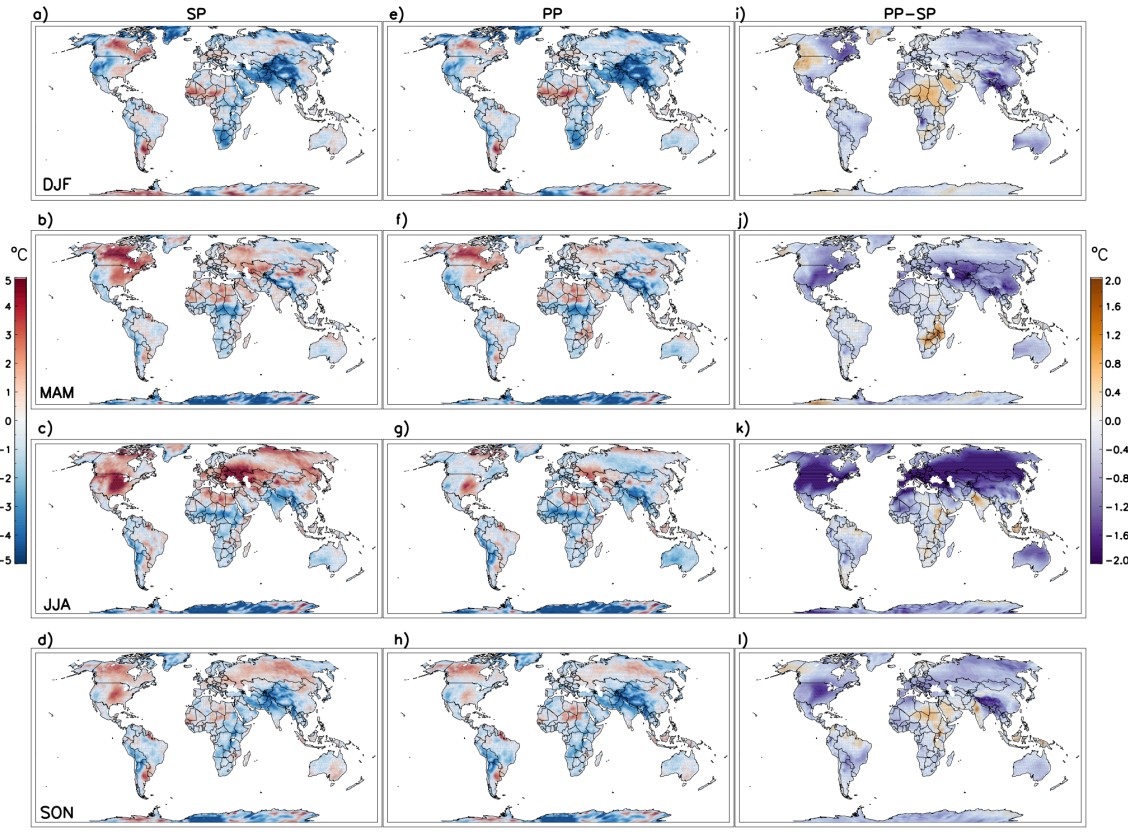


**Figure S12.** Biases of SP temperature over land in a) DJF, b) MAM, c) JJA, and d) SON,
compared with MERRA over December 1996 through November 2007. Biases of selected
PP compared with MERRA are shown in e)-h), while the differences between selected PP
and SP, i.e. the absolute increase or decrease of biases in PP with respect to the SP values,
are shown in i) - l). The PP results are the composites of the 10 selected sets, 6 IC per set.

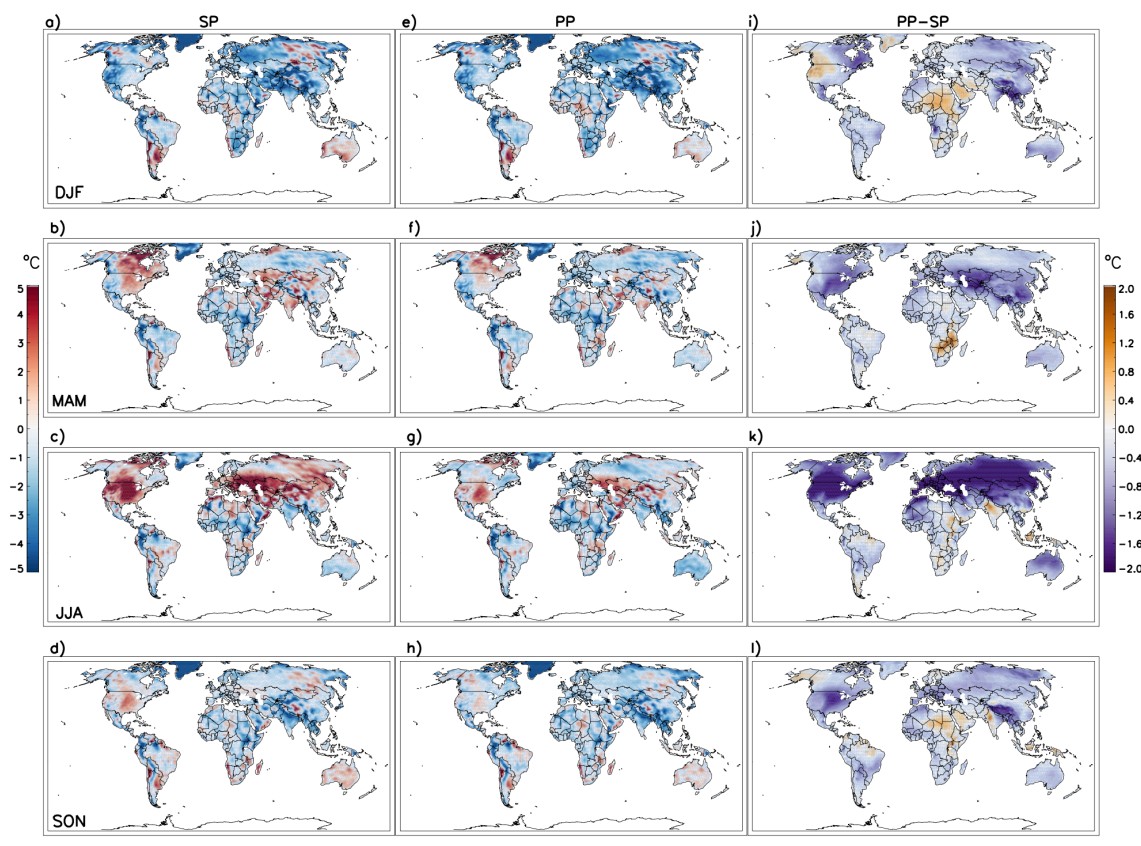


**Figure S13.** Same as Fig. S12, but for comparison with GHCN-CAMS.

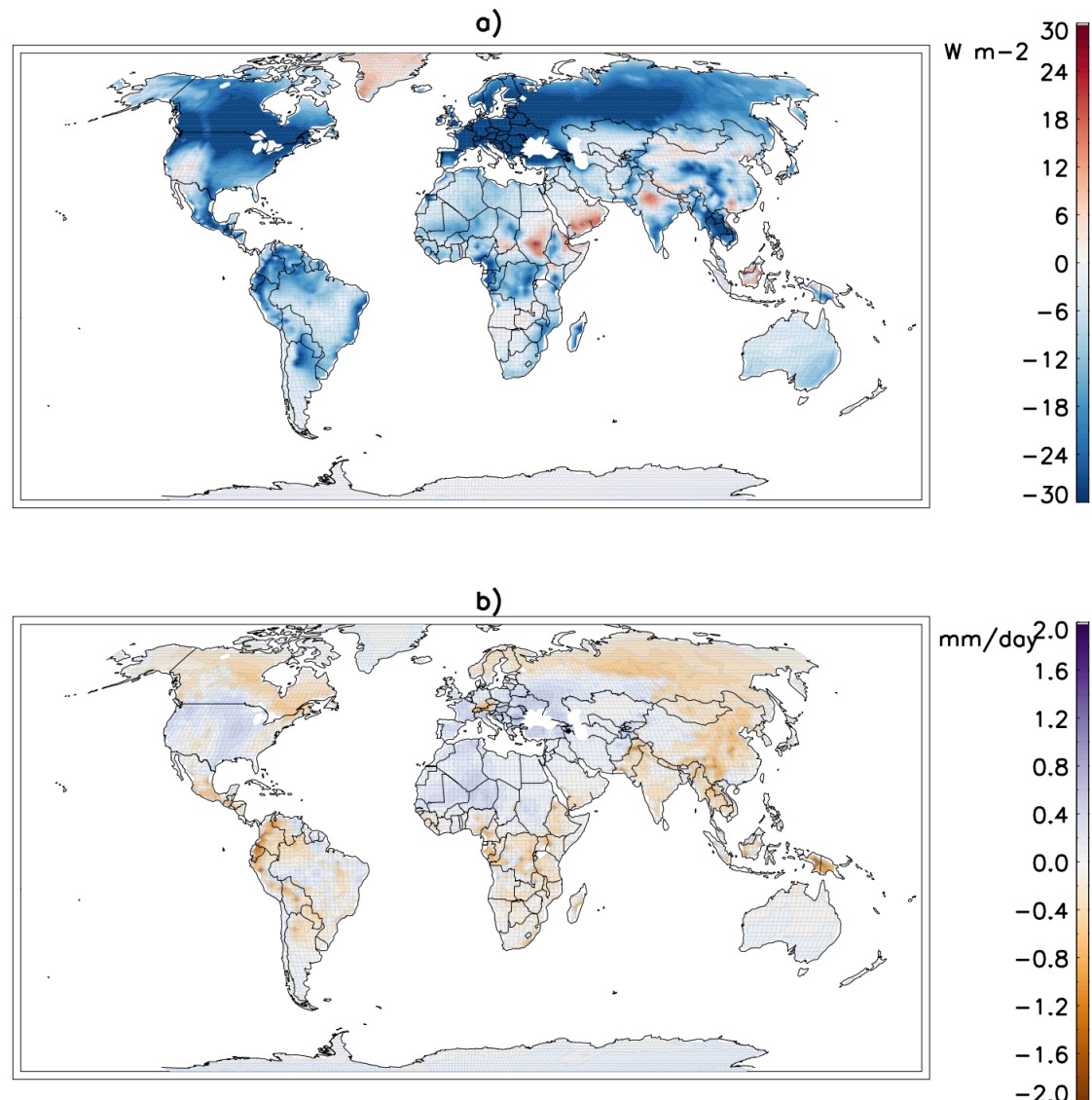


**Figure S14.** MEAN summer (JJA) differences between SP and PPset2 for a) total

downward shortwave radiation, and b) latent heat fluxes for the period Oct1988 – Sep2015.

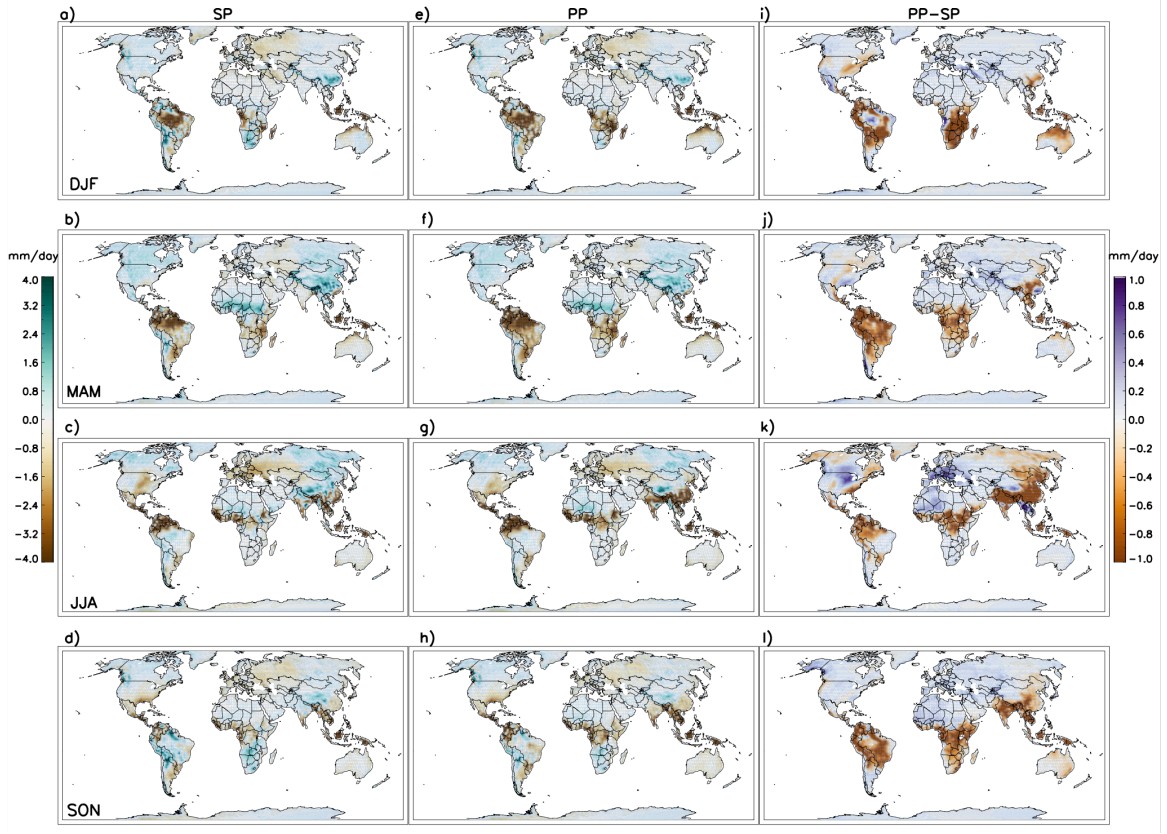


**Figure S15.** Biases of SP precipitation over land in a) DJF, b) MAM, c) JJA, and d) SON,

compared with GPCP over December 1996 through November 2007. Biases of selected PP

compared with GPCP are shown in e)-h), while the differences between selected PP and

SP, i.e. the absolute increase or decrease of biases in PP with respect to the SP values,  are

shown in i) - l). The PP results are the composites of the 10 selected sets, 6 IC per set.

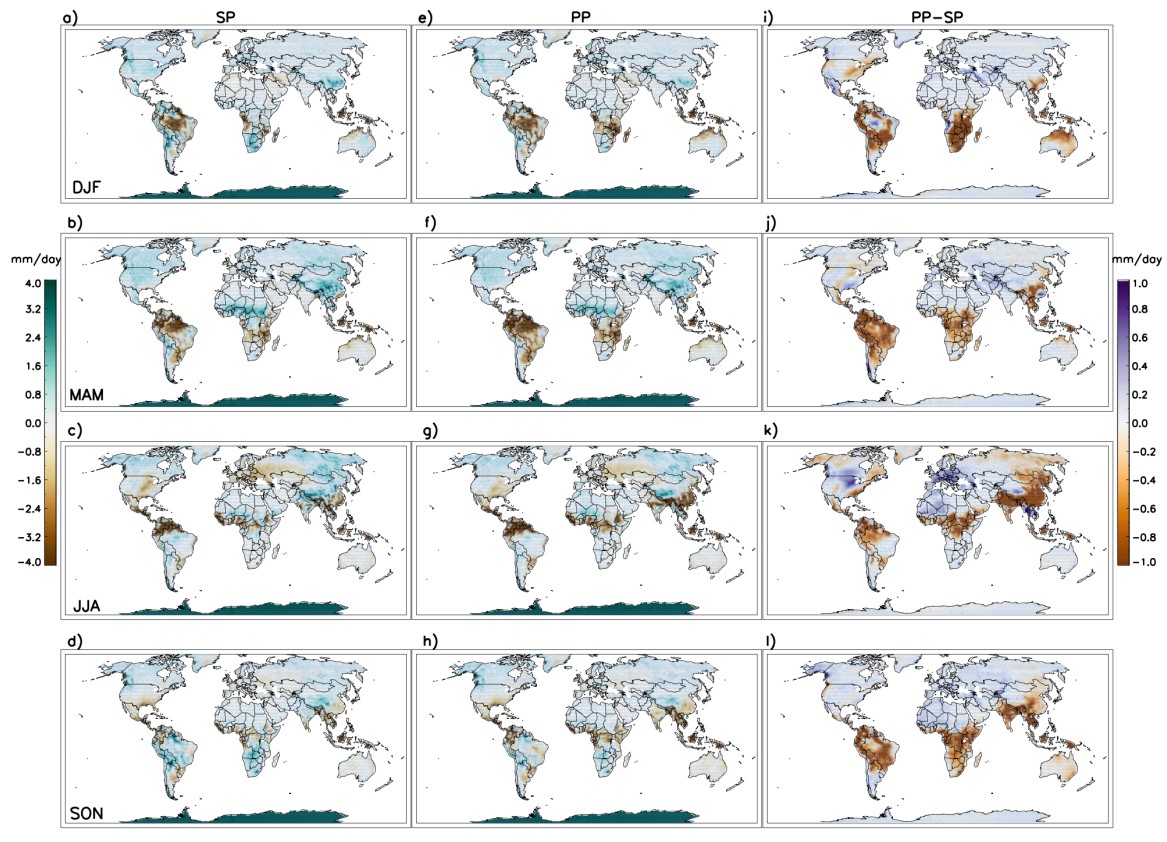


Figure S16. Same as Fig. S15, but for comparison with GPCC.

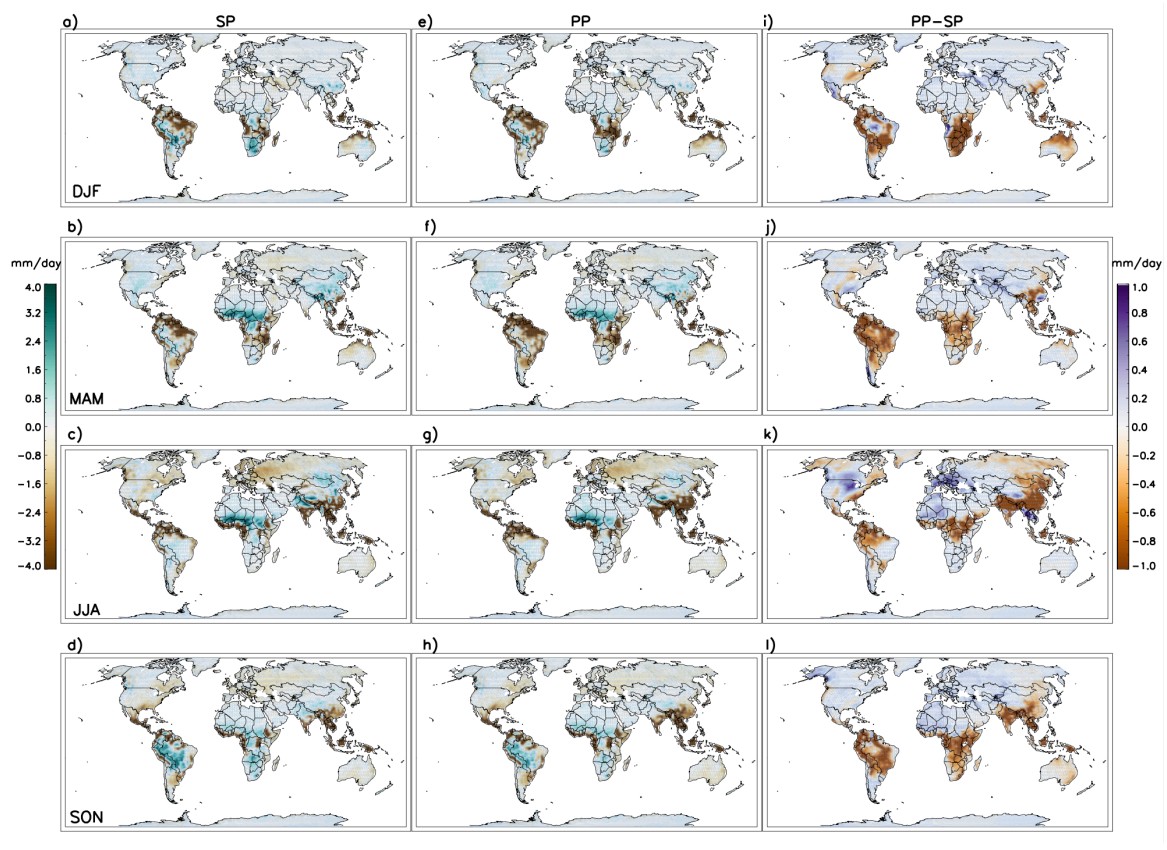


**Figure S17.** Same as Fig. S15, but for comparison with MERRA.

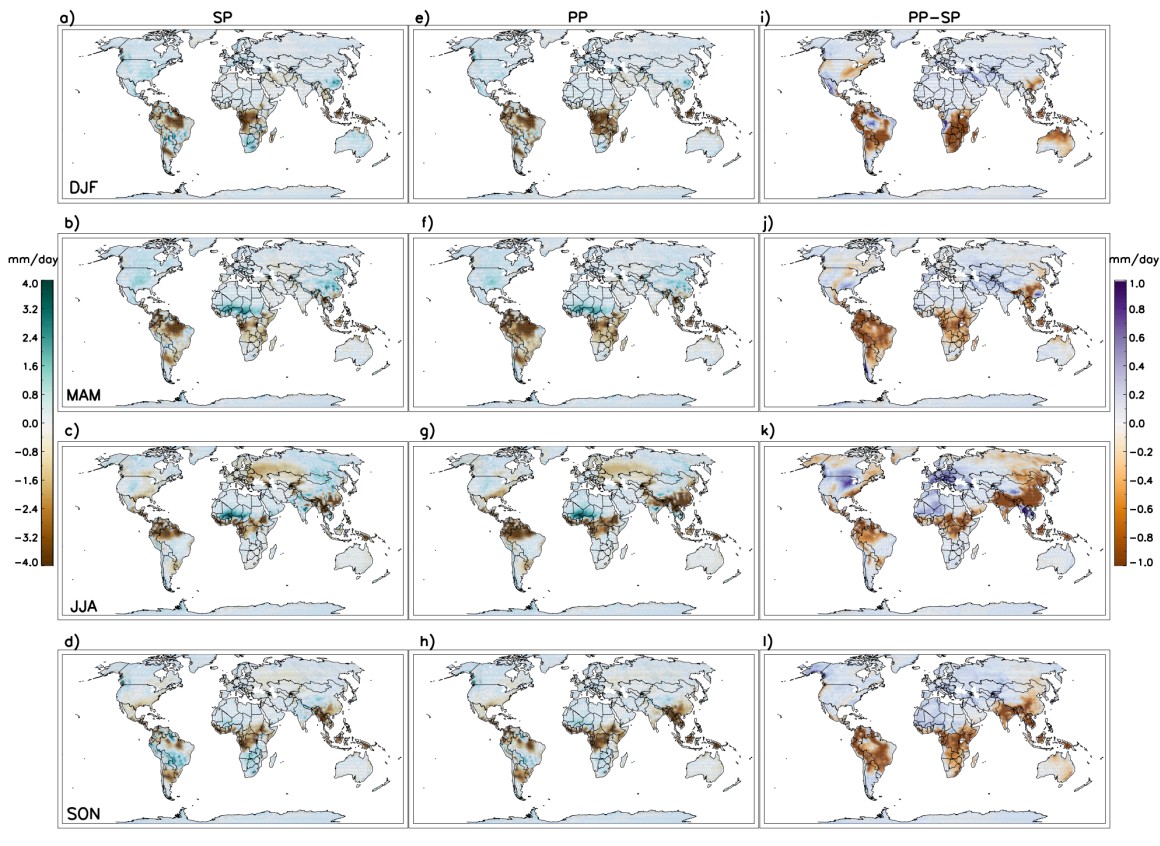


**Figure S18.** Same as Fig. S15, but for comparison with ERAI.

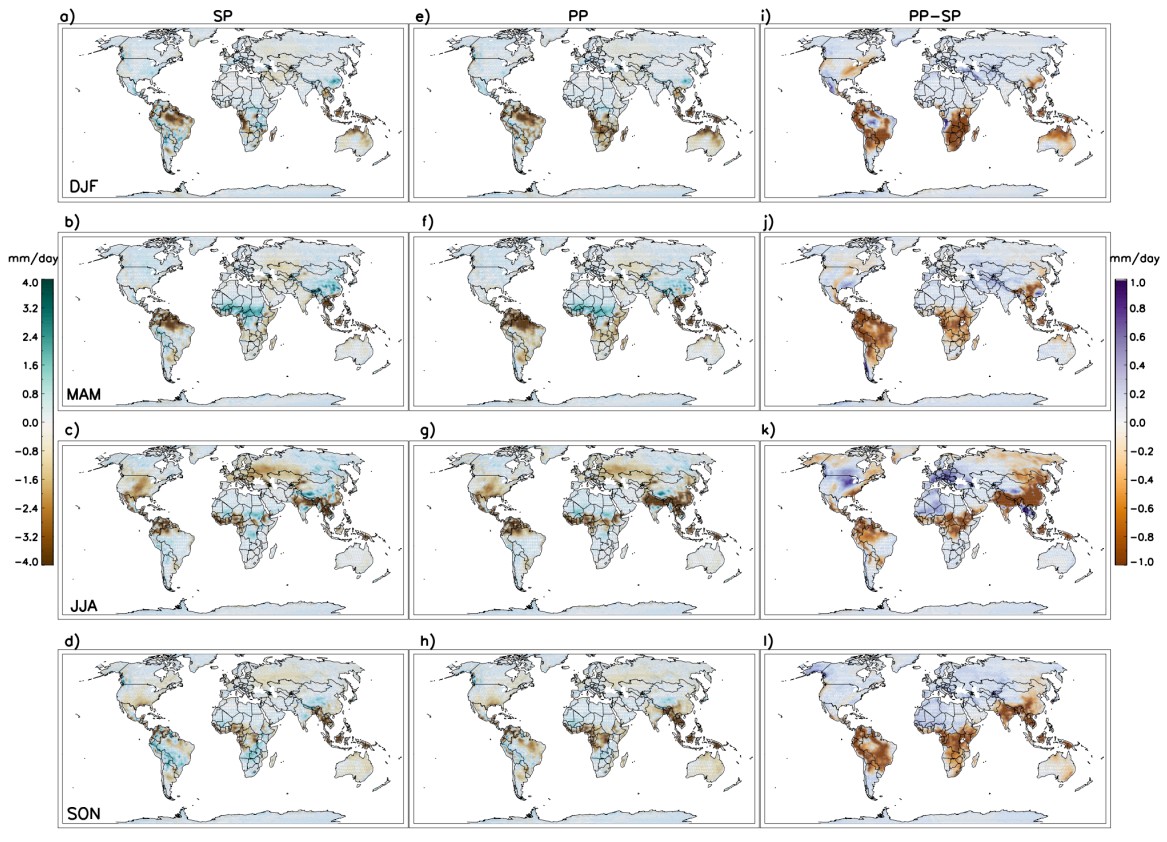


**Figure S19.** Same as Fig. S15, but for comparison with JRA-55.

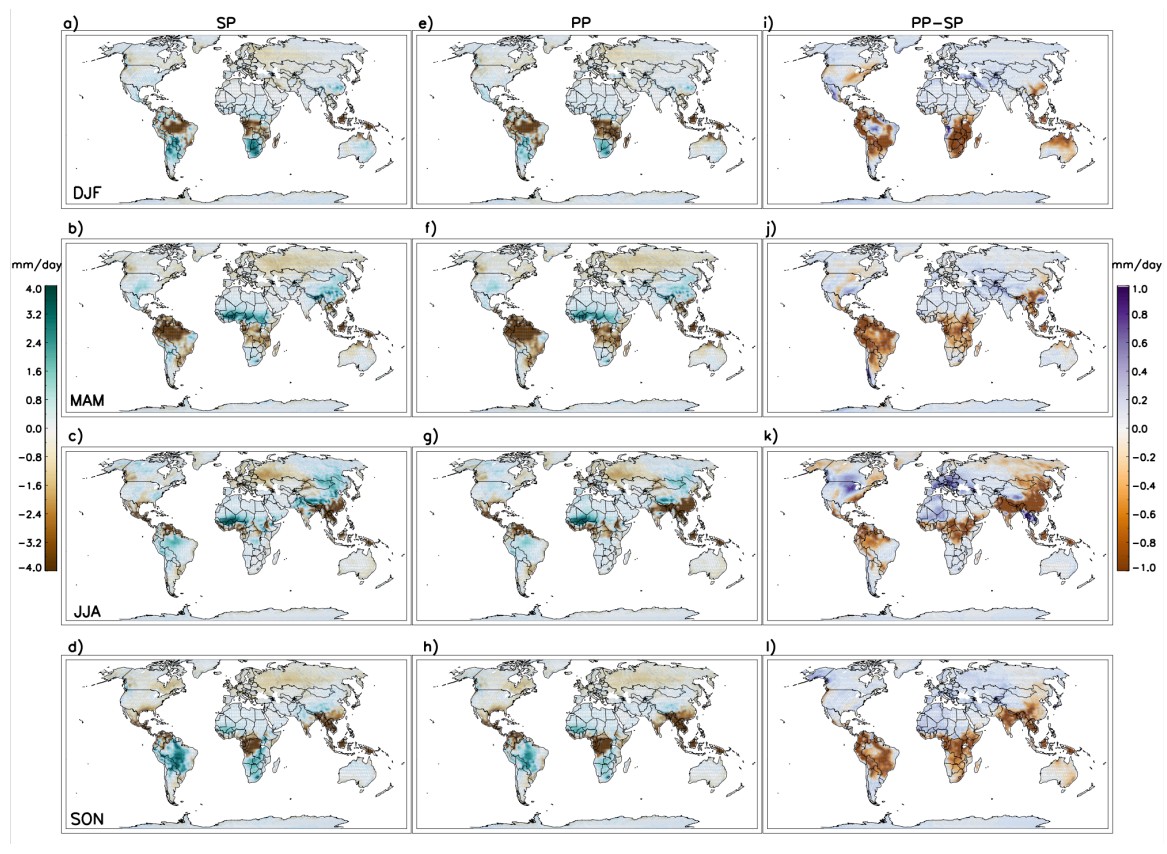

**Figure S20.** Same as Fig. S15, but for comparison with CFSR.

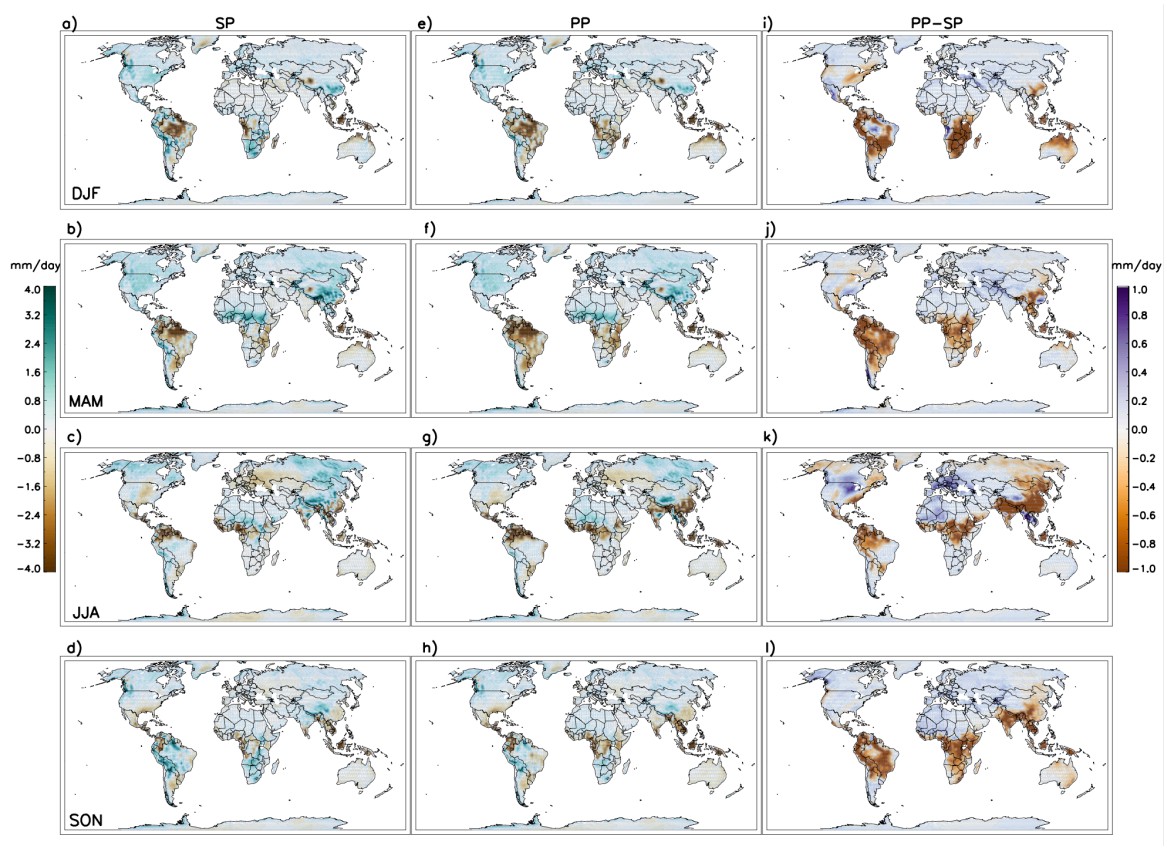


**Figure S21**. Same as Fig. S15, but for comparison with CMAP.

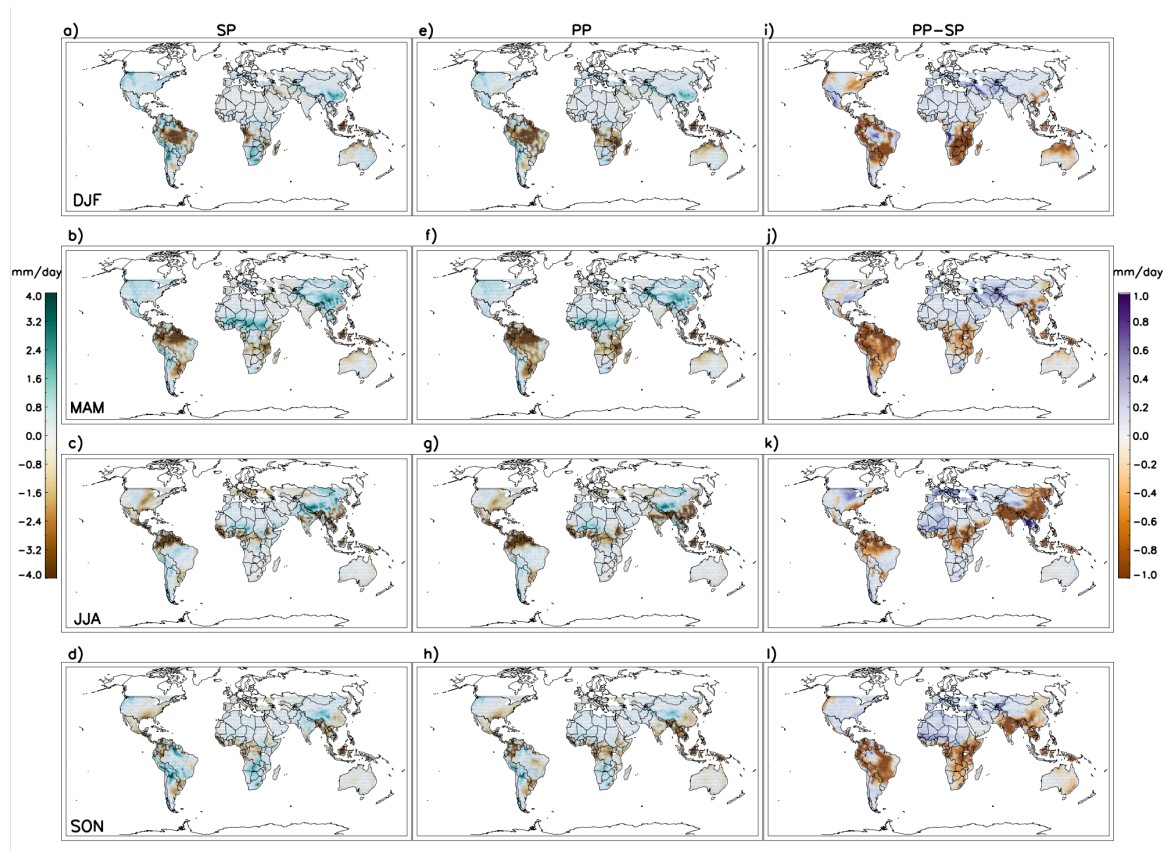


**Figure S22**. Same as Fig. S15, but for comparison with TRMM.

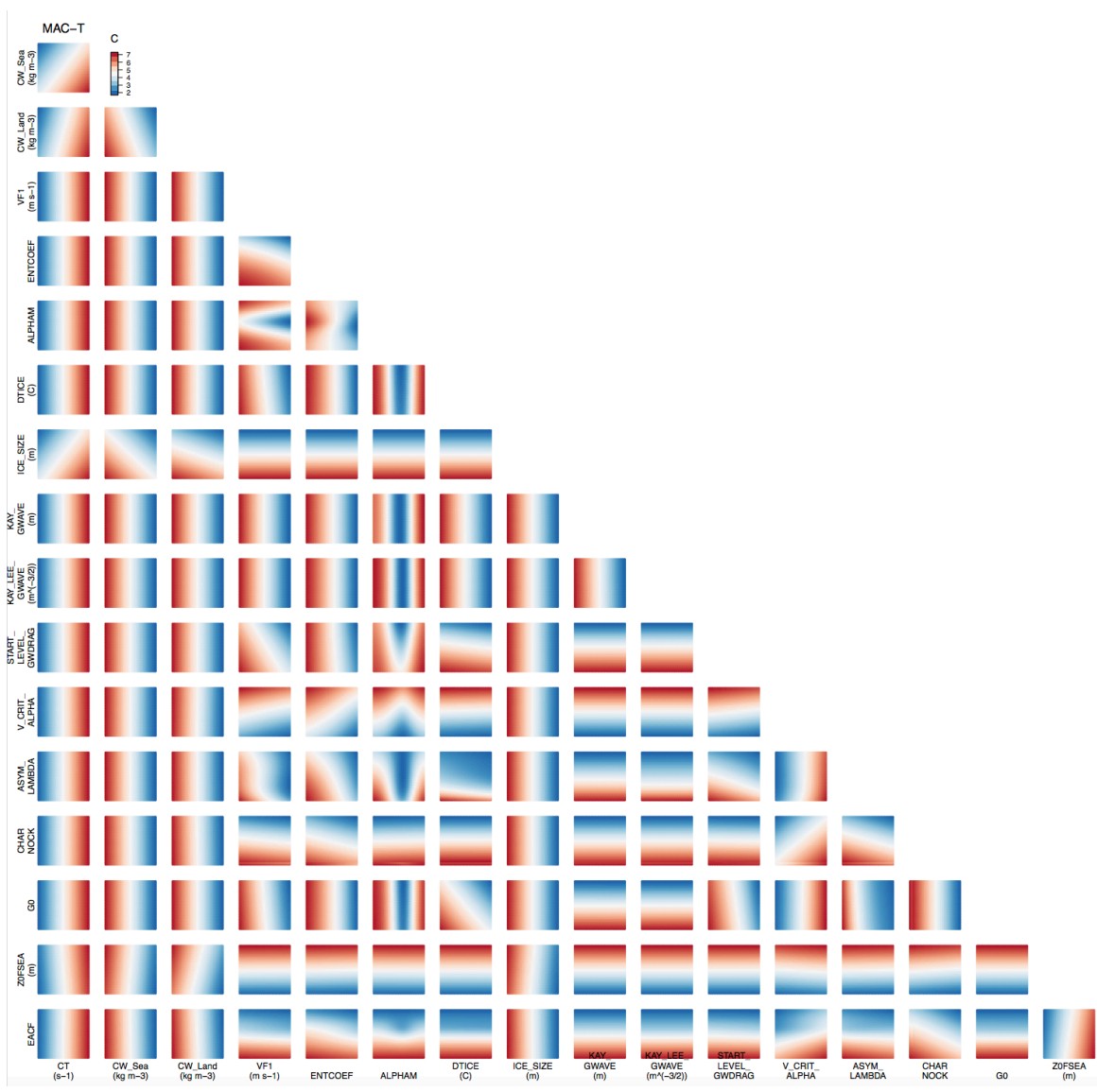

Figure S23. MAC-T biases projected into the two-dimensional spaces of each pair of input parameters using the emulator.

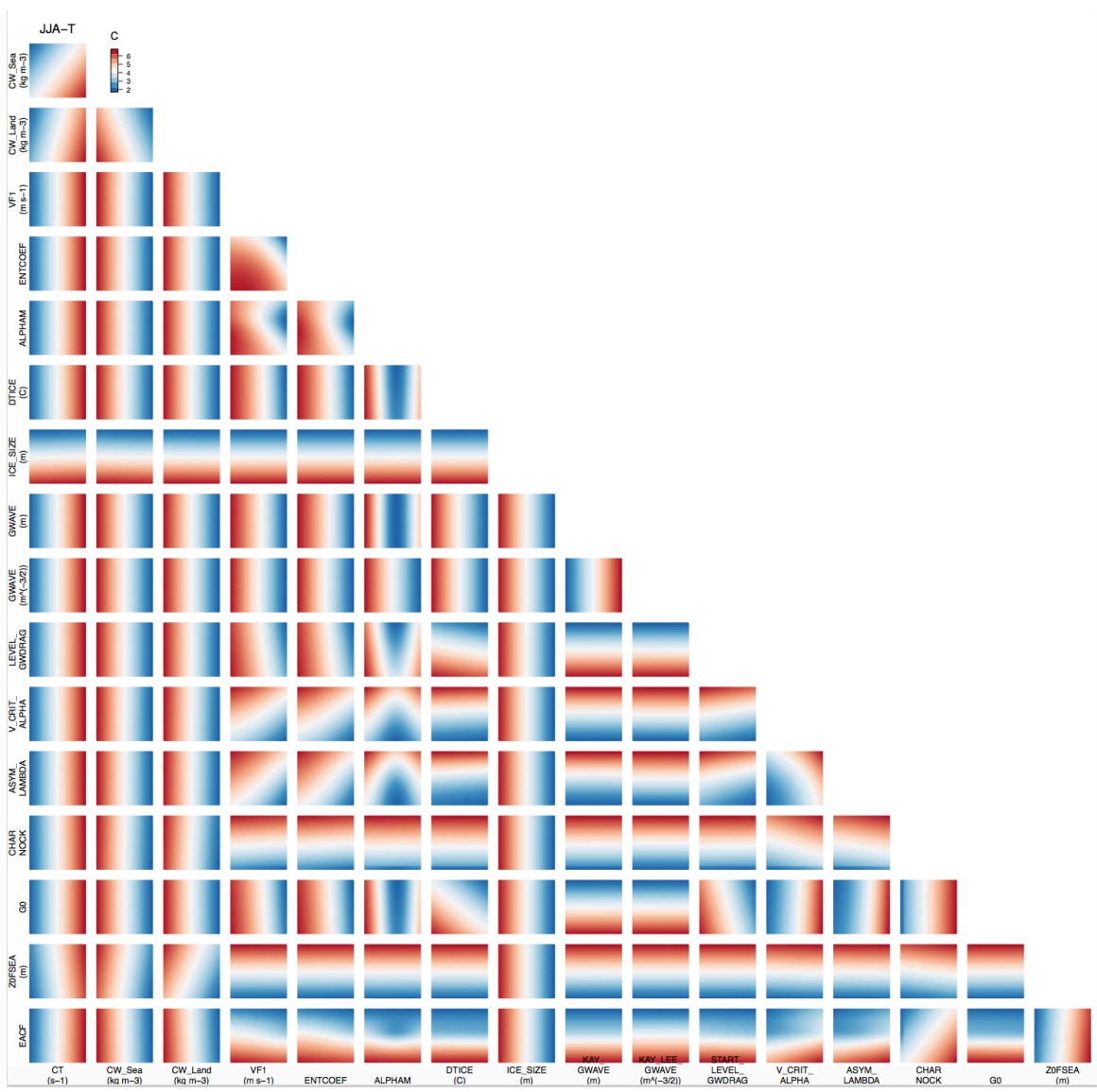


Figure S24. JJA-T biases projected into the two-dimensional spaces of each pair of input
parameters using the emulator.



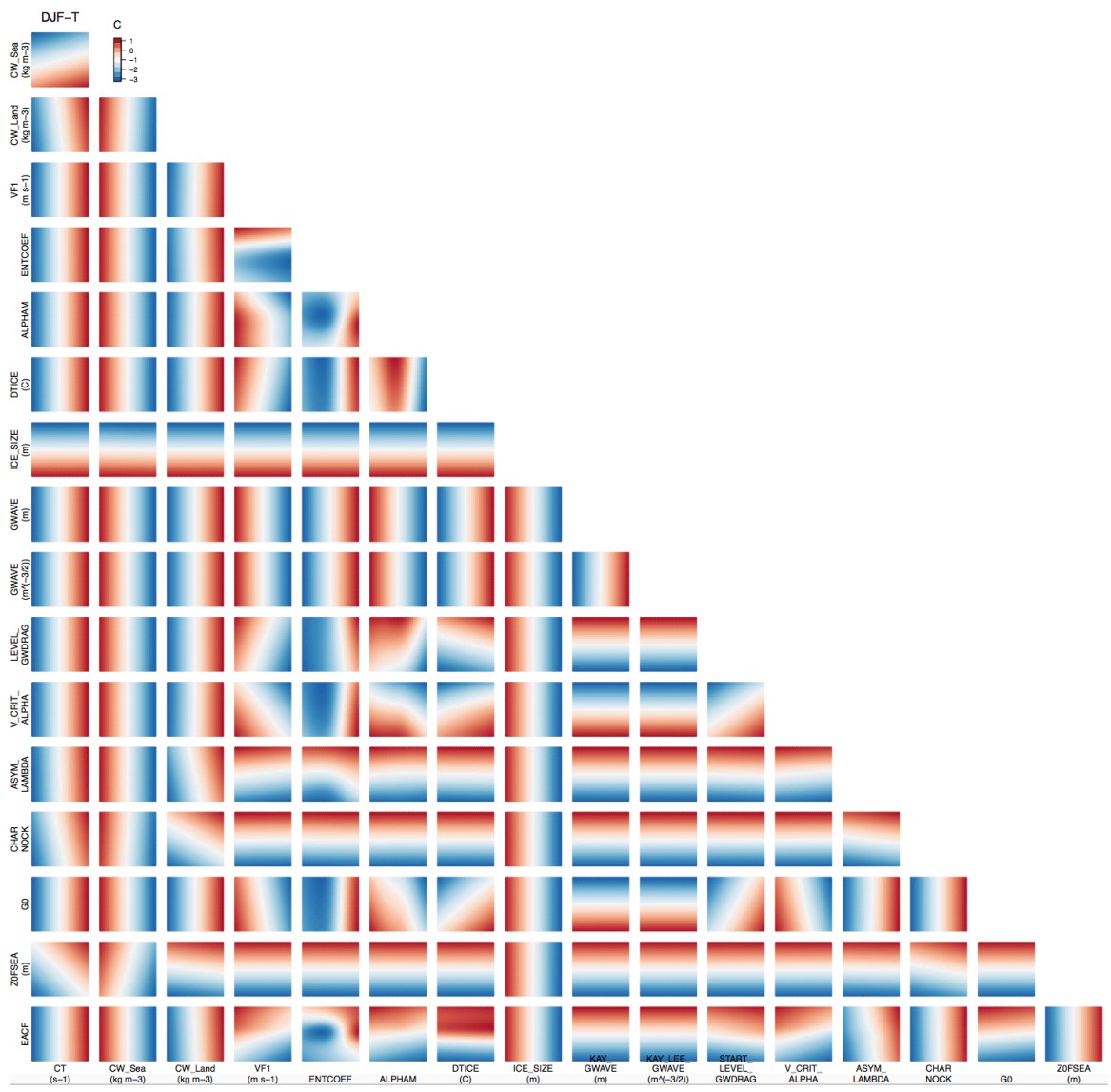


Figure S25. DJF-T biases projected into the two-dimensional spaces of each pair of input

parameters using the emulator.

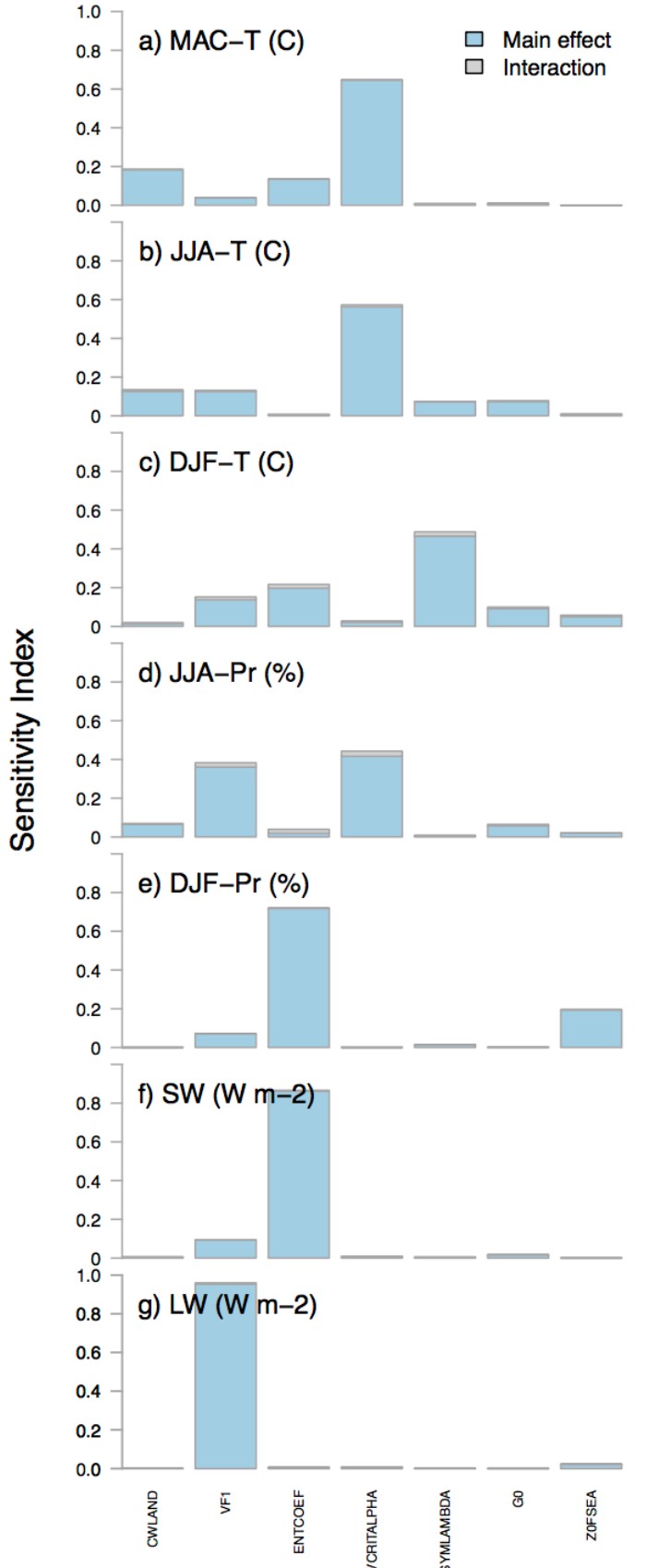


**Figure S26**. The sensitivity indices for the refined parameter space in Phase 3.

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
