# Peer review of "Reducing climate model biases by exploring parameter space with large ensembles of"

_Geoscientific Model Development, 2018_

## Referee Comment (RC1) · Anonymous Referee #1 · 23 Dec 2018

Review

Improving climate model accuracy by exploring parameter space with an O(105) member ensemble and emulator by Li et al.

The paper describes an approach to identify parameters in the atmosphere and land-surface models that control warm and dry biases in summer in the northwestern US with an aim of finding parameter combinations that reduce these regional biases. The authors construct PPEs by perturbing 17 parameters simultaneously using the weather@home modeling framework. They use an 'iterative refocusing' approach to find parameter combinations that have satisfactory performance over the NWUS while

also maintaining acceptable performance globally.

The PPE approach, in general, is extremely powerful in providing insights into model development and improvements. Parametric uncertainty is also one key area which remains relatively unexplored. So in my opinion, this paper makes an important contribution. The paper is well-structured and clear. The assumptions are clearly stated and the results sufficiently support the interpretations and conclusions. The abstract does provide a concise and complete summary. I also highly appreciated the fact that the authors clearly explain what roles each of the important parameters play in the model. Overall, I strongly feel that the paper is worthy of publication in GMD. I, however, have a few concerns that I would like the authors to address.

General comments:

Item 1: About the parameters perturbed: it is not made clear whether the parameters reside in the global model or the regional model. If the parameters belong to HadAM3P, which I think is the case, then the statement (l.206) "PPEs for parameter refinement with the aim of improving regional climate models (RCMs)" needs to be rephrased. The improvement is in terms of regional biases but the RCM in itself is not improved. If the parameters belong to the RCM, then I don't fully understand the global TOA flux constraints in Phase 1.

The purpose of using a regional model setup is unclear. Assuming that the parameters belong to HadAM3P and since the comparisons with observations are carried out at HadAM3P resolution, what role does the regional model (HadRM3) play here? In a topographically complex region such as the NWUS, model resolution likely plays an important role in model performance, but it's not discussed in the paper despite using a regional model. It also appears that the nesting is one-way wherein the RCM is nested within the global model, which suggests that any improvements in the RCM are not felt by the global model. These issues need clarification in the main text.

Item 2: The parameter refocusing is entirely carried out using AMIP-style simulations.

What are the implications of this for using these model variants in coupled model simulations? By perturbing parameters, the model response to future scenarios could be amplified or dampened relative to the standard version of the model, which may lead to differences in behavior between atmosphere-only and coupled model simulations. Is tuning a model in an AMIP mode for better regional performance useful for coupled model simulations?

Item 3: I wonder if the phrase 'perturbed parameter' should be used here instead of 'perturbed physics.' The authors perturb parameters instead of switching between schemes with different physics. For instance, Shiogama et al. (2014) generate multi-parameter, multi-physics PPEs by perturbing parameters in different versions of MIROC. So, while the words 'physics' and 'parameter' are used interchangeably in the context of PPEs, perhaps it is useful to make that distinction. [Shiogama, H., Watanabe, M., Ogura, T., Yokohata, T. and Kimoto, M., 2014. Multi‐parameter multi‐physics ensemble (MPMPE): a new approach exploring the uncertainties of climate sensitivity. Âă Atmospheric Science Letters, Âă 15(2), pp.97-102.]

Item 4: The authors argue that they use 'history matching' citing McNeall et al. and Williamson et al. Doesn't history matching require a formal statistical framework that is based on the definition of Implausibility? My reading of your approach is that it borrows ideas of from history matching (e.g., iterative refocusing) but doesn't strictly follow it. It would be useful to have a couple of sentences explaining the differences and similarities between your approach and history matching.

Item 5: (l.99) Biases in the regional model are shown in Figure S1, but it would be interesting to show what these biases look like in HadAM3P.

Item 6: Fig. S3 suggests that regional temperature and precipitation metrics are strongly anti-correlated across the PPE. Doesn't that suggest a physical link between these variables that can be exploited to find parameter combinations acceptable for both variables?

Item 7: I am a bit surprised that the sensitivity analysis shows very little interaction between the parameters especially at 10-year timescales. My reading of the paragraph starting at l.468 is that the small interaction terms are the property of your emulators. Is that the case? While entrainment coefficient and the ice fall speed are often the dominant parameters, I wonder if inadequate accounting of parameter interactions might prevent you from finding parts of parameters space that might be plausible.

Item 8: Fig. 5 – I am not sure what the purpose of Fig. 5 is since observations are not included in the figure. Yes, the selected 10 models show a substantially smaller spread compared to phases 1 and 2, but in JJA all the selected models show considerably different results compared to the SP and whether that's an improvement or not is not discussed in the text.

Item 9: Fig 6i indicates that the cold bias in DJF gets slightly worse in the PP simulations compared to SP even though there are improvements in the other seasons and regions. This needs to be mention in the paper since NWUS is the target region. On l.492, you do mention that the DJF-Pr bias worsens slightly in Phase 3. It suggests that reducing region/season-specific biases is difficult because of its implications for other seasons.

Item 10: Following on from the point above and the description on lines l.513-529 and l.662, it is clear that a more process-level evaluation is required to find parts of parameter space that truly provide improvements for right physical reasons. The authors should state explicitly (maybe at l.662) that looking at seasonal mean biases in temperature and precipitation is insufficient to fully assess model performance and a more process-based analysis could strengthen the validity of the chosen parameter sets.

Item 11: (Fig. 8) Why use only reanalysis datasets for precipitation? Aren't GPCC, CMAP, TRMM better observational precipitation products for model validation?

Item 12: (l.717) The number of initial conditions members, 10 or smaller, seems small especially because the performance is evaluated at regional scales and at climate

(∼10-year) timescales. Recent work has demonstrated that large initial conditions ensembles show very different trends on longer timescales. Can the authors please comment on how their parameter refinement might be affected by model's internal variability characteristics? Is variability in the model strongly coupled to the observed SSTs for the NWUS?

Minor comments:

l.381: It would be useful to have a brief explanation of what the 'ranges of acceptability' are for TOA fluxes here. I understand that this has been described in the Appendix and in Fig. 1 caption, but this is critical information and should be concisely described here.

l.392: missing 'a' in Yamazaki

Fig. S3 caption should read 'Same as Fig. S2'

l.423: should be 'lead to decreased outgoing . . .'

l.577: should 'increased' be 'improved'?

l.725: Please mention that you use LHS space-filling design earlier in the paper, perhaps at l.241.

It would be helpful to have a single document that contains all the supplementary figures and text.

---

## Referee Comment (RC2) · Anonymous Referee #2 · 14 Jan 2019

Review of "Improving climate model accuracy by exploring parameter space with an O(10^5) member ensemble and emulator" by Li et al.

General summary: The manuscript by Li et al applies an iterative, parameter-space refining procedure using perturbed parameter ensembles and statistical emulators to reduce regional biases of temperature and precipitation over the northwestern United States. The paper is well written and thorough, and provides a useful demonstration for reducing biases in the Hadley Centre climate models. I am support publication after the authors address the items below (listed in no particular order).

Item 1: The title seems to suggest that an ensemble of 100,000 forward simulations

was used to construct the emulators, which isn't the case. From Table 2, only a few thousand forward simulations were used. Once the emulators are trained, they can be evaluated very quickly, potentially millions to billions of times. The number of emulator evaluations O(10ˆ5) is therefore somewhat arbitrary and not significant. I recommend that the authors remove the reference to the emulator evaluations in the title because it is misleading. The authors should also consider adding information about the bias reduction goal of the study in the title (e.g. "Reducing climate model biases by exploring . . .").

Item 2: The first paragraph of the introduction describes the bias reduction goals of the study. On lines 53-54, I recommend that the authors change "simulations cast doubt on the reliability ..." to "simulations reduce the reliability ...". Later in the paragraph the authors describe prior work that found a relationship between the warm bias and shortwave radiation. This relationship motivates the need for a perturbed parameter approach (i.e. to quantify and reduce the bias). I recommend that the authors introduce PPEs in the first paragraph, rather than wait until the 5th paragraph.

Item 3: The introduction emphasizes parameter tuning as a major goal of PPEs, but using PPEs to estimate model PDFs and uncertainty is also an important application. I suggest that the authors include a statement about estimating PDFs versus parameter refinement.

Item 4: I recommend the following modification on line 146. Change "varied systematically." to "varied systematically or randomly."

Item 5: In the discussion about different types of PPE studies in the introduction, it would be worth pointing out that categories 2 and 3 may not be different from each other if a sufficient number of forward simulations are used to produce an adequate emulator over the full parameter space. With a good enough emulator, it is possible to both rule out parameter space and optimize parameter values. In this case, categories 2 and 3 are simply post-processing steps.

Item 6: Bayesian climate model calibration and MCMC are not mentioned or referenced in the introduction (e.g. Jackson et al., J. Clim. 2008), nor is optimization over multiple objectives (e.g. Neelin et al., PNAS, 2010). It would be worth including these references.

Item 7: The authors mention that little work has been done using PPEs for parameter refinement to improve RCMs. I agree with this statement, but think that it would be worth referencing prior work using PPEs for parameter refinement to improve regional climate in GCMs.

Item 8: Toward the end of the introduction, the authors describe how instead of searching for a single optimized parameter set, they consider multiple parameter sets because of the challenges of compensating errors and other effects. Rather than a limited number of parameter sets, it would be better if posterior parameter PDFs were estimated in a Bayesian sense. Doing so would understandably be outside the scope of the manuscript, though the authors should comment about the potential benefits of using parameter PDFs in their analysis.

Item 9: I would like the authors to comment about how they expect their results would differ if they swapped the order of phases 1 and 2 (i.e. first reduce biases in NWUS and then rule out regions that don't preserve energy balance).

Item 10: The authors use the so-called standard physics set (SP) as a reference point for gauging improvements from the PPE. Are the parameter values in the SP the same between the global and regional versions of the model? If so, there would appear to be a mismatch because the parameters represent unresolved processes and the standard values should be adjusted to account for differences in scale between the HadAM3P and HadRM3P. Following similar reasoning, it is difficult to see how parameter perturbations applied to the global model can be directly applied to the regional model without adjusting for scale differences. The authors should comment on and describe the implications of this potential mismatch.

Item 11: Observational uncertainty is assessed by using multiple observational datasets. For the regional bias analysis, how large are the datasets differences relative to the upscaling variability in regridding PRISM to HadRM3P?

Item 12: Presumably emulators are used for the sensitivity analysis, though this is not clear from the discussion in section 2.5. I recommend including a short summary of the emulators before the sensitivity analysis, rather than keeping all of the emulator discussion in the appendix. If emulators were used for the sensitivity analysis and are efficient to evaluate, so why not use a quantitative Sobol analysis instead of the qualitative FAST method?

Item 13: When I first read section 3.1 and interpreted the results in figure 1, I thought that the phase 2 emulator errors for SW were significant. Only later, after seeing the quality of the emulators in appendix B, did I realize that the errors were smaller than I thought. I recommend that the authors summarize the quality of the emulators earlier in the manuscript. Referring to figure 1, can the authors also comment about why there does not appear to be a very strong relationship between the LW and SW points (correlation looks like 0) and why there are no simulations in the blue ellipse with high LW?

Item 14: The OAAT relationships in figure 2 are useful as qualitative indicators of the dependence of the outputs on the inputs. While it reasonable to hold other inputs at their mean values, it would probably be more informative to use the default SP values. Can the authors regenerate the plots using SP values instead? Moreover, instead of conditioning on the values of the other inputs, it may be more useful to compute partial dependence plots that integrate over the other inputs. Unlike the OAAT plots, partial dependence plots account for interactions.

Item 15: Global sensitivity indices are presented in figure 3 using the FAST method. The text mentions that total and main effects are both computed, and that the maximum sensitivity value is 1. For nonlinear models, the sum of the total effects can be greater

than 1 because interactions are counted more than once. Can the authors clarify their statement? Also, I am wondering about the robustness of some of the differences between the sensitivity indices (e.g. for JJA-Pr). If emulators were used, can the authors estimate and include emulator uncertainty in figure 3?

Item 16: Figure 4 is highly useful, but contains a lot of information. It might be easier to digest in three separate figures (input-input, input-output, and output-output). I am also wondering about some of the differences between the phase 3 and SP values. For the input-input plots, the SP red dots tend to lie near or within the phase 3 set of points (there are exceptions for CW_land, ENTCOEF and V_CRIT_ALPHA). In the MAC-T-input and JJA-T-input plots, however, the SP and phase 3 points are completely separated from each other. It looks like the separation can be explained by ENTCOEF and V_CRIT_ALPHA, but the magnitude of the separation seems too large. For MAC-T it looks like the average difference is about 2 degrees C. But the temperature change in figure 1 is about 1 degree C when ENTCOEF is varied between 3 and 5. Can the authors explain why small changes in ENTCOEF and V_CRIT_ALPHA lead to such large (and discrete) changes in temperature? Understandably, one of the goals of the study is to reduce the temperature bias, so large changes might be expected. But the authors use an iterative and refinement strategy that I would expect to reduce the bias in small continuous steps, not the large, discrete changes that are shown.

Item 17: Given that ENTCOEF and V_CRIT_ALPHA are dominant parameters affecting temperature, it might be useful to use the emulators to further analyze and display the temperature surfaces as a function of the inputs.

Item 18: On the copy of the manuscript that I reviewed, there appears to be an artifact (i.e. a white line) at longitude 0 on all of the spatial maps (e.g. see figure 6k over Eastern Africa).

Item 19: After refining the parameter space between phases 2 and 3, the parameters that were dominant in figure 3 may no longer be dominant. Can the authors recompute

the sensitivity indices for the refined parameter space after the bias reduction?

**[GMDD](GMDD)**

---

## Author Comment (AC1) · 18 Feb 2019

Response to Referee #1

General response:

Thank you very much for these comments. We feel very encouraged from this review and have made the best attempt to respond to these constructive comments.

Response to item 1:

We thank the reviewer for these constructive comments and agree with the assessment that these issues should be clarified in the main text. We have added a few sentences in

the main text (in Section 2.3 Perturbed parameters) to clarify the parameters perturbed " the parameters reside in the global model as well as the regional model, and are set to the same values in HadAM3P and HadRM3P in the experiments used in this study, thus any improvement in regional biases is considered to have been improved through the improvement of the RCM itself".

The RCM is embedded within the GCM, and nesting is one-way where the GCM provides boundary conditions for the RCM. The purpose of the global TOA flux constraints in Phase 1 was to make sure the large scale boundary conditions for the RCM are realistic. Within weather@home the purpose of the regional model is to provide higher resolution output over the area of interest, rather than to feedback scale dependent features to the GCM. The comparisons with observations are carried out at both HadAM3P resolution and HadRM3P resolution, and the regional model biases are calculated with respect to PRISM datasets. We have changed the wording in the main text (l.333 in the original manuscript), which now reads "biases of the regional model outputs are calculated with respect to PRISM". We agree with the reviewer that, in a topographically complex region such as the NWUS, model resolution does play an important role in model performance, and perhaps the model parameters should be resolution dependent, i.e., they should be set to different values in the GCM and RCM. This is indeed an important issue that needs to be further explored. In follow-on work, we have performed additional PPEs, where the parameter values are set to be different in HadAM3P and HadRM3P, and we will attempt to address the resolution-dependency of parameter values in a following paper using those PPEs.

Response to item 2:

Thank you for these comments. We agree with the reviewer that by perturbing parameters, the model response to future scenarios could be different from the standard version of the model. As part of our ongoing work, we are looking at the different climate sensitivities of different model variants in future projections. However, it was never the intention of these experiments to come up with model variants to be used in

coupled model simulations. Parametric uncertainty exists in the ocean component as well. If an ocean component were to be coupled to the tuned AMIP mode, a systematic parameter refocusing would be required to ensure consistency across the full climate system. It is likely that different atmospheric parameter settings would be selected due to potential interactions with oceanic parameters.

Response to item 3:

Thank you for pointing this out. We agree the phrase "perturbed parameter" should be used here instead of "perturbed physics", since the latter could be interpreted as switching different physics schemes rather than change parameter values. We have changed the main text throughout to only use the phrase "perturbed parameter".

Response to item 4:

We appreciate this comment and have added a couple of sentences at the end of paragraph 10 to explain the differences and similarities between our approach and history matching. Now it reads "The method we adopt in this study borrows the idea of 'iterative refocusing' from the third category, where the parameter values are refined through phases of experiments, but our method differs from history matching in that we do not employ a formal statistical framework based on the definition of implausibility". We believe now it's much clearer in the main text that the approach adopted in our study does not strictly follow history matching.

Response to item 5:

We respectfully point out that the temperature and precipitation biases in HadAM3P are shown in Figure 6 and Figure 7 respectively.

Response to item 6:

Thank you for these comments. We agree with the reviewer that regional temperature and precipitation are strongly anti-correlated in JJA, and this does suggest a physical link that can be exploited to find parameter combinations acceptable for both variables. Multivariate parameter sensitivity is indeed one of the research questions we will explore with our additional PPEs, which are follow up experiments of our current manuscript.

Response to item 7:

The interaction terms are not the property of emulators. In the extended Fourier Amplitude sensitivity analysis (FAST; Saltelli et al., 1999), the fraction of the total variance due to interactions is computed from the parameter contribution to the residual variance (variance not accounted for by the main effects). The relative importance of the interaction term is dependant on the metrics of interest. For example, the interaction term is non-trivial for the root mean squared error of JJA 850-hPa U (V) wind component in PP simulations over the regional model domain, with respect to SP simulations, as seen in Figure 1 and Figure 2 below.

Figure 1. Sensitivity analysis of the root mean squared error of JJA 850-hPa U wind component in PP simulations over the regional model domain, with respect to SP simulations, via the FAST algorithm of Saltelli et al. (1999).

Figure 2. Sensitivity analysis of the root mean squared error of JJA 850-hPa V wind component in PP simulations over the regional model domain, with respect to SP simulations, via the FAST algorithm of Saltelli et al. (1999).

Response to item 8:

We appreciate this comment and have added a few sentences to clarify about Fig. 5 " The SP simulations have warm and dry biases over NWUS and mid-latitude land in general (as shown in Fig. 4, Fig. 6 and Fig. 7). In JJA all the selected PP model variants show considerably different results compared with the SP-cooler and wetter, i.e. reduced biases and improved model performance". We have also updated Fig. 5 to include the results from different initial conditions to demonstrate that varying model parameters has more influence on the result than varying initial conditions, which helps

clarify item 12 as well.

Response to item 9:

Respectfully, Fig. 6i shows that the PP simulations are warmer than the SP, which indicates the cold biases in DJF gets slightly better in the PP simulations. We agree with the reviewer that reducing region/season-specific biases is difficult because of its implications for other seasons and regions. Parameter refinement exercises like ours are faced with the common dilemma where model revisions yield improvement in one field or regional pattern or season, but degradation in another, or improvement in the model climatology but degradation in the interannual variability representation etc. That is why we feel strongly that any parameter refinement process is and should be tailored to the scientific objectives of the experiments, because there is no one fit for all solution; more importantly, that's why we are advocating that parameter refinement process should be more explicit and transparent, because choices and compromises made during the refinement process may significantly affect model results and influence evaluations against observed climate, hence should be taken into account in any interpretation of model results, especially in multi-model intercomparison studies to help understanding of model differences.

Response to item 10:

We appreciate this comment and have added a few sentence as suggested to explicitly state this point "Furthermore, looking at biases in seasonal mean temperature and precipitation is insufficient to fully assess model performance. As a follow-up step to this study, we recommend a process-based model evaluation and physical explanation of model improvements to further refine the parameter space that provides improvements (e.g., reduce summer biases) for the right physical reasons. For example, more accurate representation of clouds in the model could lead to better simulated downward solar radiation at the surface, as well as better simulated surface energy and water balance."

Response to item 11:

In the main text, we showed model outputs in comparison with CRU datasets, simply so that temperature and precipitation are compared with the same data source. In the supplementary information, we showed the biases of precipitation compared with other datasets, which includes GPCC (Fig. S18). Now we have added biases of precipitation compared with CMAP and TRMM in the supplementary information of the revised manuscript.

Response to item 12:

Thank you for these comments regarding initial conditions. Indeed our original intent of having multiple ICs is so that the results would be representative of the parameter perturbations instead of reflecting the influence of any particular IC. Previous work (Bellprat et al., 2011, Figs. 1-4; Covey et al., 2011, Fig. 4) suggests that varying model parameters has more influence on climate than varying initial conditions. For the metrics we used (e.g. seasonal mean temperature and precipitation), our results show the same in the updated Fig. 5. To answer if internal variability affects different parameter sets differently, we will need to generate identical large initial condition ensembles for each parameter set, which is an interesting research question but beyond the scope of this study.

In previous work (Li et al., 2015) we have compared HadRM3P-HadAM3P results with NARCCAP, and found that that the coupled HadRM3–HadCM3 (HRM3-HadCM3 in NARCCAP convention) demonstrated similar skills in simulating temperature and precipitation as HadRM3P– HadAM3P, even though HadAM3P is an atmosphere only model and SSTs are specified whereas HadCM3 is a coupled ocean–atmosphere model. This similarity between HadRM3P–HadAM3P and HadRM3–HadCM3 suggests that the dynamical coupling between ocean and atmosphere in NARCCAP did not explain most of the difference between HadRM3P–HadAM3P and the various NARCCAP RCM–GCM pairings but that the differences were due mainly to the atmospheric

dynamics.

Minor comments:

Response to comment on l.381:

Thank you for pointing this out. Indeed it would be helpful to add some explanation of 'ranges of acceptability' in the main text. We have added a brief explanation of what the 'ranges of acceptability' at the beginning of Section 3.1 in the revised manuscript.

Response to comment on l.392:

Apologies for the sloppiness here, this has been fixed.

Response to comment on l.423:

This has been changed.

Response to comment on l.577:

We respectfully point out that in SP, the model has dry biases over northern South America, equatorial Africa, and south Asia in JJA. In PP simulations, the dry biases are stronger compared with SP (Fig. 6k), so the biases are increased in PP simulations.

Response to comment on l.725:

Thank you for pointing this out. We have now stated earlier in the paper (l.241 in the original manuscript) that we used LHS space-filling design.

Response to minor comment on single document that contains all the supplementary figures and text:

We will upload a single document containing all the supplementary figures and text as revised manuscript.

Reference:

Bellprat, O., Kotlarski, S., Lüthi, D. and Schär, C., 2012. Exploring perturbed physics

ensembles in a regional climate model. Journal of Climate, 25(13), pp.4582-4599.

Covey, C., Brandon, S., Bremer, P.T., Domyancis, D., Garaizar, X., Johannesson, G., Klein, R., Klein, S.A., Lucas, D.D., Tannahill, J. and Zhang, Y., 2011. A new ensemble of perturbed-input-parameter simulations by the Community Atmosphere Model. Technical report, Lawrence Livermore National Laboratory, Livermore, CA.

Li, S., Mote, P.W., Rupp, D.E., Vickers, D., Mera, R. and Allen, M., 2015. Evaluation of a regional climate modeling effort for the western United States using a superensemble from weather@ home. Journal of Climate, 28(19), pp.7470-7488.

[Figure]

**Fig. 1.**

[Figure]

**Fig. 2.**

---

## Author Comment (AC2) · 7 Mar 2019

General response: Thank you very much for these comments. We feel very encouraged from this review and the similar encouraging comments from the other reviewer. We have made the best attempt to respond to these constructive comments.

Response to item 1: Thank you for this suggestion. The title has now been changed to "Reducing climate model biases by exploring parameter space with large ensembles of climate model simulations and statistical emulation".

Response to item 2: Thank you for these suggestion. Lines 53-54 now reads 'reduce

the reliability of' as suggested. In the revised manuscript we have moved the previous 5th paragraph which introduces parameter perturbation to earlier in the paper (2nd paragraph now) as suggested.

Response to item 3: Thank you for this suggestion. In the revised manuscript, we have added a few sentences about using PPEs to estimate model PDFs and uncertainty at the end of paragraph 7 "Besides parameter refinement, PPEs have also been used in many studies to estimate probability distribution functions (PDFs) of equilibrium climate sensitivity (e.g., Murphy et al., 2004) and transient regional climate change (e.g., Sexton et al., 2012), permitting probabilistic projection of climate change (Murphy et al., 2007, 2009; Harris et al., 2013). PPEs are becoming common as a means to assess the range of uncertainty in climate model projections (Murphy et al., 2004; Stainforth et al., 2005; Collin et al., 2006; Sanderson, 2011; Sexton et al., 2012; Shiogama et al., 2012)".

Response to item 4: Thank you for this suggestion. The wording on line 146 has been changed to "varied systematically or randomly" as suggested.

Response to item 5: Thank you very much for this suggestion. We have added a few sentences in the main text to point this out at the end of paragraph 10: "It is worth pointing out that the second and the third categories may not be different from each other if a sufficient number of model simulations are used to train a statistical emulator over the full parameter space. With a good emulator, it is possible to rule out parameter space and optimize parameter values, in which case categories two and three are post-processing steps".

Response to item 6: Thank you for pointing this out. In the revised manuscript, we have included a few references on Bayesian climate model calibration and MCMC, as well as optimization over multiple objectives in the introduction as suggested.

Response to item 7: Thank you for this suggestion. We have added a few references of prior work using PPEs for parameter refinement to improve regional climate over

Europe and North America in paragraph 11 (line 207 in the original manuscript), which now reads "However, very little (Bellprat et al., 2012; 2016) has been published on using PPEs for parameter refinement with the aim of improving regional climate models (RCMs)."

Response to item 8: Thank you for this suggestion. We have added a few sentences commenting on the potential benefits of using posterior parameter PDFs toward the end of the introduction.

Response to item 9: If we were to swap the order of phases 1 and 2, we could possibly get rid of the regional temperature biases, seeing the reduction in temperature biases is accompanied by increased TOA reflected SW radiation (Fig. 4), implying there are more clouds in those PPE simulations with reduced JJA temperature biases. Then we could end up in a corner of the parameter space where there is minimal JJA temperature biases, but out of balance TOA energy fluxes. Our premises was that TOA radiation balance is an emergent property in GCMs (Solomon et al., 2007), so we chose to carry out the parameter refining process following phases 1 and 2, preserving energy balance first, then reducing biases in NWUS.

Response to item 10: Thank you for these comments. Some model parameters are different between the global and regional model (adjusted for scale) but these are not among the parameters that were perturbed in this study. We are mindful of the possibility that among the parameters that were perturbed here, for some parameters the parameter values may be resolution-dependent, especially in a topographically complex region such as the NWUS. This is an important issue that needs to be further explored. However, currently there is no clear guidance on how the parameters should scale with changes in resolution between the global and regional model. Without information on how these should be adjusted or performing a further nested parameter sweep (which is beyond the scope of this study), it would be hard to know which parameters to adjust, to what extent are they resolution dependent, and how to adjust them. Therefore, the same values are applied in HadAM3P and HadRM3P as a first
estimation, without adjustment to account for differences in scale. But it would be an interesting and useful follow-on study to look at setting parameters differently in the global and regional model. As part of our ongoing work, we have performed additional PPEs, where the parameter values are set to be different in HadAM3P and HadRM3P, and we will attempt to address the resolution-dependency of parameter values in a following paper using those PPEs.

Response to item 11: We are not quite sure what the reviewer means by 'upscaling variability'. Assuming that means variability in upscaling methods in regridding PRISM to HadRM3P, to clarify, we chose the method where for each HadRM3P grid point, an average was taken over all the PRISM points that fall in the bounds of that HadRM3P grid point, and the averaged value was assigned to that HadRM3P grid point.

Response to item 12: Thank you for these comments and suggestions. Yes, emulators are used for sensitivity analysis. Apologies for not being clear about this. We have added a sentence in the sensitivity analysis section to clarify this "Emulators are used for the sensitivity analysis". As suggested, we have included a brief summary of the emulators (the new section 2.5. Emulators) before the sensitivity analysis (the previous section 2.5, which is now section 2.6).

FAST is also a variance-based quantitative sensitivity analysis method. The fraction of the variance due to an input parameter (main effect) is calculated as the sum of the Fourier coefficients for the frequency assigned to the input parameter and it's harmonics. The total contribution of each parameter, xi, to the output variance includes main effects and interactions with all other parameters and can be calculated by summing Fourier coefficients for the set of frequencies complementary to the frequency assigned to input the parameter xi. The residual variance, not accounted for by the main effect, is therefore attributable to interactions between the parameter xi and any of the other parameters. We agree with the reviewer that Sobol analysis could be used, but FAST is a quantitative method that can be used to identify the dominant parameters as well.

Response to item 13:

Thank you for these comments and suggestions. We have added a few sentences to summarize the quality of the emulators earlier in the manuscript (section 3.1) as suggested. Regarding the relationship between the LW and SW, the bias reduction is accompanied by reduced TOA reflected SW, suggesting changes in clouds, but given different cloud types have different radiative effects on SW and LW (Zelinka et al., 2012, Fig. 8), we do not expect a clear positive or negative correlation between SW and LW by increased clouds, without knowing if the increased clouds are high, medium, low, thin, medium, thick, or any combination of these cloud types. We suspect the reason why there are no simulations in the blue ellipse with high LW is because the net effects of changes in clouds in these PP experiments are so that they do not lead to too much infrared radiation emitting to space. To answer this in further detail, we would need to run additional experiments where detailed could covers (low, medium and high) and cloud properties (e.g., optical depth) are saved as outputs, which is beyond the scope of this study.

Response to item 14: Thank you for these comments and suggestions. We have regenerated the OAAT plot using SP values as suggested (please see Figure 1). The results are very similar to those shown in the main Figure 2 holding other inputs at their mean values.

Figure 1. One-at-a-time sensitivity analysis of magnitude of MAC-T over Northwest to each input parameter in turn, with all other parameter held at SP values. Heavy lines represent the emulator mean, and shaded areas represent the estimate of emulator uncertainty, at the $\pm 1$ SD level.

Thank you for suggesting to use partial dependence plots. We have attempted to compute and plot partial dependence plots (using scikit-learn in python https://scikit-learn.org/stable/). Figure 2 shows the results for the four most important parameters - cw_land, vf1, entcoef, and v_crit_alpha. The results are very similar to those shown in

the OAAT plots, and the results for the other parameters are very similar as well (not shown here). Since the interaction terms are quite small (from the sensitivity analysis shown in Figure 3 in the main text), it is not that surprising that the OAAT plots and the partial dependence plots are similar.

Figure 2. Partial dependence plots of magnitude of MAC-T over Northwest to CW_LAND, VF1, ENTCOEF, and V_CRIT_ALPHA.

Response to item 15: We agree with the reviewer that non-linear models can have coefficients that sum to greater than one. However, in the extended Fourier Amplitude sensitivity analysis (FAST; Saltelli et al., 1999), the fraction of the variance due to an input parameter (main effect) is calculated as the sum of the Fourier coefficients for the frequency assigned to the input parameter and it's harmonics. The total contribution of each parameter, xi, to the output variance includes main effects and interactions with all other parameters and can be calculated by summing Fourier coefficients for the set of frequencies complementary to the frequency assigned to input the parameter xi. The residual variance, not accounted for by the main effect, is therefore attributable to interactions between the parameter xi and any of the other parameters. The fraction of the total variance due to interactions is not resolved as the sum of individual effects and will never sum to a value other than 1. For further information please refer to Saltelli et al. (1999).

We have added a few sentences in 'Sensitivity Analysis' section to clarify this "In the FAST method, the fraction of the total variance due to the interactions is not resolved as the sum of individual interactions, but is computed from the parameter contribution to the residual variance , i.e., variance not accounted for by the main effects."

Regarding the emulator uncertainty, Figure 2, which shows the emulator uncertainty as the shaded area in each panel, illustrates that the contribution of the emulator uncertainty to the variance of the emulated output is small.

Response to item 16: Thank you for these comments. We agree with the reviewer that

Figure 4 contains a lot of information. In the original figure, we added the horizontal and vertical red lines to mark the transition from parameter inputs to model output metrics, hoping that would make digesting the figure easier. In the updated Fig 4 (which is shown in Figure 3 below), we have added additional labels in the figure to mark the three quadrants of the figure as a) input-input, b) input-output, and c) output-output.

Response to item 17: Thank you for this suggestion. We have plotted the temperature surfaces (MAC-T, JJA-T, and DJF-T) as a function of each pair of parameter inputs using the emulators (please see Figures 4-6).

Figure 4. MAC-T biases projected into the two-dimensional spaces of each pair of input parameters using the emulator.

Figure 5. JJA-T biases projected into the two-dimensional spaces of each pair of input parameters using the emulator.

Figure 6. DJF-T biases projected into the two-dimensional spaces of each pair of input parameters using the emulator.

Response to item 18: Thank you for point this out. This seems like an artifact from the original plotting. We will update the spatial maps to in the revised manuscript.

Response to item 19: Thank you for these comments. We have recomputed and re-plotted the sensitivity indices for the refined parameter space in phase 3 as suggested (please Figure 7, and this figure has been added to the supplementary information of the revised manuscript). The dominant parameters for SW, LW, and DJF-Pr are still the same, whereas the dominant parameters for the other output metrics are different. V_CRIT_ALPHA becomes the most important parameter for MAC-T, JJA-T and JJA-Pr, and ASYM_LAMBDA becomes the most important parameter for DJF-T. The dominant parameter is partially a function of the plausible range set for that parameter. We agree with the reviewer that the parameters that were well constrained after phase 2 may not be dominant in phase 3 simply because the perturbation range was reduced

after phase 2.

References: Bellprat, O., Kotlarski, S., Luìĺthi, D., and Schaìĺr, C.: Objective cali-
bration of regional climate models, Journal of Geophysical Research: Atmospheres,
117(D23), https://doi.org/10.1029/2012JD018262, 2012. Bellprat, O., Kotlarski, S.,
Luìĺthi, D., De EliìĄa, R., Frigon, A., Laprise, R., and Schaìĺr, C.: Objective calibra-
tion of regional climate models: application over Europe and North America, Journal of
Climate, 29(2), 819-838, https://doi.org/10.1175/JCLI-D-15-0302.1, 2016. Collins, M.,
Booth, B.B., Harris, G.R., Murphy, J.M., Sexton, D.M. and Webb, M.J.: Towards quanti-
fying uncertainty in transient climate change. Climate Dynamics, 27(2-3), pp.127-147,
2006. Harris, G.R., Sexton, D.M., Booth, B.B., Collins, M. and Murphy, J.M.: Proba-
bilistic projections of transient climate change. Climate dynamics, 40(11-12), pp.2937-
2972, 2013. Murphy, J.M., Sexton, D.M., Barnett, D.N., Jones, G.S., Webb, M.J.,
Collins, M. and Stainforth, D.A.: Quantification of modelling uncertainties in a large
ensemble of climate change simulations. Nature, 430(7001), p.768, 2004. Murphy,
J.M., Booth, B.B., Collins, M., Harris, G.R., Sexton, D.M. and Webb, M.J.: A method-
ology for probabilistic predictions of regional climate change from perturbed physics
ensembles. Philosophical Transactions of the Royal Society A: Mathematical, Physi-
cal and Engineering Sciences, 365(1857), pp.1993-2028, 2007. Murphy, J.M., Sexton,
D.M., Jenkins, G.J., Booth, B.B., Brown, C.C., Clark, R.T., Collins, M., Harris, G.R.,
Kendon, E.J., Betts, R.A. and Brown, S.J.: UK climate projections science report: cli-
mate change projections, 2009. Sanderson, B.M.: A multimodel study of parametric
uncertainty in predictions of climate response to rising greenhouse gas concentrations.
Journal of Climate, 24(5), pp.1362-1377, 2011. Sexton, D.M. and Murphy, J.M.: Mul-
tivariate probabilistic projections using imperfect climate models. Part II: robustness
of methodological choices and consequences for climate sensitivity. Climate Dynam-
ics, 38(11-12), pp.2543-2558, 2012. Shiogama, H., Watanabe, M., Yoshimori, M.,
Yokohata, T., Ogura, T., Annan, J.D., Hargreaves, J.C., Abe, M., Kamae, Y., O'ishi, R.
and Nobui, R.: Perturbed physics ensemble using the MIROC5 coupled atmosphere–
ocean GCM without flux corrections: experimental design and results. Climate dynamics, 39(12), pp.3041-3056, 2012. Solomon, S., Qin, D., Manning, M., Averyt, K. and Marquis, M. eds.: Climate change 2007-the physical science basis: Working group I contribution to the fourth assessment report of the IPCC (Vol. 4). Cambridge university press, 2007. Stainforth, D.A., Aina, T., Christensen, C., Collins, M., Faull, N., Frame, D.J., Kettleborough, J.A., Knight, S., Martin, A., Murphy, J.M. and Piani, C.:. Uncertainty in predictions of the climate response to rising levels of greenhouse gases. Nature, 433(7024), p.403, 2005. Zelinka, M.D., Klein, S.A. and Hartmann, D.L.: Computing and partitioning cloud feedbacks using cloud property histograms. Part I: Cloud radiative kernels. Journal of Climate, 25(11), pp.3715-3735, 2012.

[Figure]

**Fig. 1.** One-at-a-time sensitivity analysis of magnitude of MAC-T over Northwest to each input parameter in turn, with all other parameter held at SP values. Heavy lines represent the emulator mean, and shaded

[Figure]

**Fig. 2.** Partial dependence plots of magnitude of MAC-T over Northwest to CW_LAND, VF1, ENTCOEF, and V_CRIT_ALPHA.

[Figure]

**Fig. 3.** Updated Figure 4 of the main text.

[Figure]

**Fig. 4.** MAC-T biases projected into the two-dimensional spaces of each pair of input parameters using the emulator.

**Fig. 5.** JJA-T biases projected into the two-dimensional spaces of each pair of input parameters using the emulator.

**Fig. 6.** DJF-T biases projected into the two-dimensional spaces of each pair of input parameters using the emulator.

[Figure]

**Fig. 7.** The sensitivity indices for the refined parameter space in Phase 3.